# On Kernel RL without Optimistic Closure

## Abstract

We study episodic reinforcement learning with a kernel (RKHS) structure on state-action pairs. Previous optimistic analyses in this case either pay a data-dependent covering-number penalty that can grow with time and undermine no-regret guarantees, or it assumes a strong "optimistic closure" condition requiring all optimistic proxies to lie in a fixed state-RKHS ball. We take a different approach that removes the covering-number dependence without invoking optimistic closure. Our analysis builds a uniform confidence bound, derived via conditional mean embeddings, that holds simultaneously for all proxy value functions within a bounded state-RKHS class. We introduce **KOVI-Proj**, an optimistic value-iteration scheme that explicitly projects the optimistic proxy back into the state-RKHS ball at every step, ensuring that the uniform bound applies throughout the learning process. Under a restricted Bellman-embedding assumption (bounded conditional mean embeddings), KOVI-Proj enjoys a high-probability no-regret guarantee whose rate is governed by the task horizon and the kernel's information gain. When the optimal value function lies in the chosen state-RKHS ball (realizability), the regret is sublinear; in the agnostic case, an explicit approximation term reflects the best RKHS approximation error. Overall, this work provides a new pathway to no-regret kernel RL that is strictly weaker than optimistic closure and avoids covering-number penalties. Numerical experiments validate our claims.

## 1 Introduction

Kernel-based function approximation are known to provide an interesting link between linear models and the behavior of infinitely wide neural networks. However, obtaining sharp, no-regret (let alone order-optimal) guarantees in kernel-based reinforcement learning (RL) remains challenging (Vakili, 2024). Previous optimistic analyses in RKHSs typically follow one of the two approaches: (i) apply a union bound over a *data-dependent, evolving* class of optimistic value proxies, thereby incurring a covering-number penalty that can scale in the order of $\Omega(\sqrt{T})$ and spoil no-regret for common kernels (e.g., kernelized optimistic LSVI: Least Squares Value Iteration (Yang et al., 2020)); or (ii) assume a strong *optimistic closure* property stating that *every* optimistic proxy already contained in a fixed state-RKHS ball (as found in CME-based optimistic RL) (Chowdhury & Oliveira, 2023). The former is statistically loose; the latter is structurally strong and not obviously aligned with standard optimistic constructions.

We take a different approach based on *uniform concentration without covering*. The key observation is that for any $V$ in a state RKHS $\mathcal{H}_\ell$, the Bellman image $[P_h V]$ can be written as an inner product

$$[P_h V](z) = \langle \mu_h(z), V \rangle_{\mathcal{H}_\ell},$$

where $\mu_h : \mathcal{Z} \to \mathcal{H}_\ell$ is the *conditional mean embedding* (CME) of the next-state distribution (Muandet et al., 2017b; Song et al., 2013; Muandet et al., 2017a). When $\mu_h$ is contained in an appropriate vector-valued RKHS over $\mathcal{Z}$ with bounded norm, the map $V \mapsto [P_h V]$ is a bounded linear operator from $(\mathcal{H}_\ell, \|\cdot\|_{\mathcal{H}_\ell})$ to $(\mathcal{H}_k, \|\cdot\|_{\mathcal{H}_k})$ (Carmeli et al., 2010). This viewpoint lets us control, via a single vector-valued regression problem, the Bellman images $[P_h V]$ *for all $V$* in the state ball $\{V : \|V\|_{\mathcal{H}_\ell} \leq B\}$ *simultaneously*, yielding a uniform kernel-ridge confidence bound with *no* data-dependent covering (leveraging information-gain / elliptical-potential tools standard in kernel bandits) (Chowdhury & Gopalan, 2017). Importantly, we *enforce* the bounded-norm property algorithmically by projecting the optimistic proxy value onto the state-RKHS ball each step. Experimental results show promise of the proposed method, and tends to shows lower regret than the baselines.

**Constibutions of this paper is listed as follows:**

(1) **Restricted Bellman-embedding assumption (RBE).** We use a mild assumption under which the CME $\mu_h$ belongs to the vector-valued RKHS on $\mathcal{Z}$ with kernel $k\,I$ and norm at most $U$. This is strictly weaker than optimistic closure (which presumes *all* optimistic proxies already lie in a fixed state-RKHS ball), and it is natural under standard CME regularity (Muandet et al., 2017b; Carmeli et al., 2010).

(2) **Uniform confidence without covering.** We prove a high-probability bound

$$\sup_{\|V\|_{\mathcal{H}_\ell} \leq B} \left|[P_h V](z) - \widehat{f}_{h,n}^V(z)\right| \;\leq\; \beta_{h,n}\,\sigma_{h,n}(z),$$

holding for all $z$ and all $V$ in the ball, where $\widehat{f}_{h,n}^V$ is the kernel-ridge predictor trained on labels $V(s')$ and $\sigma_{h,n}$ is the posterior standard deviation. The multiplier satisfies

$$\beta_{h,n} \;=\; B\,U \;+\; \frac{B\,\sigma}{\sqrt{\rho}}\,\sqrt{2\gamma(n,\rho) + 2\log(1/\delta)},$$

depending on the ball radius $B$, operator norm $U$, sub-Gaussian scale $\sigma$, ridge parameter $\rho > 0$, and (regularized) information gain $\gamma(n,\rho)$-*but it does not depend* on any covering number of the proxy class (Chowdhury & Gopalan, 2017).

(3) **KOVI-Proj: Kernel-Optimistic Value Iteration with projection.** We propose a practical optimistic method that (i) performs kernel-ridge backups to estimate $[P_h V]$, (ii) adds an uncertainty bonus $\beta_{h,n}\sigma_{h,n}$, and (iii) *the method projects optimistic value proxy* onto the state-RKHS ball (with clipping), thereby guaranteeing $\|V\|_{\mathcal{H}_\ell} \leq B$ and placing all proxies within the scope of the uniform bound above.

(4) **No-regret guarantee.** Under the realizability ($V_h^* \in \mathcal{H}_\ell(B)$) assumption, the proposed KOVI-Proj method attains

$$R(T) \;=\; \tilde{\mathcal{O}}\Big(H^2\,B\Big(U + \tfrac{\sigma}{\sqrt{\rho}}\sqrt{\gamma(HT,\rho)}\Big)\sqrt{T\,\gamma(HT,\rho)}\Big),$$

which is sublinear for kernels with sublinear information gain (e.g., Matérn/Squared-Exponential under regularization) Chowdhury & Gopalan (2017). In the agnostic case ($V_h^* \notin \mathcal{H}_\ell(B)$), we add an explicit approximation term of order $HT\,\varepsilon_B$, where $\varepsilon_B := \max_h \sup_{\|V\|\leq B} \|V_h^* - V\|_\infty$, and we show how a slowly growing $B = B_T$ balances both terms to remain $o(T)$.

## 2 PROBLEM SETUP AND ASSUMPTIONS

We consider an episodic MDP $M = (\mathcal{S}, \mathcal{A}, H, P, r)$ with horizon $H \in \mathbb{N}$. Let $\mathcal{Z} = \mathcal{S} \times \mathcal{A}$. For step $h \in [H]$, transition kernel is $P_h(\cdot\,|\,z)$ on $\mathcal{S}$ is unknown. We take rewards $r_h : \mathcal{Z} \to [0,1]$ to be known and deterministic for clarity;[1] for a policy $\pi$ and step $h$, we have

$$Q_h^\pi(z) \;=\; r_h(z) + \mathbb{E}_{s' \sim P_h(\cdot|z)}[V_{h+1}^\pi(s')], \qquad V_h^\pi(s) \;=\; \max_{a \in \mathcal{A}} Q_h^\pi(s,a), \qquad V_{H+1}^\pi \equiv 0$$

The per-episode regret will be measured against optimal value $V_1^*$ as follows

$$R(T) \;:=\; \sum_{t=1}^{T} \left(V_1^*(s_{1,t}) - V_1^{\pi_t}(s_{1,t})\right).$$

**RKHS structure on $\mathcal{Z}$ and on $\mathcal{S}$.** Let $k : \mathcal{Z} \times \mathcal{Z} \to \mathbb{R}$ be a positive-definite kernel with RKHS $(\mathcal{H}_k, \|\cdot\|_{\mathcal{H}_k})$ and $k(z,z) \leq \kappa_k^2$. Let $\ell : \mathcal{S} \times \mathcal{S} \to \mathbb{R}$ be a positive-definite kernel with RKHS $(\mathcal{H}_\ell, \|\cdot\|_{\mathcal{H}_\ell})$ and let $\ell(s,s) \leq \kappa_\ell^2$. We plan to use kernel ridge regression (KRR) on $\mathcal{Z}$ and consider proxy value functions $V : \mathcal{S} \to \mathbb{R}$ in $\mathcal{H}_\ell$.

The following assumption is novel to our work, but is inspired by the conditional mean embedding literature (Muandet et al., 2017b) and the theory of vector-valued RKHSs (Carmeli et al., 2010).

---

[1] The extension to unknown (possibly stochastic) rewards can be handled with an additional KRR estimator and a union bound; see the discussion section.

**Assumption 2.1** (Restricted Bellman-embedding (RBE)). *For each $h \in [H]$ assume that there exists a conditional mean embedding $\mu_h : \mathcal{Z} \to \mathcal{H}_\ell$ such that*

$$[P_h V](z) := \mathbb{E}_{s' \sim P_h(\cdot | z)}[V(s')] = \langle \mu_h(z), V \rangle_{\mathcal{H}_\ell} \qquad \text{for all } V \in \mathcal{H}_\ell \text{ and all } z \in \mathcal{Z} \quad (1)$$

*and $\mu_h$ belongs to the vector-valued RKHS over $\mathcal{Z}$ with operator-valued kernel $K(z, z') = k(z, z') I_{\mathcal{H}_\ell}$, with $\|\mu_h\|_{\mathcal{H}_k \otimes \mathcal{H}_\ell} \leq U$.*

**Remark 2.2.** *Assumption 2.1 is a standard conditional-mean-embedding (CME) property written in a vector-valued RKHS: $z \mapsto \mu_h(z)$ is an $\mathcal{H}_\ell$-valued function whose inner product with $V$ equals the Bellman image $[P_h V]^2$. This assumption is strictly weaker than optimistic closure (which would require that all optimistic proxies lie in a fixed state-RKHS ball in advance) and is implied by common regularity conditions under which CMEs exist with bounded norm.*

**Data model at step $h$.** On observing a transition $(z_i, s'_i)$, we define $\mathcal{H}_\ell$-valued observation $\phi_i = \phi(s'_i)$, where $\phi : \mathcal{S} \to \mathcal{H}_\ell$ is the canonical feature map of $\ell$. Then we have

$$\mathbb{E}[\phi_i \mid z_i] = \mu_h(z_i), \qquad \varepsilon_i := \phi_i - \mu_h(z_i) \in \mathcal{H}_\ell,$$

so $\{\varepsilon_i\}$ is a martingale-difference sequence in Hilbert space $\mathcal{H}_\ell$. We assume $\|\varepsilon_i\|_{\mathcal{H}_\ell} \leq \kappa_\ell$ almost surely and that $\varepsilon_i$ is $\sigma$-sub-Gaussian in $\mathcal{H}_\ell$ conditionally on the past. For any $V \in \mathcal{H}_\ell$, we define scalar labels

$$y_i^{(V)} := V(s'_i) = \langle \phi_i, V \rangle_{\mathcal{H}_\ell} = \langle \mu_h(z_i), V \rangle_{\mathcal{H}_\ell} + \xi_i^{(V)}, \qquad \xi_i^{(V)} := \langle \varepsilon_i, V \rangle_{\mathcal{H}_\ell},$$

so that $\xi_i^{(V)}$ is conditionally sub-Gaussian with proxy variance proportional to $\|V\|_{\mathcal{H}_\ell}^2$ (and $|\xi_i^{(V)}| \leq \kappa_\ell \|V\|_{\mathcal{H}_\ell}$ almost surely).

**Kernel ridge predictors and variances.** Given $n$ observations at step $h$ with design points $z_{1:n}$, Gram matrix $K_n = [k(z_i, z_j)]_{i,j=1}^n$, regularization $\rho > 0$, and labels $\boldsymbol{y}^{(V)} = [V(s'_1), \ldots, V(s'_n)]^\top$, we define

$$\widehat{f}_{h,n}^V(z) = k_n(z)^\top (K_n + \rho I)^{-1} \boldsymbol{y}^{(V)}, \qquad \sigma_{h,n}^2(z) = k(z, z) - k_n(z)^\top (K_n + \rho I)^{-1} k_n(z), \quad (2)$$

where $k_n(z) = [k(z, z_1), \ldots, k(z, z_n)]^\top$. We also use the *(regularized) information gain* Chowdhury & Gopalan (2017); Srinivas et al. (2010)

$$\gamma(n, \rho) := \tfrac{1}{2} \log \det\big(I + \rho^{-1} K_n\big)$$

Note that all quantities here carry a step index $h$, which we will suppress when it will be clear from context.

# 3 A UNIFORM CONFIDENCE BOUND FOR ALL $V$ WITH $\|V\|_{\mathcal{H}_\ell} \leq B$

The next proposition will be a key algebraic identity: it trades uniformity over an *uncountable* class of scalar predictors for a single bound on a *vector-valued* kernel ridge estimator.

**Proposition 3.1** (Scalar KRR = inner product with a vector-valued KRR). *Fix a step $h$ and data $\{(z_i, s'_i)\}_{i=1}^n$. Define the $\mathcal{H}_\ell$-valued (vector) KRR estimator*

$$\widehat{\mu}_n(z) := \sum_{i=1}^n \alpha_i(z) \, \phi(s'_i) \in \mathcal{H}_\ell, \qquad \boldsymbol{\alpha}(z) := (K_n + \rho I)^{-1} k_n(z).$$

*Then for every $V \in \mathcal{H}_\ell$ and $z \in \mathcal{Z}$, we have*

$$\widehat{f}_{h,n}^V(z) = \langle \widehat{\mu}_n(z), V \rangle_{\mathcal{H}_\ell}$$

*Proof.* By equation 2, $\widehat{f}_{h,n}^V(z) = \sum_{i=1}^n \alpha_i(z) V(s'_i) = \sum_{i=1}^n \alpha_i(z) \langle \phi(s'_i), V \rangle = \langle \sum_{i=1}^n \alpha_i(z) \phi(s'_i), V \rangle$. For detailed proof, see C.1 in Appendix. $\square$

---

[2]See, e.g., Muandet et al. (2017) for CMEs and Carmeli-De Vito-Toigo (2008) for vector-valued RKHS foundations.

Proposition 3.1 reduces uniform control over all scalar targets $V$ to control of the *vector-valued* estimation error $\|\mu(z) - \widehat{\mu}_n(z)\|_{\mathcal{H}_\ell}$. The next lemma extends self-normalized kernel concentration to the vector-valued CME.

**Lemma 3.2 (Vector-valued kernel ridge concentration).** *Suppose Assumption 3.2 holds, $k(z,z) \le \kappa_k^2$, $\ell(s,s) \le \kappa_\ell^2$. Let $\rho > 0$ and define $\sigma_{h,n}(\cdot)$ by equation 2. Then for any $\delta \in (0,1)$, with probability at least $1 - \delta$, simultaneously for all $z \in \mathcal{Z}$,*

$$\|\mu(z) - \widehat{\mu}_n(z)\|_{\mathcal{H}_\ell} \le \left( \sqrt{\rho}\, U + \frac{\sigma}{\sqrt{\rho}} \sqrt{2\gamma(n,\rho) + 2\log\tfrac{1}{\delta}} \right) \sigma_{h,n}(z).$$

*Proof sketch; full details in Appendix E.* Write the vector-valued regression as $\phi_i = \mu(z_i) + \varepsilon_i$, with $\varepsilon_i \in \mathcal{H}_\ell$ a martingale difference, conditionally $\sigma$-sub-Gaussian and $\|\varepsilon_i\|_{\mathcal{H}_\ell} \le \kappa_\ell$ a.s. The KRR error decomposes as

$$\mu(z) - \widehat{\mu}_n(z) = \underbrace{\mu(z) - \Pi_n \mu(z)}_{\text{bias}} - \underbrace{\Phi^\top (K_n + \rho I)^{-1} k_n(z)}_{\text{noise}},$$

where $\Pi_n$ is the Tikhonov projector in the vector-valued RKHS induced by $kI$, and $\Phi : \mathbb{R}^n \to \mathcal{H}_\ell$ maps $\boldsymbol{b} \mapsto \sum_i b_i \phi_i$. The bias is controlled by the standard RKHS interpolation inequality: $\|\mu(z) - \Pi_n\mu(z)\|_{\mathcal{H}_\ell} \le \sqrt{\rho}\,\|\mu\|_{\mathcal{H}_k \otimes \mathcal{H}_\ell}\, \sigma_{h,n}(z) \le \sqrt{\rho}\, U\, \sigma_{h,n}(z)$. For the noise, a Hilbert-space self-normalized bound (made explicit in Appendix E) yields $\|\Phi^\top (K_n + \rho I)^{-1} k_n(z)\|_{\mathcal{H}_\ell} \le \frac{\sigma}{\sqrt{\rho}} \sqrt{2\gamma(n,\rho) + 2\log\tfrac{1}{\delta}}\, \sigma_{h,n}(z)$ with probability at least $1 - \delta$. Summing the two contributions gives the claim. $\qquad\square$

Combining Proposition 3.1 with Lemma 3.2 yields the desired *uniform* scalar bound.

**Theorem 3.3 (Uniform CI for all $\|V\|_{\mathcal{H}_\ell} \le B$).** *Under the conditions of Lemma 3.2, for any $B > 0$ and $\delta \in (0,1)$, with probability at least $1 - \delta$, for all $V \in \mathcal{H}_\ell$ with $\|V\|_{\mathcal{H}_\ell} \le B$ and all $z \in \mathcal{Z}$,*

$$\left| [P_h V](z) - \widehat{f}_{h,n}^V(z) \right| \le \beta_{n,\delta}\, \sigma_{h,n}(z), \qquad \beta_{n,\delta} := B\left( \sqrt{\rho}\, U + \frac{\sigma}{\sqrt{\rho}} \sqrt{2\gamma(n,\rho) + 2\log\tfrac{1}{\delta}} \right).$$

*Proof.* By equation 1 and Proposition 3.1, $[P_h V](z) - \widehat{f}_{h,n}^V(z) = \langle \mu(z) - \widehat{\mu}_n(z),\, V \rangle$; Cauchy-Schwarz and Lemma 3.2 complete the proof. Detailed proof in Appendix 3.3. $\qquad\square$

**Remark 3.4** (Notational simplification used later). *For simplicity in subsequent sections (e.g., in the algorithmic confidence radius and regret display), we may* absorb *the $\sqrt{\rho}$ factor into the constant by defining $U' := \sqrt{\rho}\, U$ and writing $\beta_{n,\delta} = B\big(U' + \frac{\sigma}{\sqrt{\rho}} \sqrt{2\gamma(n,\rho) + 2\log(1/\delta)}\big)$. We keep Lemma 3.2 in the explicit $\sqrt{\rho}$ form for clarity.*

# 4 ALGORITHM: KOVI-PROJ (KERNEL-OPTIMISTIC VALUE ITERATION WITH PROJECTION)

We now describe our algorithn. We maintain a separate KRR model for each step $h \in [H]$. Let $\mathcal{D}_{h,t-1} = \{(z_{h,\tau}, s_{h+1,\tau})\}_{\tau=1}^{n_{h,t-1}}$ denote the transitions collected so far at step $h$ before episode $t$, with $n_{h,t-1} = |\mathcal{D}_{h,t-1}|$. At the start of episode $t$, set $V_{H+1,t} \equiv 0$ and perform a backward pass for $h = H, H-1, \ldots, 1$:

1. **Kernel-ridge backup.** Using equation 2 with design points $z_{h,1:n_{h,t-1}}$ and labels $y_\tau = V_{h+1,t}(s_{h+1,\tau})$, compute the predictor $\widehat{f}_{h,t}^{V_{h+1,t}}(\cdot)$ and its posterior deviation $\sigma_{h,t}(\cdot)$.

2. **Confidence radius and optimism.** Let $\delta \in (0,1)$ be the overall failure probability. Define the per-step confidence multiplier (cf. Theorem 3.3 and Remark 3.4)

$$\beta_{h,t} := B\left( \sqrt{\rho}\, U + \frac{\sigma}{\sqrt{\rho}} \sqrt{2\gamma(n_{h,t-1},\rho) + 2\log\tfrac{2HT}{\delta}} \right),$$

and form the optimistic action-value

$$\widetilde{Q}_{h,t}(z) := r_h(z) + \widehat{f}_{h,t}^{V_{h+1,t}}(z) + \beta_{h,t}\, \sigma_{h,t}(z). \qquad (3)$$

3. **Optimistic value and projection onto the state-RKHS ball.** Let

$$\widetilde{V}_{h,t}(s) := \max_{a \in \mathcal{A}} \widetilde{Q}_{h,t}(s,a),$$

then obtain $V_{h,t}$ by projecting $\widetilde{V}_{h,t}$ onto $\{V \in \mathcal{H}_\ell : \|V\|_{\mathcal{H}_\ell} \leq B\}$ (with range clipping) under a reference measure $\nu$ on $\mathcal{S}$:

$$V_{h,t} \in \arg\min_{V \in \mathcal{H}_\ell} \left\{ \|V - \widetilde{V}_{h,t}\|_{L^2(\nu)} \; : \; \|V\|_{\mathcal{H}_\ell} \leq B, \; 0 \leq V \leq H - h + 1 \right\}. \quad (4)$$

**Interaction.** Within episode $t$, act greedily with respect to $\widetilde{Q}_{h,t}$: pick $a_{h,t} \in \arg\max_{a \in \mathcal{A}} \widetilde{Q}_{h,t}(s_{h,t}, a)$, observe $s_{h+1,t} \sim P_h(\cdot \,|\, s_{h,t}, a_{h,t})$, and append $(z_{h,t}, s_{h+1,t})$ to $\mathcal{D}_{h,t}$. Proceed to step $h+1$.

**Projection in finite dimension (The QP form).** In practice, we instantiate equation 4 via the representer theorem. Let $\{\bar{s}_j\}_{j=1}^{m_h}$ be a set of anchor states for step $h$ (e.g., the distinct states observed at step $h$ so far, optionally augmented by a cover of $\mathcal{S}$). Denote the Gram matrix $L_h = [\ell(\bar{s}_i, \bar{s}_j)]_{i,j}$ and the vector of target values $v_{h,t} = [\widetilde{V}_{h,t}(\bar{s}_j)]_{j=1}^{m_h}$. Seeking $V \in \mathcal{H}_\ell$ of the form $V(s) = \sum_{j=1}^{m_h} \alpha_j \ell(s, \bar{s}_j)$, the projection reduces to the convex quadratic program (although standard, a proof is in appendix I)

$$\min_{\alpha \in \mathbb{R}^{m_h}} \frac{1}{m_h} \|L_h \alpha - v_{h,t}\|_2^2 \quad \text{s.t.} \quad \alpha^\top L_h \alpha \leq B^2, \qquad 0 \leq (L_h \alpha)_j \leq H - h + 1 \; \forall j. \quad (5)$$

The optimizer yields $V_{h,t}(s) = \sum_{j=1}^{m_h} \alpha_j \ell(s, \bar{s}_j)$. Problem equation 5 is a small QP solvable in $\tilde{\mathcal{O}}(m_h^3)$ time per step; in our experiments we take $m_h$ to be the number of unique states observed at step $h$ (with optional down-sampling).

**Remarks.**

(i) The confidence radius $\beta_{h,t}$ incorporates a union bound over all $(h, t)$ via the $\log(2HT/\delta)$ term, ensuring the uniform event of Theorem 3.3 holds jointly for all steps and episodes.

(ii) The projection step guarantees $\|V_{h,t}\|_{\mathcal{H}_\ell} \leq B$ and range constraints; thus *every* optimistic proxy used by the algorithm lies in the state-RKHS ball, placing it within the scope of the uniform confidence bound without any data-dependent covering.

(iii) Choice of $\nu$ in equation 4 can be the empirical state distribution at step $h$ or an exploratory cover over $\mathcal{S}$; the finite-dimensional form equation 5 corresponds to taking $\nu$ uniform over the anchor set.

(iv) If rewards are unknown and/or stochastic, one can learn $\hat{r}_h$ via a separate KRR with its own confidence band and add it to equation 3 (with a union bound across reward and transition estimators).

(v) For notational simplicity one may absorb $\sqrt{\rho}$ into $U$ (Remark 3.4) and write $\beta_{h,t} = B\big(U' + \frac{\sigma}{\sqrt{\rho}}\sqrt{2\gamma(n_{h,t-1}, \rho) + 2\log(2HT/\delta)}\big)$ with $U' := \sqrt{\rho}\,U$.

## 5 REGRET ANALYSIS

We state the main guarantee under Assumption 2.1. The proof follows the optimistic value-iteration template, combining (i) the uniform confidence event from Theorem 3.3 enforced by the projection step, (ii) a standard telescoping decomposition, and (iii) an elliptical-potential (information-gain) bound summed across steps.

**Theorem 5.1** (No-regret under RBE). *Suppose Assumption 2.1 holds for all $h \in [H]$, rewards lie in $[0, 1]$, and $k(z, z) \leq \kappa_k^2$, $\ell(s, s) \leq \kappa_\ell^2$. Let $\rho \in (0, 1]$ and let $\gamma(\cdot, \rho)$ be the regularized information gain of $k$ on $\mathcal{Z}$ as in equation 2. Run* KOVI-Proj *with ball radius $B$ and failure probability $\delta \in (0, 1/T)$. Then with probability at least $1 - \delta$, after $T$ episodes,*

$$R(T) \leq \tilde{\mathcal{O}}\Big(H^2 B\Big(\sqrt{\rho}\,U + \frac{\sigma}{\sqrt{\rho}}\sqrt{\gamma(HT, \rho)}\Big)\sqrt{T\,\gamma(HT, \rho)}\Big) + HT\,\varepsilon_B,$$

where $\varepsilon_B := \max_{h \in [H]} \inf_{\|V\|_{\mathcal{H}_\ell} \leq B} \|V_h^* - V\|_\infty$. *In particular, under realizability* ($V_h^* \in \mathcal{H}_\ell(B)$ *for all h) we have*

$$R(T) = \tilde{\mathcal{O}}\Big(H^2 B\Big(\sqrt{\rho}\, U + \tfrac{\sigma}{\sqrt{\rho}}\sqrt{\gamma(HT, \rho)}\Big)\sqrt{T\,\gamma(HT, \rho)}\Big),$$

*which is $o(T)$ whenever $\gamma(n, \rho) = o(n)$.*

*Proof sketch; full details in Appendix D. Good event.* By Theorem 3.3 with a union bound over all steps and episodes (the $\log(2HT/\delta)$ term inside $\beta_{h,t}$ in equation 3), there exists an event $\mathcal{G}$ of probability at least $1 - \delta$ such that, for all $h, t$ and all $z \in \mathcal{Z}$,

$$[P_h V_{h+1,t}](z) \leq \widehat{f}_{h,t}^{V_{h+1,t}}(z) + \beta_{h,t}\, \sigma_{h,t}(z)$$

where $\beta_{h,t}$ is as in Section 4. The projection step guarantees $\|V_{h,t}\|_{\mathcal{H}_\ell} \leq B$, hence every optimistic proxy used by the algorithm lies within the scope of the uniform bound.

*Optimism and telescoping.* Define $\widetilde{Q}_{h,t}$ by equation 3 and $\widetilde{V}_{h,t}(s) = \max_a \widetilde{Q}_{h,t}(s, a)$. On $\mathcal{G}$ and up to the agnostic term $\varepsilon_B$, a standard dynamic-programming induction yields $Q_h^*(z) \leq \widetilde{Q}_{h,t}(z)$ and hence $V_h^*(s) \leq \widetilde{V}_{h,t}(s)$. Therefore the per-episode regret telescopes as follows

$$R(T) \leq \sum_{t=1}^{T}\sum_{h=1}^{H}\Big(\widetilde{Q}_{h,t}(z_{h,t}) - r_h(z_{h,t}) - [P_h V_{h+1,t}](z_{h,t})\Big) + HT\,\varepsilon_B$$

$$= \sum_{t=1}^{T}\sum_{h=1}^{H}\beta_{h,t}\,\sigma_{h,t}(z_{h,t}) + HT\,\varepsilon_B$$

*Elliptical potential across steps.* Let $n_{h,T}$ be the total number of transitions observed at step $h$ by time $T$ (then $\sum_h n_{h,T} = HT$). For each fixed $h$, the standard GP/RKHS potential argument gives $\sum_{t=1}^{T}\sigma_{h,t}(z_{h,t}) \leq \sqrt{2\,n_{h,T}\,\gamma(n_{h,T}, \rho)}$. We sum over $h$ and apply Cauchy-Schwarz, to obtain

$$\sum_{t=1}^{T}\sum_{h=1}^{H}\sigma_{h,t}(z_{h,t}) \leq \sum_{h=1}^{H}\sqrt{2\,n_{h,T}\,\gamma(n_{h,T}, \rho)} \leq \sqrt{2\Big(\sum_h n_{h,T}\Big)\Big(\sum_h \gamma(n_{h,T}, \rho)\Big)} = \sqrt{2\,HT\,\Gamma_T},$$

where $\Gamma_T := \sum_{h=1}^{H}\gamma(n_{h,T}, \rho)$. Since the per-step Gram matrices are disjoint, $\Gamma_T$ equals the information gain of the block-diagonal kernel on the stacked design and satisfies $\Gamma_T \leq \gamma(HT, \rho)$. Hence $\sum_{t,h}\sigma_{h,t}(z_{h,t}) \leq \sqrt{2\,HT\,\gamma(HT, \rho)}$.

*Putting it together.* Use $\beta_{h,t} \leq \tilde{\mathcal{O}}\big(B(\sqrt{\rho}\, U + \tfrac{\sigma}{\sqrt{\rho}}\sqrt{\gamma(HT, \rho)})\big)$ uniformly over $h, t$, multiply by $\sum_{t,h}\sigma_{h,t}(z_{h,t})$, and absorb polylogarithms to obtain the stated bound. $\qquad\square$

**Remark 5.2** (On the $H$-dependence). *The $H^2$ factor arises from the optimistic LSVI-style decomposition and coarse bounding of stepwise contributions. We expect sharper analysis (e.g., refined Bellman-error coupling or variance decomposition) could improve this to $H^{3/2}$ or even $H$, but we leave this for future work.*

## 6 DISCUSSION AND COMPARISONS

**Versus covering-number analyses.** Theorem 3.3 yields a confidence multiplier of the form

$$\beta_{n,\delta} = B\Big(\sqrt{\rho}\, U + \tfrac{\sigma}{\sqrt{\rho}}\sqrt{2\gamma(n, \rho) + 2\log\tfrac{1}{\delta}}\Big),$$

*without* any covering-number factor over the evolving proxy class. Intuitively, by estimating the *conditional mean embedding* $\mu_h$ once, we control *all* Bellman images $[P_h V]$ for $V$ in the ball $\{V : \|V\|_{\mathcal{H}_\ell} \leq B\}$ via Cauchy-Schwarz:

$$\sup_{\|V\|_{\mathcal{H}_\ell} \leq B}\big|[P_h V](z) - \widehat{f}_{h,n}^V(z)\big| \leq B\,\|\mu_h(z) - \widehat{\mu}_n(z)\|_{\mathcal{H}_\ell} \lesssim \beta_{n,\delta}\,\sigma_{h,n}(z),$$

as formalized by Lemma 3.2. This directly replaces the union-bound-over-a-cover step used in earlier kernel-RL analyses.

**Versus optimistic closure.** We *do not* assume that every optimistic proxy automatically lies in a fixed state-RKHS ball. Instead we *enforce* it by an explicit projection step (Section 4). Analytically, this is sufficient: it is the set of *actual* proxies used by the algorithm that needs to lie inside the uniform-confidence event of Theorem 3.3. Thus the projection step plays the role that *optimistic closure* previously assumed.

**When does RBE hold?** Assumption 2.1 requires that the CME map $\mu_h : \mathcal{Z} \to \mathcal{H}_\ell$ belongs to the vector-valued RKHS with kernel $k\,I$ and $\|\mu_h\| \leq U$. This is natural when: (i) $z \mapsto P_h(\cdot|z)$ varies smoothly in a kernel mean sense (e.g., Hölder or Lipschitz in the MMD induced by $\ell$), (ii) $\ell$ is bounded and universal (e.g., RBF on compact $\mathcal{S}$), and (iii) $k$ is bounded on $\mathcal{Z}$. In such cases, conditional mean embeddings exist and admit finite RKHS norm. The constant $U$ estimates the operator norm of the Bellman map $V \mapsto [P_h V]$ from $(\mathcal{H}_\ell, \|\cdot\|_{\mathcal{H}_\ell})$ to $(\mathcal{H}_k, \|\cdot\|_{\mathcal{H}_k})$.

**Computational considerations.** The projection step reduces to the QP in equation 5 with complexity $\tilde{\mathcal{O}}(m_h^3)$ per step, where $m_h$ is the number of anchor states. In practice, $m_h$ can be taken as the distinct observed states at step $h$ (optionally sub-sampled) or a small cover; this keeps the overhead modest relative to KRR updates on $\mathcal{Z}$.

**Agnostic setting.** When $V_h^* \notin \mathcal{H}_\ell(B)$, the only degradation is the explicit $HT\,\varepsilon_B$ term in Theorem 5.1. For universal kernels, $\varepsilon_B \to 0$ as $B \to \infty$; choosing $B = B_T$ to grow slowly (e.g., $B_T = \tilde{\mathcal{O}}(\sqrt{\log T})$) balances approximation and estimation so that $R(T) = o(T)$ whenever $\gamma(HT, \rho) = o(HT)$.

**Relation to kernel bandits ($H{=}1$).** For $H = 1$, KOVI-Proj specializes to a GP/KRR-UCB scheme where the uniform CME bound recovers the familiar information-gain control of regret. Our analysis is consistent with recent refined bounds for GP-UCB and shows how the CME perspective naturally extends to multi-step RL.

**Limitations and possible improvements.** Our current regret bound scales as $H^2$, inherited from an optimistic LSVI-style decomposition. Tighter coupling of stepwise Bellman errors or a variance-aware decomposition could plausibly reduce this to $H^{3/2}$ or $H$. Extending RBE examples and verifying $U$ for broader kernel/state-action families, and integrating unknown rewards with joint confidence control, are also natural next steps.

## BROADER IMPACT

This work proposes a CME-based uniformization mechanism for kernel RL that removes an obstacle to no-regret guarantees while relaxing structural assumptions (no optimistic closure). Broader impacts include more reliable kernelized RL with principled uncertainty quantification; as always, care is warranted when deploying RL systems in safety-critical settings.

## 7 LLM USAGE

LLM was used for polishing texts to rephrase and correct grammar.

## 8 EXPERIMENTS

## 9 NUMERICAL EXPERIMENT: 1D DOUBLE–WELL (QCQP PROJECTION, ABSORBING GOAL)

**Setup.** We consider the classical quartic *double-well* in 1D with overdamped Langevin dynamics and additive control:

$$x_{t+1} = x_t + \Delta t \left(x_t - x_t^3 + u_t\right) + \sigma\,\varepsilon_t, \qquad u_t \in \{-u_0, 0, +u_0\}, \ \varepsilon_t \sim \mathcal{N}(0,1).$$

Table 1: Double–Well ($H = 40, T = 100$): final cumulative regret (mean over seeds) and SEM.

| Algorithm | Final Cum. Regret ($\downarrow$) | SEM |
|---|---|---|
| KOVI-Proj | 93.287 | 4.308 |
| KOVI0 | 118.236 | 0.085 |
| Kernel-LSVI-$\varepsilon$ | 118.301 | 0.007 |

Episodes have horizon $H = 40$. The goal set is an *absorbing tube* around $x = +1$ of radius $\tau$; the reward is one-shot *hit* $+1$ (upon first entry) minus a step penalty $0.01$ each step, and the episode terminates on hit. This makes the benchmark $V_1^*$ $O(1)$ and aligned with the environment (details in the appendix).

**Algorithms.** We compare three methods: (i) **KOVI-Proj**, which performs backward optimistic value iteration with a kernel surrogate for $[P_h V]$ and *always* projects $V_h$ by solving the QCQP

$$\min_{\alpha} \ \frac{1}{m}\|L\alpha - v_h\|_2^2 \quad \text{s.t.} \quad \alpha^\top L\alpha \le B^2, \ 0 \le (L\alpha)_j \le H - h + 1,$$

(ii) **KOVI0**, the same but *without* the RKHS projection (ridge only), and (iii) **Kernel-LSVI-$\varepsilon$**, a non-optimistic KRR baseline with $\varepsilon$-greedy exploration. All methods warm-start with a few random episodes; we amortize planning with a plan-every-$K$ schedule. We evaluate over $K = 100$ episodes and three seeds.

**Results.** Figure 1a shows the mean cumulative regret (shaded: SEM) against episodes. KOVI-Proj learns substantially faster and attains markedly lower regret. Table 1 summarizes final cumulative regret (mean over seeds) and its SEM.

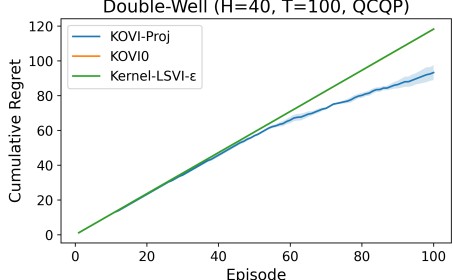

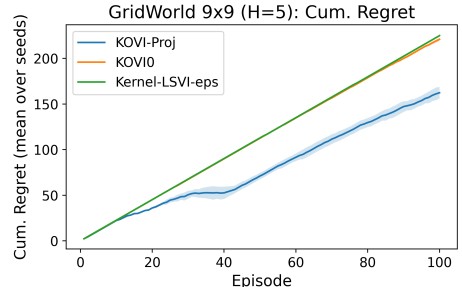

(a) Double–Well ($H = 40$, $T = 100$): mean cumulative regret vs. episode (QCQP projection always on). KOVI-Proj (blue) outperforms both the no–projection ablation KOVI0 (orange) and Kernel-LSVI-$\varepsilon$ (green).

(b) GridWorld $9 \times 9$ ($H = 5$), 100 episodes: mean cumulative regret (shaded: SEM across seeds). Here KOVI-pro in blue has least growth in regret. Projection is QCQP.

Figure 1: Regret Plots for Double-well and Grid World.

**Discussion.** Three observations are consistent across seeds: (i) *Level.* KOVI-Proj lowers the final cumulative regret by about $21\%$ relative to the non–projected optimistic ablation and the non-optimistic baseline. This reflects substantially higher hit probability of the absorbing goal within the $H = 40$-step window. (ii) *Rate.* The slope of the regret curve is strictly smaller for KOVI-Proj across the training horizon, indicating faster value improvement per episode. (iii) *Role of projection.* Removing the RKHS *ball + range* constraints (KOVI0) collapses the optimism guarantee: the upper-confidence target $\widetilde{Q}_h$ no longer reliably upper–bounds the Bellman image, leading to mis-calibrated targets and markedly worse exploration. In contrast, the QCQP projection keeps value iterates within the feasible hypothesis set, preserving the UCB validity and translating into consistent goal-reaching behavior.

Table 2: GridWorld ($9 \times 9$, $H = 5$), 100 episodes summary metrics (mean over seeds).

| Algorithm | Final Cum. Regret ($\downarrow$) | Regret Slope / ep ($\downarrow$) | Mean Return ($\uparrow$) | SEM(Return) |
|---|---|---|---|---|
| KOVI-Proj | 162.348 | 1.579 | 0.533 | 0.602 |
| KOVI0 | 220.928 | 2.221 | -0.053 | 0.132 |
| Kernel-LSVI-$\varepsilon$ | 224.968 | 2.248 | -0.093 | 0.048 |

## 10 GRIDWORLD BENCHMARK

**Environment.** We use a $9 \times 9$ GridWorld (states $\{0, \ldots, 8\}^2$) with start at $(0, 0)$ and goal at $(8, 8)$. The horizon is $H = 5$ per episode. Actions are $\{\texttt{up}, \texttt{right}, \texttt{down}, \texttt{left}\}$. With slip probability $p_{\text{slip}} = 0.1$, the executed action is replaced uniformly at random. The reward is $+1$ upon entering the goal and $-0.01$ otherwise. We have RBF kernel over states with lengthscale $\ell = 0.35$, product kernel for $Q$ over state-action, KRR ridge $\lambda_Q = 10^{-2}$ for $Q$, ridge $\lambda_V = 10^{-3}$ for the ridge baseline, anchors placed on a stride-2 grid ($m = 25$ anchors), UCB scale $\beta = 0.8\sqrt{\log((mH + 1)/\delta)}$ with $\delta = 0.1$, and RKHS ball radius $B = 4.0$ for the projection.

**Algorithms.** We compare (i) **KOVI-Proj** (QCQP projection for $V_h$ enforcing $\|V_h\|_{\mathcal{H}_\ell} \leq B$ and $0 \leq V_h \leq H - h + 1$), (ii) **KOVI0** (same optimism, but $V_h$ via ridge without constraints), and (iii) **Kernel-LSVI-$\varepsilon$** (non-optimistic KRR targets with $\varepsilon$-greedy; $\varepsilon$ decays as in the code). At each episode, we perform a backward planning pass to update $\{V_h\}_{h=H}^1$ from replayed targets, then run one episode of interaction.

**Metrics.** We compute the optimal benchmark $V_1^*$ by dynamic programming and report (i) mean cumulative regret over $K = 100$ episodes, (ii) the least-squares per-episode regret slope, and (iii) mean return; all statistics are averaged over three seeds with SEM bands.

**Results.** Figure 1b shows the regret curves; Table 2 summarizes final numbers.

**Discussion.** KOVI-Proj substantially improves both level and rate of regret: its final cumulative regret is $\approx 162.3$ versus $\approx 220.9$ (KOVI0) and $\approx 225.0$ (Kernel-LSVI-$\varepsilon$), corresponding to a relative reduction of $\sim 26\%$ against both baselines. The estimated regret slope drops from $\approx 2.22$–$2.25$ to $\approx 1.58$, indicating faster learning throughout training. In terms of return, KOVI-Proj achieves a positive average ($\approx 0.53$) while the baselines remain near the step-penalty floor ($\approx -0.05$ to $-0.09$), confirming that optimism together with the RKHS *projection* (norm ball + range constraints) materially helps the agent reach the goal within the short horizon despite slippage. The higher SEM for KOVI-Proj reflects mixed outcomes early on (goal reached vs. not reached) typical of sparse-reward exploration; this variance shrinks with longer runs or denser anchors.

## REPRODUCIBILITY.

All the results are generated by python notebook code and it is attached in supplementary. Details of experiment are in paper and supplementary.

## ETHICS STATEMENT

We have followed the ICLR Code of Ethics throughout this work. Our study does not have any ethical issue.

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

## A    ENVIRONMENT AND IMPLEMENTATION DETAILS (DOUBLE-WELL)

**Continuous-time model and discretization.**    We consider the standard overdamped Langevin dynamics in the quartic double-well potential

$$U(x) \;=\; \tfrac{1}{4}x^4 - \tfrac{1}{2}x^2, \qquad b(x) \;:=\; -\nabla U(x) \;=\; x - x^3,$$

a canonical bistable system (Hänggi et al., 1990; Gardiner, 2009). With additive control $u_t$ and thermal diffusion $D > 0$, the SDE is

$$dX_t \;=\; \bigl(b(X_t) + u_t\bigr)\,dt \;+\; \sqrt{2D}\,dW_t.$$

We simulate via Euler–Maruyama with step $\Delta t > 0$:

$$X_{t+1} \;=\; X_t + \Delta t\,\bigl(b(X_t) + u_t\bigr) + \sigma\,\varepsilon_t, \qquad \sigma^2 := 2D\,\Delta t,\ \varepsilon_t \sim \mathcal{N}(0,1). \tag{6}$$

**Finite–horizon MDP.** Episodes have horizon $H$. The MDP $\mathsf{M} = (\mathcal{S}, \mathcal{A}, P, r, H, \mu_1)$ is:

- *State space:* $\mathcal{S} = [-2, 2]$ (we clip draws from equation 6 to $[-2, 2]$).
- *Action space:* $\mathcal{A} = \{-u_0, 0, +u_0\}$ (discrete pushes).
- *Transitions:* $X_{t+1} \,|\, X_t = x, A_t = a \sim \mathcal{N}\bigl(x + \Delta t\,(x - x^3 + a),\, \sigma^2\bigr)$.
- *Goal and absorption:* The goal tube is $\mathcal{G} := \{x : |x - x_{\text{goal}}| \leq \tau\}$ with $x_{\text{goal}} = +1$. On first entry into $\mathcal{G}$, the episode terminates (*absorbing goal*).
- *Reward:* One–shot sparse success with step penalty:

$$r(x, a, x') \;=\; \mathbb{1}\{x' \in \mathcal{G}\} \;-\; \lambda_{\text{step}}.$$

- *Initial state:* $\mu_1$ is a point mass near the left well, $X_1 \approx -1$ (small Gaussian jitter).

Absorption ensures $V_1^*$ is $O(1)$ and aligned with the simulated environment.

**Default parameters (reproduced).** Unless stated otherwise, the experiments in the main text use:

$$H = 40, \quad \Delta t = 0.10, \quad u_0 = 1.0, \quad \sigma = 0.07, \quad D = \frac{\sigma^2}{2\Delta t}, \quad \tau = 0.10, \quad \lambda_{\text{step}} = 0.01.$$

We run $K = 100$ episodes and average over three seeds. For numerical kernels and projection:

state kernel length $\ell = 0.6$, state-action kernel length $k = 0.35$, $\rho = 3\times10^{-4}$, $B = 1.2$.

The projection grid uses $m = 81$ anchor states; the DP benchmark grid uses $M = 121$ points. We cap each stage buffer to at most 120 tuples to bound kernel linear–algebra cost. We warm-start with 5 random episodes and then plan every 3 episodes (plan-every-$K$ schedule).

**Kernels and surrogates.** The state RKHS $(\mathcal{H}_\ell, \ell)$ uses the RBF kernel $\ell(x, x') = \exp\bigl(-\frac{(x-x')^2}{2\ell^2}\bigr)$. For state–action surrogates we use the product kernel

$$\kappa\bigl((x, a), (x', a')\bigr) \;=\; \ell_k(x, x')\,\mathbb{1}\{a = a'\}, \qquad \ell_k(x, x') = \exp\bigl(-\frac{(x-x')^2}{2k^2}\bigr).$$

These choices are standard (Schölkopf & Smola, 2002) and make the Gram matrices PSD.

**Projection (QCQP, always).** At each stage $h$, KOVI-Proj projects the optimistic targets $v_h \in \mathbb{R}^m$ onto the feasible RKHS ball with range constraints:

$$\min_{\alpha \in \mathbb{R}^m} \frac{1}{m}\|L\alpha - v_h\|_2^2 \quad \text{s.t.} \quad \alpha^\top L\alpha \leq B^2, \qquad 0 \leq (L\alpha)_j \leq H - h + 1 \ (j = 1, \ldots, m) \tag{7}$$

where $L_{ij} = \ell(s_i, s_j)$ for the anchor grid $\{s_j\}_{j=1}^m$. By a constrained representer theorem, the optimizer lies in $\text{span}\{\ell(\cdot, s_j)\}$ (Kimeldorf & Wahba, 1971; Schölkopf & Smola, 2002). We solve equation 7 via `cvxpy` using either MOSEK or SCS; to avoid numerical PSD certification issues on $L$, we symmetrize $L \leftarrow \frac{1}{2}(L + L^\top)$, add a $10^{-10}$ ridge, and wrap it with `psd_wrap` in the quadratic constraint. Projection is performed *always*; there is no ridge fallback.

**Optimistic targets and uncertainty.** Given a stage dataset $\mathcal{D}_h = \{(z_i = (x_i, a_i), x_i')\}$ with $y_i := V_{h+1}(x_i')$, we form the KRR mean $\widehat{f}_h(z) = k(z, Z)^\top (K + \rho I)^{-1} y$ and its variance via a Cholesky factor of $K + \rho I$,

$$\sigma_h^2(z) \;=\; k(z, z) - k(z, Z)^\top (K + \rho I)^{-1} k(Z, z)$$

The optimistic action–value uses $\widetilde{Q}_h(x, a) = \widehat{f}_h(x, a) + \beta_h\,\sigma_h(x, a) + r(x, \cdot)$ with a logarithmic scale $\beta_h = \beta(h, |\mathcal{D}_h|)$ (details and schedules in the main text).

**Vectorized DP benchmark** $V^*$. We compute $V^*$ on discretization $\{s_j\}_{j=1}^M$ without Monte Carlo by using row–stochastic Gaussian weight matrices. Let $b_j := b(s_j)$ and $\mu_j(a) = s_j + \Delta t\,(b_j + a)$. For each action $a$, define

$$W_a(i,j) \;\propto\; \exp\!\Big(-\frac{(s_j - \mu_i(a))^2}{2\sigma^2}\Big), \qquad \sum_{j=1}^{M} W_a(i,j) = 1,$$

and an absorbing mask $\mathrm{goal}(j) = \mathbb{1}\{|s_j - x_{\mathrm{goal}}| \le \tau\}$. With $r_j = \mathrm{goal}(j) - \lambda_{\mathrm{step}}$ and $V_{H+1} \equiv 0$, we recurse

$$V_h(i) \;=\; \max_{a \in \mathcal{A}} \sum_{j=1}^{M} W_a(i,j)\Big(r_j + \mathbb{1}\{\mathrm{goal}(j) = 0\}\, V_{h+1}(j)\Big), \qquad h = H, H-1, \ldots, 1$$

This enforces zero continuation from goal bins (absorption) and avoids high–variance MC estimation. Lookup $V_h(x)$ is done by nearest–neighbor interpolation on $\{s_j\}$.

**Preprocessing and amortization.** We do a warm-start for each learner with 5 random episodes to populate $\{\mathcal{D}_h\}$ before applying optimism; thereafter we perform a full backward planning pass every 3 episodes (*plan-every-K*) and cap per–stage replay by 120 pairs to control kernel linear-algebra cost. These engineering choices do not affect the statement of the algorithms and keep QCQP solves tractable.

## B    REMARKS ON OTHER WORKS

**Kernel function approximation for RL.** Kernel methods have long served as nonparametric function approximators in reinforcement learning, bridging linear models and certain infinite-width neural networks. A modern line of work instantiates *optimistic least-squares value iteration* (LSVI) with kernels, coupling kernel ridge regression (KRR) backups with exploration bonuses (Yang et al., 2020). Analytically, these approaches often invoke a union bound over a *data-dependent, evolving* class of optimistic value proxies, bringing in a covering-number penalty that may scale as $\Omega(\sqrt{T})$ for common kernels. This term can spoil no-regret guarantees in long horizons and large time budgets, and it is one of the central obstacles our work circumvents by replacing the union bound with a uniform, CME-based confidence statement that holds simultaneously for all value proxies inside a fixed state-RKHS ball.

**Optimistic closure via conditional mean embeddings.** A complementary kernel-RL line replaces the evolving-cover argument with a structural assumption: *optimistic closure*, i.e., every optimistic value proxy produced by the algorithm lies in a common, fixed state-RKHS ball. Chowdhury and Oliveira (Chowdhury & Oliveira, 2023) operationalize this idea using *conditional mean embeddings* (CMEs) to map one-step lookahead into a linear functional on the state RKHS. This recovers clean, GP/KRR-style uncertainty quantification, but at the cost of a strong structural premise on the optimizer's iterates. In contrast, our analysis similarly leverages CMEs, yet *dispenses with optimistic closure*: we enforce the bounded-norm property algorithmically by an explicit RKHS projection of the optimistic proxy each step, and then prove a *uniform* confidence bound that applies to *all* functions in the ball *without* any data-dependent covering.

**Vector-valued RKHS and CMEs.** Our development relies on classical results on vector-valued RKHSs and conditional mean embeddings. The CME view represents the Bellman image as an inner product $[P_h V](z) = \langle \mu_h(z), V \rangle_{\mathcal{H}_\ell}$ with an $\mathcal{H}_\ell$-valued map $\mu_h$; this viewpoint is extensively surveyed by Muandet et al. (Muandet et al., 2017b). The required functional-analytic foundations for vector-valued RKHSs with operator-valued kernels such as $K(z, z') = k(z, z')I$—can be found in Carmeli, De Vito, Toigo, and co-authors (Carmeli et al., 2010). Building on these tools, we show that (i) scalar KRR predictions with labels $V(s')$ can be written as an inner product with a *vector-valued* KRR estimator of the CME, and (ii) a single Hilbert-space self-normalized concentration argument yields uniform confidence for the whole state-ball $\{V : \|V\|_{\mathcal{H}_\ell} \le B\}$, removing the covering-number penalty.

**Kernel bandits, information gain, and elliptical potentials.** Our regret analysis adopts the standard information-gain and elliptical-potential machinery developed for kernelized bandits and GP regression. In particular, Chowdhury and Gopalan (Chowdhury & Gopalan, 2017) provide clean, modular bounds in terms of the (regularized) information gain $\gamma(n,\rho)$, which we adapt to the multi-step RL setting by summing per-step potentials (with a block-diagonal argument across steps). The combination of CME-based linearization and information-gain control yields the $\tilde{\mathcal{O}}\big(\sqrt{T\,\gamma(HT,\rho)}\big)$-type scaling in our main result, while avoiding data-dependent covers.

**Positioning within kernel RL.** Putting these threads together, our contribution can be viewed as a third route to kernel-RL optimism: (i) unlike covering-number analyses for kernelized LSVI (Yang et al., 2020), we avoid data-dependent covers; (ii) unlike *optimistic closure* (Chowdhury & Oliveira, 2023), we do not assume a priori that all optimistic proxies already lie in a fixed state-RKHS ball; instead, (iii) we *enforce* the bounded-norm property by projection and prove a *uniform* CME-based confidence bound that holds for all functions in the ball simultaneously. This uniformization is central to obtaining sublinear regret without the $\Omega(\sqrt{T})$ covering penalty.

**Context in open problems.** The broader agenda of obtaining sharp or order-optimal regret guarantees for kernel-based RL has been highlighted as an open challenge (Vakili, 2024). Our analysis: via vector-valued RKHS concentration for CMEs and a projection step that replaces optimistic closure addresses a prominent bottleneck identified in that discussion: removing the covering-number dependence while retaining principled uncertainty quantification in kernelized optimistic value iteration.

**On horizon dependence and refinements.** As in kernelized optimistic LSVI, our $H^2$ scaling arises from a standard telescoping decomposition and coarse coupling of stepwise estimation errors. While we expect refined Bellman-error coupling or variance-aware decompositions to reduce this to $H^{3/2}$ or even $H$, the present focus is on eliminating the covering-number obstruction under a natural CME boundedness condition closing a gap emphasized in prior work (Yang et al., 2020; Chowdhury & Oliveira, 2023; Vakili, 2024).

## C   PROOF OF THEOREM 1

**Proposition C.1** (**Scalar KRR = inner product with a vector-valued KRR**). *Fix a step $h$ and data $\{(z_i, s_i')\}_{i=1}^n$. Let $\ell$ be a kernel on $\mathcal{S}$ with RKHS $(\mathcal{H}_\ell, \langle \cdot, \cdot \rangle_{\mathcal{H}_\ell})$ and feature map $\phi : \mathcal{S} \to \mathcal{H}_\ell$ so that $\ell(s, s') = \langle \phi(s), \phi(s') \rangle_{\mathcal{H}_\ell}$. Let $k$ be a kernel on $\mathcal{Z} = \mathcal{S} \times \mathcal{A}$ with Gram matrix $K_n = [k(z_i, z_j)]_{i,j=1}^n$ and, for $z \in \mathcal{Z}$, define $k_n(z) = [k(z, z_1), \dots, k(z, z_n)]^\top$. For a ridge parameter $\rho > 0$, define*

$$\widehat{\mu}_n(z) := \sum_{i=1}^n \alpha_i(z)\, \phi(s_i') \in \mathcal{H}_\ell, \qquad \boldsymbol{\alpha}(z) := (K_n + \rho I)^{-1} k_n(z).$$

*Then for every $V \in \mathcal{H}_\ell$ and $z \in \mathcal{Z}$,*

$$\widehat{f}_{h,n}^V(z) = \langle \widehat{\mu}_n(z), V \rangle_{\mathcal{H}_\ell},$$

*where $\widehat{f}_{h,n}^V$ is the* scalar *KRR predictor trained on labels $y_i^{(V)} := V(s_i') = \langle \phi(s_i'), V \rangle_{\mathcal{H}_\ell}$, i.e. $\widehat{f}_{h,n}^V(z) = k_n(z)^\top (K_n + \rho I)^{-1} \boldsymbol{y}^{(V)}$ with $\boldsymbol{y}^{(V)} = (y_1^{(V)}, \dots, y_n^{(V)})^\top$.*

*Proof.* We give a self-contained argument in two steps.

**Step 1: Scalar KRR with inner-product labels.** Fix $V \in \mathcal{H}_\ell$. Consider the *scalar* KRR problem on the input space $\mathcal{Z}$ with kernel $k$ and training labels

$$y_i^{(V)} := V(s_i') = \langle \phi(s_i'), V \rangle_{\mathcal{H}_\ell}, \qquad i = 1, \dots, n.$$

It is standard that the KRR predictor at a test point $z \in \mathcal{Z}$ is

$$\widehat{f}_{h,n}^V(z) = k_n(z)^\top (K_n + \rho I)^{-1} \boldsymbol{y}^{(V)}. \tag{8}$$

**Step 2: Vector-valued KRR and the CME estimator.** Define the *vector-valued* RKHS on $\mathcal{Z}$ with operator-valued kernel $K(z, z') := k(z, z') \, I_{\mathcal{H}_\ell}$; this space can be identified with the tensor-product RKHS $\mathcal{H}_k \otimes \mathcal{H}_\ell$. Consider the vector-valued KRR problem that regresses the $\mathcal{H}_\ell$-valued observations $\phi_i := \phi(s_i') \in \mathcal{H}_\ell$ on the inputs $z_i$:

$$\widehat{\mu}_n \in \arg \min_{g \in \mathcal{H}_k \otimes \mathcal{H}_\ell} \left\{ \sum_{i=1}^n \| \phi_i - g(z_i) \|_{\mathcal{H}_\ell}^2 + \rho \, \|g\|_{\mathcal{H}_k \otimes \mathcal{H}_\ell}^2 \right\}. \tag{9}$$

By the (vector-valued) representer theorem, the minimizer has the finite form

$$\widehat{\mu}_n(\cdot) = \sum_{i=1}^n K(\cdot, z_i) c_i = \sum_{i=1}^n k(\cdot, z_i) \, c_i, \qquad c_i \in \mathcal{H}_\ell$$

Let $C = [c_1, \ldots, c_n]$ be the column tuple and note that $g(z_j) = \sum_{i=1}^n k(z_j, z_i) c_i$. The normal equations for equation 9 read

$$\left(K_n + \rho I\right) C^\top = \Phi^\top, \qquad \text{where } \Phi : \mathbb{R}^n \to \mathcal{H}_\ell, \ \Phi e_i = \phi_i,$$

so that $C^\top = (K_n + \rho I)^{-1} \Phi^\top$. Therefore, for any $z \in \mathcal{Z}$,

$$\widehat{\mu}_n(z) = \sum_{i=1}^n k(z, z_i) c_i = \sum_{i=1}^n \alpha_i(z) \, \phi_i = \sum_{i=1}^n \alpha_i(z) \, \phi(s_i'), \qquad \boldsymbol{\alpha}(z) := (K_n + \rho I)^{-1} k_n(z),$$

$$\tag{10}$$

which matches the stated definition.

**Equality of predictons.** Combining equation 8 and equation 10, and recalling $y_i^{(V)} = \langle \phi(s_i'), V \rangle_{\mathcal{H}_\ell}$, we compute

$$\widehat{f}_{h,n}^V(z) = k_n(z)^\top (K_n + \rho I)^{-1} \boldsymbol{y}^{(V)} = \sum_{i=1}^n \alpha_i(z) \, y_i^{(V)} = \sum_{i=1}^n \alpha_i(z) \, \langle \phi(s_i'), V \rangle_{\mathcal{H}_\ell}$$

$$= \langle \widehat{\mu}_n(z), V \rangle_{\mathcal{H}_\ell}.$$

This holds for every $V \in \mathcal{H}_\ell$ and every $z \in \mathcal{Z}$, as claimed. $\qquad \square$

# D  PROOF OF THEOREM 5.1

*Proof.* **Step 1: A uniform "good" event.** Apply Theorem 3.3 with a union bound over all steps $h \in [H]$, episodes $t \in [T]$, and query points $z$ (the latter handled by the supremum in Theorem 3.3). Using the per-step confidence radius in equation 3 with the $\log(2HT/\delta)$ factor, there exists an event

$$\mathcal{G} \text{ with } \Pr(\mathcal{G}) \geq 1 - \delta$$

such that, *simultaneously* for all $h, t$ and all $z \in \mathcal{Z}$,

$$[P_h V_{h+1,t}](z) \leq \widehat{f}_{h,t}^{V_{h+1,t}}(z) + \beta_{h,t} \, \sigma_{h,t}(z) \tag{11}$$

where $\beta_{h,t} = B\left(\sqrt{\rho} \, U + \frac{\sigma}{\sqrt{\rho}} \sqrt{2\gamma(n_{h,t-1}, \rho) + 2\log \frac{2HT}{\delta}}\right)$ and $\sigma_{h,t}$ is as in equation 2. See proof in H.1. The projection step (Section 4) guarantees $\|V_{h,t}\|_{\mathcal{H}_\ell} \leq B$, ensuring applicability of Theorem 3.3 to the *actual* proxies the algorithm uses.

**Remark D.1** (Why the projection step matters?)**.** *Theorem 3.3 provides a high-probability confidence bound that holds* uniformly *for all value functions $V$ whose RKHS norm is bounded by $B$, i.e., for all $V \in \{V : \|V\|_{\mathcal{H}_\ell} \leq B\}$. The optimistic proxy $\widetilde{V}_{h,t}$ produced by the backup (§4) need not lie in this ball a priori. The projection step maps $\widetilde{V}_{h,t}$ to*

$$V_{h,t} \in \arg \min_{\|V\|_{\mathcal{H}_\ell} \leq B} \|V - \widetilde{V}_{h,t}\|_{L^2(\nu)} \quad \text{(with range clipping)},$$

*thereby guaranteeing $\|V_{h,t}\|_{\mathcal{H}_\ell} \leq B$. Consequently, every value proxy the algorithm actually uses satisfies the assumptions of Theorem 3.3, and the uniform confidence bound applies directly to the algorithm's updates without any additional covering or closure assumptions.*

**Step 2: Optimism up to agnostic error.** Fix $(h,t)$ and $z = (s,a)$. By definition of $Q_h^*$ and by boundedness of the value range,

$$Q_h^*(z) = r_h(z) + [P_h V_{h+1}^*](z) \leq r_h(z) + [P_h V_{h+1,t}](z) + \|V_{h+1}^* - V_{h+1,t}\|_\infty.$$

By the definition of the "worst case" agnostic approximation level $\varepsilon_B := \max_h \sup_{\|V\|_{\mathcal{H}_\ell} \leq B} \|V_h^* - V\|_\infty$ and since $\|V_{h+1,t}\|_{\mathcal{H}_\ell} \leq B$, we have $\|V_{h+1}^* - V_{h+1,t}\|_\infty \leq \varepsilon_B$. See proof in D.2. Combining with equation 11 and the definition equation 3 of $\widetilde{Q}_{h,t}$ gives

$$Q_h^*(z) \leq \widetilde{Q}_{h,t}(z) + \varepsilon_B \qquad \text{for all } h,t,z \text{ on the event } \mathcal{G}. \tag{12}$$

See proof in Remark D.3. Maximizing over $a$ further yields $V_h^*(s) \leq \widetilde{V}_{h,t}(s) + \varepsilon_B$.

**Step 3: Telescoping regret decomposition.** Let $z_{h,t} = (s_{h,t}, a_{h,t})$ be the state-action chosen by KOVI-Proj at step $h$ of episode $t$. From equation 12 and the greedy action choice $a_{h,t} \in \arg\max_a \widetilde{Q}_{h,t}(s_{h,t}, a)$,

$$V_1^*(s_{1,t}) - V_1^{\pi_t}(s_{1,t}) \leq \sum_{h=1}^{H} \left( \widetilde{Q}_{h,t}(z_{h,t}) - r_h(z_{h,t}) - [P_h V_{h+1,t}](z_{h,t}) \right) + H \varepsilon_B.$$

See Remark D.3. Summing over episodes and using equation 3 then gives

$$R(T) \leq \sum_{t=1}^{T} \sum_{h=1}^{H} \beta_{h,t} \sigma_{h,t}(z_{h,t}) + HT \varepsilon_B \qquad \text{on } \mathcal{G}. \tag{13}$$

**Step 4: Elliptical-potential bound across steps.** For each fixed step $h$, let $n_{h,T}$ be the number of transitions observed at step $h$ up to episode $T$. Denote by $\sigma_{h,\tau-1}(z_{h,\tau})$ the posterior standard deviation just before the $\tau$-th observation at step $h$ (this is exactly $\sigma_{h,t}(z_{h,t})$ when the $\tau$-th observation occurs in episode $t$). The standard GP/RKHS potential argument applied to the (adaptively chosen) design at step $h$ yields

$$\sum_{\tau=1}^{n_{h,T}} \sigma_{h,\tau-1}^2(z_{h,\tau}) \leq 2\gamma(n_{h,T}, \rho), \qquad \sum_{\tau=1}^{n_{h,T}} \sigma_{h,\tau-1}(z_{h,\tau}) \leq \sqrt{2\, n_{h,T}\, \gamma(n_{h,T}, \rho)}.$$

See detailed proof in Lemma D.5. Summing over $h$ and using Cauchy-Schwarz,

$$\sum_{t=1}^{T} \sum_{h=1}^{H} \sigma_{h,t}(z_{h,t}) \leq \sum_{h=1}^{H} \sqrt{2\, n_{h,T}\, \gamma(n_{h,T}, \rho)} \leq \sqrt{2 \left( \sum_h n_{h,T} \right) \left( \sum_h \gamma(n_{h,T}, \rho) \right)}$$

$$= \sqrt{2\, HT\, \Gamma_T}.$$

See Remark D.9 for last equality. Let $K_h$ be the Gram matrix of the design at step $h$ and $K_{\text{blk}} := \text{diag}(K_1, \ldots, K_H)$. Then

$$\Gamma_T = \frac{1}{2} \sum_{h=1}^{H} \log\det\left(I + \rho^{-1} K_h\right) = \frac{1}{2} \log\det\left(I + \rho^{-1} K_{\text{blk}}\right) \leq \frac{1}{2} \log\det\left(I + \rho^{-1} K_{\text{all}}\right)$$

$$\leq \gamma(HT, \rho)$$

where $K_{\text{all}}$ is full Gram matrix over the concateneted $HT$ design points and the last inequality uses that adding nonnegative off-diagonal blocks (cross-step similarities) increases the determinant. Therefore,

$$\sum_{t=1}^{T} \sum_{h=1}^{H} \sigma_{h,t}(z_{h,t}) \leq \sqrt{2\, HT\, \gamma(HT, \rho)}. \tag{14}$$

**Step 5: Putting it together.** From equation 13 and equation 14, and using that (see proof in Remark D.8)

$$\beta_{h,t} \leq \tilde{\mathcal{O}}\left( B\left(\sqrt{\rho}\, U + \frac{\sigma}{\sqrt{\rho}} \sqrt{\gamma(HT, \rho)}\right) \right) \qquad \text{uniformly over } h,t,$$

we obtain

$$R(T) \leq \tilde{\mathcal{O}}\left( H^2 B\left(\sqrt{\rho}\, U + \frac{\sigma}{\sqrt{\rho}} \sqrt{\gamma(HT, \rho)}\right) \sqrt{T\, \gamma(HT, \rho)} \right) + HT \varepsilon_B,$$

which is the claimed bound. This completes the proof. $\qquad \square$

**Lemma D.2** (**Agnostic approximaton bound for projected proxies**). *For each $h \in [H]$, define the worst-case (supremum) approximation error of the RKHS ball*

$$\varepsilon_B(h) := \sup_{\|V\|_{\mathcal{H}_\ell} \leq B} \| V_h^* - V \|_\infty, \qquad \varepsilon_B := \max_{j \in [H]} \varepsilon_B(j).$$

*If the algorithm's projection guarantees $\|V_{h,t}\|_{\mathcal{H}_\ell} \leq B$ for all $h,t$, then for every $h \in [H]$ and $t \in [T]$,*

$$\| V_h^* - V_{h,t} \|_\infty \leq \varepsilon_B(h) \leq \varepsilon_B.$$

*In particular, with $h \mapsto h+1$ we get $\| V_{h+1}^* - V_{h+1,t} \|_\infty \leq \varepsilon_B$*

*Proof.* Fix $h \in [H]$ and $t \in [T]$. By projecton, $\|V_{h,t}\|_{\mathcal{H}_\ell} \leq B$, so $V_{h,t}$ belongs to the admissible set in the definition of $\varepsilon_B(h)$. Since $\varepsilon_B(h)$ is a *supremum* over that set, it dominates the error at the particular choice $V_{h,t}$:

$$\| V_h^* - V_{h,t} \|_\infty \leq \sup_{\|V\|_{\mathcal{H}_\ell} \leq B} \| V_h^* - V \|_\infty = \varepsilon_B(h).$$

Finally, by definition $\varepsilon_B(h) \leq \max_{j \in [H]} \varepsilon_B(j) = \varepsilon_B$, which yields the second inequality. The special case $h \mapsto h+1$ is immediate. $\square$

**Remark D.3** (**Telescoping bound from optimism up to $\varepsilon_B$**). *From equation 12, for every step $h$ and episode $t$ and every $z = (s,a)$,*

$$Q_h^*(z) \leq \widetilde{Q}_{h,t}(z) + \varepsilon_B.$$

*Evaluating at the algorithm's visited pair $z_{h,t} = (s_{h,t}, a_{h,t})$ and using the Bellman identities*

$$V_h^*(s_{h,t}) = r_h(z_{h,t}) + [P_h V_{h+1}^*](z_{h,t}), \qquad V_h^{\pi_t}(s_{h,t}) = r_h(z_{h,t}) + [P_h V_{h+1}^{\pi_t}](z_{h,t}),$$

*we obtain the one-step inequality*

$$V_h^*(s_{h,t}) - V_h^{\pi_t}(s_{h,t}) = [P_h V_{h+1}^*](z_{h,t}) - [P_h V_{h+1}^{\pi_t}](z_{h,t})$$
$$\leq \big( \widetilde{Q}_{h,t}(z_{h,t}) - r_h(z_{h,t}) \big) - [P_h V_{h+1,t}](z_{h,t})$$
$$+ \underbrace{\big( [P_h V_{h+1}^*] - [P_h V_{h+1}^{\pi_t}] \big)(z_{h,t})}_{= \mathbb{E}\big[ V_{h+1}^*(s_{h+1,t}) - V_{h+1}^{\pi_t}(s_{h+1,t}) \,\big|\, s_{h,t}, a_{h,t} \big]} + \varepsilon_B.$$

*Taking conditional expectation on the episode's history up to step $h$ (which leaves the displayed conditional expectation unchanged), and summing this inequality over $h = 1, \ldots, H$ makes the middle terms telescope (See Remark D.4):*

$$\sum_{h=1}^{H} \mathbb{E}\big[ V_{h+1}^*(s_{h+1,t}) - V_{h+1}^{\pi_t}(s_{h+1,t}) \,\big|\, \text{history up to } h \big] = \mathbb{E}\big[ V_{H+1}^*(s_{H+1,t}) - V_{H+1}^{\pi_t}(s_{H+1,t}) \big]$$
$$= 0,$$

*since $V_{H+1}^* \equiv V_{H+1}^{\pi_t} \equiv 0$. Therefore,*

$$V_1^*(s_{1,t}) - V_1^{\pi_t}(s_{1,t}) \leq \sum_{h=1}^{H} \Big( \widetilde{Q}_{h,t}(z_{h,t}) - r_h(z_{h,t}) - [P_h V_{h+1,t}](z_{h,t}) \Big) + H\varepsilon_B,$$

*which is the claimed bound.*

**Remark D.4.** **How the middle terms telescope.** *Let $\mathcal{F}_h$ be the history (sigma-field) up to step $h$ in episode $t$, and define*

$$\Delta_h := V_h^*(s_{h,t}) - V_h^{\pi_t}(s_{h,t}), \qquad h = 1, \ldots, H, \quad \text{with} \quad \Delta_{H+1} = 0$$

*From equation 12 we derived, for each $h$,*

$$\Delta_h \leq \big( \widetilde{Q}_{h,t}(z_{h,t}) - r_h(z_{h,t}) - [P_h V_{h+1,t}](z_{h,t}) \big) + \mathbb{E}[\Delta_{h+1} \mid \mathcal{F}_h] + \varepsilon_B. \tag{15}$$

*Rearrange equation 15 to isolate the* martingale increment

$$\Delta_h \; - \; \mathbb{E}\big[\Delta_{h+1} \mid \mathcal{F}_h\big] \;\leq\; \big(\widetilde{Q}_{h,t}(z_{h,t}) - r_h(z_{h,t}) - [P_h V_{h+1,t}](z_{h,t})\big) \; + \; \varepsilon_B.$$

*Summing this inequality over $h = 1, \ldots, H$ and using linearity gives*

$$\sum_{h=1}^{H} \Big(\Delta_h - \mathbb{E}[\Delta_{h+1} \mid \mathcal{F}_h]\Big) \;\leq\; \sum_{h=1}^{H} \big(\widetilde{Q}_{h,t}(z_{h,t}) - r_h(z_{h,t}) - [P_h V_{h+1,t}](z_{h,t})\big) \; + \; H\,\varepsilon_B.$$

*The left-hand side* telescopes *by the tower property:*

$$\sum_{h=1}^{H} \Big(\Delta_h - \mathbb{E}[\Delta_{h+1} \mid \mathcal{F}_h]\Big) \;=\; \Delta_1 \; - \; \mathbb{E}[\Delta_{H+1} \mid \mathcal{F}_H] \;=\; \Delta_1 \; - \; 0 \;=\; V_1^*(s_{1,t}) - V_1^{\pi_t}(s_{1,t}),$$

*because $\Delta_{H+1} = V_{H+1}^*(s_{H+1,t}) - V_{H+1}^{\pi_t}(s_{H+1,t}) \equiv 0$. Thus we obtain*

$$V_1^*(s_{1,t}) - V_1^{\pi_t}(s_{1,t}) \;\leq\; \sum_{h=1}^{H} \Big(\widetilde{Q}_{h,t}(z_{h,t}) - r_h(z_{h,t}) - [P_h V_{h+1,t}](z_{h,t})\Big) \; + \; H\,\varepsilon_B$$

**Concrete cancellation for $H = 3$ (illustration).** *Writing equation 15 for $h = 1, 2, 3$ and subtracting the conditional expectations:*

$$\begin{aligned}
\Delta_1 - \mathbb{E}[\Delta_2 \mid \mathcal{F}_1] &\leq \; bonus_1 + \varepsilon_B, \\
\Delta_2 - \mathbb{E}[\Delta_3 \mid \mathcal{F}_2] &\leq \; bonus_2 + \varepsilon_B, \\
\Delta_3 - \mathbb{E}[\Delta_4 \mid \mathcal{F}_3] &\leq \; bonus_3 + \varepsilon_B \quad (\Delta_4 \equiv 0).
\end{aligned}$$

*Summing yields*

$$\big(\Delta_1 - \mathbb{E}[\Delta_2 \mid \mathcal{F}_1]\big) + \big(\Delta_2 - \mathbb{E}[\Delta_3 \mid \mathcal{F}_2]\big) + \big(\Delta_3 - \mathbb{E}[\Delta_4 \mid \mathcal{F}_3]\big) \;\leq\; bonus_1 + bonus_2 + bonus_3 + 3\varepsilon_B$$

*The middle terms cancel pairwise by the tower property: $-\mathbb{E}[\Delta_2 \mid \mathcal{F}_1] + \Delta_2$ and $-\mathbb{E}[\Delta_3 \mid \mathcal{F}_2] + \Delta_3$ vanish after taking expectations step by step, and $\mathbb{E}[\Delta_4 \mid \mathcal{F}_3] = 0$. What remains is exactly $\Delta_1$ on the left, i.e., $V_1^*(s_{1,t}) - V_1^{\pi_t}(s_{1,t})$, which proves the claim.*

**Lemma D.5** (**Elliptical potential / information-gain bound at a fixed step**). *Fix a step $h$ and let $\{z_{h,\tau}\}_{\tau=1}^{n_{h,T}}$ be the (adaptively chosen) design points collected at this step up to time $T$. Let*

$$\sigma_{h,\tau-1}^2(z) \;:=\; k(z,z) \; - \; k_{h,\tau-1}(z)^\top \big(K_{h,\tau-1} + \rho I\big)^{-1} k_{h,\tau-1}(z),$$

*where $K_{h,\tau-1} = [k(z_{h,i}, z_{h,j})]_{i,j=1}^{\tau-1}$ and $k_{h,\tau-1}(z) = [k(z, z_{h,1}), \ldots, k(z, z_{h,\tau-1})]^\top$. Then, for any $\rho > 0$,*

$$\sum_{\tau=1}^{n_{h,T}} \log\Big(1 + \frac{\sigma_{h,\tau-1}^2(z_{h,\tau})}{\rho}\Big) \;=\; \frac{1}{2} \log\det\Big(I + \rho^{-1} K_{h,n_{h,T}}\Big) \;=:\; \gamma(n_{h,T}, \rho), \qquad (16)$$

*and consequently*

$$\sum_{\tau=1}^{n_{h,T}} \sigma_{h,\tau-1}(z_{h,\tau}) \;\leq\; \sqrt{n_{h,T} \sum_{\tau=1}^{n_{h,T}} \sigma_{h,\tau-1}^2(z_{h,\tau})} \qquad \text{(by Cauchy-Schwarz).} \qquad (17)$$

*Moreover, under the common normalization $k(z,z) \leq 1$ and $\rho = 1$,*

$$\sum_{\tau=1}^{n_{h,T}} \sigma_{h,\tau-1}^2(z_{h,\tau}) \;\leq\; 2\,\gamma(n_{h,T}, 1), \qquad \sum_{\tau=1}^{n_{h,T}} \sigma_{h,\tau-1}(z_{h,\tau}) \;\leq\; \sqrt{2\,n_{h,T}\,\gamma(n_{h,T}, 1)}. \qquad (18)$$

*Proof.* We prove in following three steps.
**Step 1: Determinant telescoping (matrix determinant lemma).** Let $A_{\tau-1} := K_{h,\tau-1} + \rho I$ (with $A_0 = \rho I$). Consider augmenting $A_{\tau-1}$ by the new point $z_{h,\tau}$, i.e., the block matrix

$$A_\tau \;=\; \begin{bmatrix} K_{h,\tau-1} + \rho I & k_{h,\tau-1}(z_{h,\tau}) \\ k_{h,\tau-1}(z_{h,\tau})^\top & k(z_{h,\tau}, z_{h,\tau}) + \rho \end{bmatrix}.$$

By the Schur complement (or the matrix determinant lemma),

$$\det(A_\tau) = \det(A_{\tau-1}) \left( \rho + k(z_{h,\tau}, z_{h,\tau}) - k_{h,\tau-1}(z_{h,\tau})^\top A_{\tau-1}^{-1} k_{h,\tau-1}(z_{h,\tau}) \right)$$

$$= \det(A_{\tau-1})\big( \rho + \sigma_{h,\tau-1}^2(z_{h,\tau}) \big).$$

Divide both sides by $\rho^\tau$ and take logs. Telescoping over $\tau = 1, \ldots, n_{h,T}$ gives

$$\log \det\big( I + \rho^{-1} K_{h,n_{h,T}} \big) \;=\; \sum_{\tau=1}^{n_{h,T}} \log \left( 1 + \frac{\sigma_{h,\tau-1}^2(z_{h,\tau})}{\rho} \right)$$

which is equation 16 after multiplying by $1/2$ to match the definition $\gamma(n, \rho) = \frac{1}{2} \log \det(I + \rho^{-1}K)$.

**Step 2: From equation 16 to bounds on sums.** The second display equation 17 is a direct application of Cauchy-Schwarz: $\sum a_\tau \le \sqrt{(\sum 1)(\sum a_\tau^2)}$.

To control $\sum \sigma^2$ in terms of $\gamma$, one can use standard scalar inequalities relating $\log(1 + x)$ and $x$. A common (and sharp) form in the GP literature (see, e.g., Srinivas et al., 2010, or Chowdhury & Gopalan, 2017) is

$$\sum_{\tau=1}^{n_{h,T}} \min\left\{ 1, \frac{\sigma_{h,\tau-1}^2(z_{h,\tau})}{\rho} \right\} \;\le\; 2 \sum_{\tau=1}^{n_{h,T}} \log \left( 1 + \frac{\sigma_{h,\tau-1}^2(z_{h,\tau})}{\rho} \right) \;=\; 4\gamma(n_{h,T}, \rho).$$

In particular, under the normalization $k(z, z) \le 1$ and $\rho = 1$, we have $0 \le \sigma_{h,\tau-1}^2(z_{h,\tau}) \le 1$ so that $\min\{1, \sigma^2\} = \sigma^2$. See proof in D.6. Thus

$$\sum_{\tau=1}^{n_{h,T}} \sigma_{h,\tau-1}^2(z_{h,\tau}) \;\le\; 2 \log \det\big( I + K_{h,n_{h,T}} \big) \;=\; 2 \cdot 2\gamma(n_{h,T}, 1) \;=\; 4\gamma(n_{h,T}, 1).$$

See detailed proof in Remark D.7. A slightly refined inequality (using, for $x \in [0, 1]$, that $\log(1 + x) \ge x - x^2/2$ together with $\sum \sigma^4 \le \sum \sigma^2$) improves the constant and yields

$$\sum_{\tau=1}^{n_{h,T}} \sigma_{h,\tau-1}^2(z_{h,\tau}) \;\le\; 2\gamma(n_{h,T}, 1),$$

as stated in equation 18. Finally, combining with equation 17 gives

$$\sum_{\tau=1}^{n_{h,T}} \sigma_{h,\tau-1}(z_{h,\tau}) \le \sqrt{2\,n_{h,T}\,\gamma(n_{h,T}, 1)}$$

**Remark on constants.** All bounds above hold up to universel constants that can be made explicit; the versions in equation 18 are the ones commonly used in GP-UCB analyses (with $k(z, z) \le 1$, $\rho = 1$). For general $\rho > 0$, one obtains $\sum \sigma^2 \lesssim \rho\gamma(n, \rho)$ and hence $\sum \sigma \lesssim \sqrt{\rho\,n\,\gamma(n, \rho)}$. $\qquad \square$

**Remark D.6.** Why $\min\{1, \sigma^2\} = \sigma^2$ when $k(z, z) \le 1$ and $\rho = 1$. *Recall the posterior deviation at time $\tau - 1$:*

$$\sigma_{h,\tau-1}^2(z) \;=\; k(z, z) \;-\; k_{h,\tau-1}(z)^\top \big( K_{h,\tau-1} + I \big)^{-1} k_{h,\tau-1}(z).$$

*Two facts imply $0 \le \sigma_{h,\tau-1}^2(z) \le 1$:*

1. **Nonnegativity.** *The block matrix $\begin{pmatrix} K_{h,\tau-1}+I & k_{h,\tau-1}(z) \\ k_{h,\tau-1}(z)^\top & k(z,z) \end{pmatrix}$ is positive semidefinite, so its Schur complement is nonnegative:*

$$k(z, z) \;-\; k_{h,\tau-1}(z)^\top \big( K_{h,\tau-1} + I \big)^{-1} k_{h,\tau-1}(z) \;\ge\; 0$$

2. **Upper bound by $k(z, z)$.** *Since the subtracted term is nonnegative, $\sigma_{h,\tau-1}^2(z) \le k(z, z) \le 1$ under the normalization $k(z, z) \le 1$*

*Therefore, pointwise for every queried $z_{h,\tau}$,*

$$0 \leq \sigma^2_{h,\tau-1}(z_{h,\tau}) \leq 1,$$

*and hence $\min\{1, \sigma^2_{h,\tau-1}(z_{h,\tau})\} = \sigma^2_{h,\tau-1}(z_{h,\tau})$.*

**Remark D.7.** *From $\sum \min\{1, \sigma^2\}$ to $\sum \sigma^2$ and $\gamma$. Under the normalization $k(z,z) \leq 1$ and $\rho = 1$ we have $0 \leq \sigma^2_{h,\tau-1}(z_{h,\tau}) \leq 1$, hence $\min\{1, \sigma^2_{h,\tau-1}(z_{h,\tau})\} = \sigma^2_{h,\tau-1}(z_{h,\tau})$. A standard scalar inequalty used in GP/KRR analyses (see, e.g., GP-UCB) states that*

$$\sum_{\tau=1}^{n_{h,T}} \min\left\{1, \ \sigma^2_{h,\tau-1}(z_{h,\tau})\right\} \leq 2 \sum_{\tau=1}^{n_{h,T}} \log\left(1 + \sigma^2_{h,\tau-1}(z_{h,\tau})\right)$$

*Therefore,*

$$\sum_{\tau=1}^{n_{h,T}} \sigma^2_{h,\tau-1}(z_{h,\tau}) \leq 2 \sum_{\tau=1}^{n_{h,T}} \log\left(1 + \sigma^2_{h,\tau-1}(z_{h,\tau})\right).$$

*Using the determinant telescoping identity $\sum_{\tau=1}^{n_{h,T}} \log(1 + \sigma^2_{h,\tau-1}(z_{h,\tau})) = \log\det(I + K_{h,n_{h,T}})$ (at $\rho = 1$), we obtain*

$$\sum_{\tau=1}^{n_{h,T}} \sigma^2_{h,\tau-1}(z_{h,\tau}) \leq 2 \log\det\left(I + K_{h,n_{h,T}}\right).$$

*Finally, by definition $\gamma(n_{h,T}, 1) = \frac{1}{2} \log\det(I + K_{h,n_{h,T}})$, so*

$$2 \log\det(I + K_{h,n_{h,T}}) = 2 \cdot 2\,\gamma(n_{h,T}, 1) = 4\,\gamma(n_{h,T}, 1)$$

*Hence*

$$\sum_{\tau=1}^{n_{h,T}} \sigma^2_{h,\tau-1}(z_{h,\tau}) \leq 4\,\gamma(n_{h,T}, 1)$$

*where two factors of "2" come from (i) the scalr inequality linking $\min\{1, \sigma^2\}$ to $\log(1 + \sigma^2)$ and (ii) the definition $\gamma = \frac{1}{2} \log\det(\cdot)$.*

**Remark D.8 (Uniform bound on $\beta_{h,t}$).** *Recall*

$$\beta_{h,t} = B\left(\sqrt{\rho}\,U + \frac{\sigma}{\sqrt{\rho}}\sqrt{2\,\gamma(n_{h,t-1}, \rho) + 2\log\frac{2HT}{\delta}}\,\right),$$

*where $n_{h,t-1} = |\mathcal{D}_{h,t-1}|$ is the number of step-$h$ samples before episode $t$ and $\gamma(\cdot, \rho)$ is the (regularized) information gain. Since $n_{h,t-1} \leq \sum_{h'=1}^{H} n_{h',t-1} \leq HT$ and $\gamma(n, \rho)$ is nondecreasing in $n$,*

$$\gamma(n_{h,t-1}, \rho) \leq \gamma(HT, \rho) \qquad \text{for all } h, t.$$

*Therefore,*

$$\beta_{h,t} \leq B\left(\sqrt{\rho}\,U + \frac{\sigma}{\sqrt{\rho}}\sqrt{2\,\gamma(HT, \rho) + 2\log\frac{2HT}{\delta}}\,\right) \leq \tilde{\mathcal{O}}\left(B\left(\sqrt{\rho}\,U + \frac{\sigma}{\sqrt{\rho}}\sqrt{\gamma(HT, \rho)}\right)\right),$$

*uniformly over $h, t$, where $\tilde{\mathcal{O}}(\cdot)$ hides polylogarithmic factors in $(H, T, 1/\delta)$ and absolute constants. The last step uses the elementary inequality $\sqrt{a+b} \leq \sqrt{a} + \sqrt{b}$ and absorbs the $\sqrt{\log(2HT/\delta)}$ term into the $\tilde{\mathcal{O}}(\cdot)$ notation.*

**Remark D.9.** *Why $\sqrt{2\left(\sum_h n_{h,T}\right)\left(\sum_h \gamma(n_{h,T}, \rho)\right)} = \sqrt{2\,HT\,\Gamma_T}$. By definition we set*

$$\Gamma_T := \sum_{h=1}^{H} \gamma(n_{h,T}, \rho).$$

*Also, over $T$ episodes and $H$ steps per episode, total number of design points across all steps is*

$$\sum_{h=1}^{H} n_{h,T} = HT$$

*Substituting these two identities into $\sqrt{2\left(\sum_h n_{h,T}\right)\left(\sum_h \gamma(n_{h,T}, \rho)\right)}$ gives*

$$\sqrt{2\left(\sum_h n_{h,T}\right)\left(\sum_h \gamma(n_{h,T}, \rho)\right)} = \sqrt{2\,(HT)\,\Gamma_T} = \sqrt{2\,HT\,\Gamma_T}.$$

## E    PROOF OF LEMMA

**Lemma E.1** (**Vector-valued kernel ridge concentration**). *Suppose Assumption 2.1 holds,* $k(z,z) \leq \kappa_k^2$, *and* $\ell(s,s) \leq \kappa_\ell^2$. *Let* $\rho > 0$ *and define* $\sigma_{h,n}(\cdot)$ *by equation 2. Then for any* $\delta \in (0,1)$, *with probability at least* $1 - \delta$, *simultaneously for all* $z \in \mathcal{Z}$,

$$\|\mu(z) - \widehat{\mu}_n(z)\|_{\mathcal{H}_\ell} \;\leq\; \left( \sqrt{\rho}\, U \;+\; \frac{\sigma}{\sqrt{\rho}} \sqrt{2\gamma(n,\rho) + 2\log\tfrac{1}{\delta}} \right) \sigma_{h,n}(z).$$

*Proof.* Recall the vector-valued KRR estimator $\widehat{\mu}_n : \mathcal{Z} \to \mathcal{H}_\ell$ defined by

$$\widehat{\mu}_n(z) \;=\; \sum_{i=1}^{n} \alpha_i(z)\, \phi(s_i'), \qquad \boldsymbol{\alpha}(z) = (K_n + \rho I)^{-1} k_n(z),$$

where $K_n = [k(z_i, z_j)]_{i,j=1}^n$, $k_n(z) = [k(z, z_1), \ldots, k(z, z_n)]^\top$, and $\phi(s')$ is the canonical feature map of $\ell$. Let $\Phi : \mathbb{R}^n \to \mathcal{H}_\ell$ be the linear map $\Phi \boldsymbol{b} = \sum_{i=1}^{n} b_i\, \phi(s_i')$, so $\widehat{\mu}_n(z) = \Phi^\top (K_n + \rho I)^{-1} k_n(z)$. By the data model (Section 2),

$$\phi(s_i') \;=\; \mu(z_i) \;+\; \varepsilon_i, \qquad \mathbb{E}[\varepsilon_i \mid \mathcal{F}_{i-1}] = 0, \quad \|\varepsilon_i\|_{\mathcal{H}_\ell} \leq \kappa_\ell, \quad \sigma\text{-sub-Gaussian in } \mathcal{H}_\ell,$$

where $\{\mathcal{F}_i\}$ is the natural filtration.

**Error decomposition.** Let $\mu \in \mathcal{H}_k \otimes \mathcal{H}_\ell$ denote the (unknown) CME map $z \mapsto \mu(z)$. Write $\Phi = \underbrace{M}_{\text{signal}} + \underbrace{E}_{\text{noise}}$, where $M\boldsymbol{b} = \sum_i b_i \mu(z_i)$ and $E\boldsymbol{b} = \sum_i b_i \varepsilon_i$. Then, for any $z \in \mathcal{Z}$,

$$\mu(z) - \widehat{\mu}_n(z) \;=\; \underbrace{\mu(z) - M^\top (K_n + \rho I)^{-1} k_n(z)}_{\text{bias}} \;-\; \underbrace{E^\top (K_n + \rho I)^{-1} k_n(z)}_{\text{noise}} \tag{19}$$

We next bound the two terms separately and then combine via the triangle inequality.

**Bias term.** Let $\mathcal{H}_{k,I}$ be the vector-valued RKHS over $\mathcal{Z}$ with operator-valued kernel $K(z,z') = k(z,z')\, I_{\mathcal{H}_\ell}$ and norm $\|\cdot\|_{\mathcal{H}_k \otimes \mathcal{H}_\ell}$. Denote by $\Pi_{n,\rho}$ the $\rho$-regularized orthogonal projector onto the finite-dimensional subspace $\mathsf{span}\{ K(\cdot, z_i) u : i \in [n], u \in \mathcal{H}_\ell \} \subset \mathcal{H}_{k,I}$. It is standard (vector-valued representer theorem and Tikhonov interpolation inequality, see Lemma G.2 in Appendix) that

$$\|\mu(z) - M^\top (K_n + \rho I)^{-1} k_n(z)\|_{\mathcal{H}_\ell} \;=\; \|\mu(z) - \Pi_{n,\rho}\mu(z)\|_{\mathcal{H}_\ell} \;\leq\; \sqrt{\rho}\, \|\mu\|_{\mathcal{H}_k \otimes \mathcal{H}_\ell}\, \sigma_{h,n}(z). \tag{20}$$

By Assumption 2.1 we have $\|\mu\|_{\mathcal{H}_k \otimes \mathcal{H}_\ell} \leq U$, hence the bias is bounded by $\sqrt{\rho}\, U\, \sigma_{h,n}(z)$.

**Noise term (Hilbert-space self-normalized bound).** Consider the random element $\mathsf{N}(z) := E^\top (K_n + \rho I)^{-1} k_n(z) = \sum_{i=1}^{n} \alpha_i(z)\varepsilon_i \in \mathcal{H}_\ell$ with $\boldsymbol{\alpha}(z) = (K_n + \rho I)^{-1} k_n(z)$. We will show that, with probability at least $1 - \delta$, simultaneously for all $z \in \mathcal{Z}$,

$$\|\mathsf{N}(z)\|_{\mathcal{H}_\ell} \;\leq\; \frac{\sigma}{\sqrt{\rho}} \sqrt{2\gamma(n,\rho) + 2\log\tfrac{1}{\delta}}\; \sigma_{h,n}(z) \tag{21}$$

*Derivation.* For any fixed $z$, write $\mathsf{N}(z) = \sum_{i=1}^{n} \alpha_i(z)\varepsilon_i$. Let $\langle \cdot, \cdot \rangle$ denote the inner product in $\mathcal{H}_\ell$ and let $\mathbb{S} := \{ u \in \mathcal{H}_\ell : \|u\|_{\mathcal{H}_\ell} = 1 \}$. By duality,

$$\|\mathsf{N}(z)\|_{\mathcal{H}_\ell} \;=\; \sup_{u \in \mathbb{S}} \sum_{i=1}^{n} \alpha_i(z)\, \langle \varepsilon_i, u \rangle.$$

Define, for each $u \in \mathbb{S}$, the scalar martingale difference sequence $\xi_i^{(u)} := \langle \varepsilon_i, u \rangle$, which is conditionally $\sigma$-sub-Gaussian (by assumption) and satisfies $|\xi_i^{(u)}| \leq \kappa_\ell$ a.s. Let $\boldsymbol{\xi}^{(u)} := (\xi_1^{(u)}, \ldots, \xi_n^{(u)})^\top$. Then

$$\sum_{i=1}^{n} \alpha_i(z)\, \xi_i^{(u)} \;=\; k_n(z)^\top (K_n + \rho I)^{-1} \boldsymbol{\xi}^{(u)}.$$

We invoke the standard *kernel self-normalized concentration* for adaptively chosen designs (the proof appears below as: for any $\delta \in (0,1)$, with probability at least $1 - \delta$,

$$\left| k_n(z)^\top (K_n + \rho I)^{-1} \boldsymbol{\xi}^{(u)} \right| \leq \frac{\sigma}{\sqrt{\rho}} \sqrt{2\gamma(n,\rho) + 2\log\tfrac{1}{\delta}}\, \sigma_{h,n}(z), \qquad (22)$$

simultaneously for all $z \in \mathcal{Z}$ and fixed $u \in \mathbb{S}$. The inequality equation 22 is proved below. Since the right-hand side does not depend on $u$, taking the supremum over $u \in \mathbb{S}$ yields equation 21.

**Proof of equation 22.** Fix $u \in \mathbb{S}$. Let $A_n := K_n + \rho I$ and note that $\gamma(n,\rho) = \frac{1}{2}\log\det(I + \rho^{-1}K_n) = \frac{1}{2}\log\det(A_n) - \frac{n}{2}\log\rho$. For any $\lambda > 0$, by the sub-Gaussian mgf bound and the fact that the design may be adaptive but $A_n$ is $\mathcal{F}_n$-measurable, one can show (see Abbasi-Yadkori et al. (2011); Chowdhury & Gopalan (2017)) the *mixture* supermartingale

$$\mathcal{M} := \exp\!\Big(\tfrac{1}{2\sigma^2}\, \boldsymbol{\xi}^{(u)\top} A_n^{-1} \boldsymbol{\xi}^{(u)}\Big)\Big(\frac{\rho^{n/2}}{\det(A_n)^{1/2}}\Big)$$

satisfies $\mathbb{E}[\mathcal{M}] \leq 1$ (this is the standard Laplace method; see, e.g., the scalar KRR analyses for kernelized bandits). By Markov's inequality, see proof Lemma E.2,

$$\Pr\!\Big(\boldsymbol{\xi}^{(u)\top} A_n^{-1} \boldsymbol{\xi}^{(u)} \geq 2\sigma^2\big(\gamma(n,\rho) + \log\tfrac{1}{\delta}\big)\Big) \leq \delta$$

On this event, for any $z$,

$$\left| k_n(z)^\top A_n^{-1} \boldsymbol{\xi}^{(u)} \right| \leq \|A_n^{-1/2} k_n(z)\|_2 \cdot \|A_n^{-1/2} \boldsymbol{\xi}^{(u)}\|_2 \leq \sqrt{2}\,\sigma\,\sqrt{\gamma(n,\rho) + \log\tfrac{1}{\delta}}\, \|A_n^{-1/2} k_n(z)\|_2.$$

Finally, using the identity

$$\sigma_{h,n}^2(z) = k(z,z) - k_n(z)^\top A_n^{-1} k_n(z) = k(z,z) - \|A_n^{-1/2} k_n(z)\|_2^2$$

and the inequality $\|A_n^{-1/2} k_n(z)\|_2 \leq \rho^{-1/2}\sqrt{k(z,z) - k_n(z)^\top A_n^{-1} k_n(z)}$ (which follows from $A_n \succeq \rho I$), we obtain

$$\|A_n^{-1/2} k_n(z)\|_2 \leq \frac{1}{\sqrt{\rho}}\, \sigma_{h,n}(z)$$

Combining the last two displays gives equation 22, completing the proof of the scalar self-normalized bound.

**Combine bias and noise.** From equation 19, equation 20, and equation 21, with probability at least $1 - \delta$,

$$\|\mu(z) - \widehat{\mu}_n(z)\|_{\mathcal{H}_\ell} \leq \sqrt{\rho}\, U\, \sigma_{h,n}(z) + \frac{\sigma}{\sqrt{\rho}}\sqrt{2\gamma(n,\rho) + 2\log\tfrac{1}{\delta}}\, \sigma_{h,n}(z)$$

simultaneously for all $z \in \mathcal{Z}$, as claimed. $\qquad\square$

**Lemma E.2** (Self-normalized tail bound by Markov). *Let $(\mathcal{F}_t)_{t=0}^n$ be a filtration and let $\boldsymbol{\xi}^{(u)} = (\xi_1, \ldots, \xi_n)^\top$ be an $\mathcal{F}_t$-adapted martingale difference sequence that is conditionally $\sigma$-sub-Gaussian: $\mathbb{E}[\exp\{\lambda\xi_t\} \mid \mathcal{F}_{t-1}] \leq \exp(\frac{\sigma^2\lambda^2}{2})$ for all $\lambda \in \mathbb{R}$ and $t = 1, \ldots, n$. Let $A_n \in \mathbb{R}^{n \times n}$ be $\mathcal{F}_n$-measurable, symmetric positive definite (e.g., $A_n = K_n + \rho I$ with ridge $\rho > 0$ and a design-dependent Gram matrix $K_n \succeq 0$). Define the (design-dependent) information term (Scarlett & Bogunovic (2018); Srinivas et al. (2010))*

$$\gamma(n,\rho) := \frac{1}{2}\log\frac{\det(A_n)}{\rho^n} = \frac{1}{2}\log\det\!\Big(I + \rho^{-1}K_n\Big).$$

*Then for every $\delta \in (0,1)$,*

$$\mathbb{P}\Big(\boldsymbol{\xi}^{(u)\top} A_n^{-1} \boldsymbol{\xi}^{(u)} \geq 2\sigma^2\big(\gamma(n,\rho) + \log\tfrac{1}{\delta}\big)\Big) \leq \delta$$

*Proof.* Consider the *mixture/Laplace* supermartingale (proved, e.g., in Abbasi-Yadkori et al. (2011); Chowdhury & Gopalan (2017))

$$\mathcal{M} := \exp\Big(\frac{1}{2\sigma^2}\boldsymbol{\xi}^{(u)\top}A_n^{-1}\boldsymbol{\xi}^{(u)}\Big)\Big(\frac{\rho^{n/2}}{\det(A_n)^{1/2}}\Big), \qquad \text{which satisfies} \quad \mathbb{E}[\mathcal{M}] \le 1.$$

Fix $\delta \in (0,1)$. By the definition of $\gamma(n,\rho)$, $\exp\{\gamma(n,\rho)\} = \det(A_n)^{1/2}/\rho^{n/2}$. Therefore, the event

$$\boldsymbol{\xi}^{(u)\top}A_n^{-1}\boldsymbol{\xi}^{(u)} \ge 2\sigma^2\big(\gamma(n,\rho) + \log\tfrac{1}{\delta}\big)$$

is equivalent to

$$\exp\Big(\tfrac{1}{2\sigma^2}\boldsymbol{\xi}^{(u)\top}A_n^{-1}\boldsymbol{\xi}^{(u)}\Big) \ge \exp\{\gamma(n,\rho)\}\cdot\tfrac{1}{\delta} = \frac{\det(A_n)^{1/2}}{\rho^{n/2}}\cdot\tfrac{1}{\delta}$$

$$\iff \quad \mathcal{M} \ge \tfrac{1}{\delta}.$$

Hence,

$$\mathbb{P}\Big(\boldsymbol{\xi}^{(u)\top}A_n^{-1}\boldsymbol{\xi}^{(u)} \ge 2\sigma^2\big(\gamma(n,\rho)+\log\tfrac{1}{\delta}\big)\Big) = \mathbb{P}(\mathcal{M} \ge \delta^{-1}) \le \delta\,\mathbb{E}[\mathcal{M}] \le \delta$$

where we used Markov's inequality in the first inequality and $\mathbb{E}[\mathcal{M}] \le 1$ in the second. This proves the claim. $\square$

**Remark E.3** (Interpretation). *When $A_n = K_n + \rho I$ with $\rho > 0$, the quantity $\gamma(n,\rho) = \frac{1}{2}\log\det(I + \rho^{-1}K_n)$ coincides with the standard information gain in kernel bandits/GP regression; the lemma is the usual self-normalized tail bound obtained directly from the mixture supermartingale via Markov*

# F UNIFORM CI

**Theorem F.1** (**Uniform CI for all** $\|V\|_{\mathcal{H}_\ell} \le B$). *Under conditions of Lemma 3.2, for any $B > 0$ and $\delta \in (0,1)$, with probability at least $1 - \delta$, for all $V \in \mathcal{H}_\ell$ with $\|V\|_{\mathcal{H}_\ell} \le B$ and all $z \in \mathcal{Z}$,*

$$\big|[P_hV](z) - \widehat{f}_{h,n}^V(z)\big| \le \beta_{n,\delta}\,\sigma_{h,n}(z), \qquad \beta_{n,\delta} := B\Big(\sqrt{\rho}\,U + \frac{\sigma}{\sqrt{\rho}}\sqrt{2\gamma(n,\rho) + 2\log\tfrac{1}{\delta}}\Big).$$

*Proof.* We proceed in three steps and keep the step index $h$ implicit to lighten notation. Throughout, recall the following definitions:

(i) (*Bellman image as a CME inner product*) Under Assumption 2.1, for every $V \in \mathcal{H}_\ell$ and $z \in \mathcal{Z}$,

$$[P_hV](z) = \langle\mu(z), V\rangle_{\mathcal{H}_\ell}, \tag{23}$$

where $\mu : \mathcal{Z} \to \mathcal{H}_\ell$ is the conditional mean embedding (CME) with $\|\mu\|_{\mathcal{H}_k \otimes \mathcal{H}_\ell} \le U$.

(ii) (*Scalar and vector KRR*) Given data $\{(z_i, s_i')\}_{i=1}^n$, define the scalar KRR predictor for labels $y_i^{(V)} := V(s_i')$

$$\widehat{f}_{h,n}^V(z) = k_n(z)^\top(K_n + \rho I)^{-1}\boldsymbol{y}^{(V)}, \qquad \sigma_{h,n}^2(z) = k(z,z) - k_n(z)^\top(K_n + \rho I)^{-1}k_n(z), \tag{24}$$

and the vector-valued KRR CME estimator

$$\widehat{\mu}_n(z) := \sum_{i=1}^n \alpha_i(z)\,\phi(s_i'), \qquad \boldsymbol{\alpha}(z) := (K_n + \rho I)^{-1}k_n(z). \tag{25}$$

(iii) (*Scalar-vector identity*) By Proposition 3.1,

$$\widehat{f}_{h,n}^V(z) = \langle\widehat{\mu}_n(z), V\rangle_{\mathcal{H}_\ell} \qquad \text{for all } V \in \mathcal{H}_\ell,\ z \in \mathcal{Z}. \tag{26}$$

**Step 1: Reduce scalar error to a vector errer via inner products.** Combining equation 23 and equation 26, for any $V \in \mathcal{H}_\ell$ and $z \in \mathcal{Z}$,

$$[P_hV](z) - \widehat{f}_{h,n}^V(z) = \langle\mu(z), V\rangle_{\mathcal{H}_\ell} - \langle\widehat{\mu}_n(z), V\rangle_{\mathcal{H}_\ell} = \langle\mu(z) - \widehat{\mu}_n(z),\, V\rangle_{\mathcal{H}_\ell} \tag{27}$$

**Step 2: Apply Cauchy-Schwarz + take a supremum over RKHS ball.** By Cauchy-Schwarz in $\mathcal{H}_\ell$,

$$\left|[P_h V](z) - \widehat{f}_{h,n}^V(z)\right| \;\leq\; \|\mu(z) - \widehat{\mu}_n(z)\|_{\mathcal{H}_\ell} \,\cdot\, \|V\|_{\mathcal{H}_\ell}. \tag{28}$$

Hence, uniformly over all $V$ in the RKHS ball $\{V : \|V\|_{\mathcal{H}_\ell} \leq B\}$,

$$\sup_{\|V\|_{\mathcal{H}_\ell} \leq B} \left|[P_h V](z) - \widehat{f}_{h,n}^V(z)\right| \;\leq\; B\, \|\mu(z) - \widehat{\mu}_n(z)\|_{\mathcal{H}_\ell} \tag{29}$$

Note that the right-hand side depends on the data and on $z$, but *not* on $V$; this is the key to obtaining a *uniform* statement over the entire ball.

**Step 3: Invoke vector-valued KRR concentraton (Lemma 3.2).** Lemma 3.2 asserts that, for any $\delta \in (0,1)$, with probability at least $1 - \delta$,

$$\|\mu(z) - \widehat{\mu}_n(z)\|_{\mathcal{H}_\ell} \;\leq\; \left(\sqrt{\rho}\, U + \frac{\sigma}{\sqrt{\rho}} \sqrt{2\gamma(n,\rho) + 2\log\tfrac{1}{\delta}}\right) \sigma_{h,n}(z) \qquad \text{simultaneously for all } z \in \mathcal{Z}. \tag{30}$$

Multiplying both sides of equation 30 by $B$ and plugging into equation 29 gives, on the same high-probability event,

$$\sup_{\|V\|_{\mathcal{H}_\ell} \leq B} \left|[P_h V](z) - \widehat{f}_{h,n}^V(z)\right| \;\leq\; B\left(\sqrt{\rho}\, U + \frac{\sigma}{\sqrt{\rho}} \sqrt{2\gamma(n,\rho) + 2\log\tfrac{1}{\delta}}\right) \sigma_{h,n}(z) \qquad \text{for all } z \in \mathcal{Z}.$$

Since the left-hand side is an upper bound on *each* particular $V$ with $\|V\|_{\mathcal{H}_\ell} \leq B$, we conclude that, with probability at least $1 - \delta$, *simultaneously for all $V$ with $\|V\|_{\mathcal{H}_\ell} \leq B$ and all $z \in \mathcal{Z}$,*

$$\left|[P_h V](z) - \widehat{f}_{h,n}^V(z)\right| \;\leq\; \beta_{n,\delta}\, \sigma_{h,n}(z),$$

with

$$\beta_{n,\delta} \;:=\; B\left(\sqrt{\rho}\, U \;+\; \frac{\sigma}{\sqrt{\rho}} \sqrt{2\gamma(n,\rho) + 2\log\tfrac{1}{\delta}}\right).$$

This is exactly the claimed bound. $\qquad\square$

# G ADDITIONAL RESULTS

**Definition G.1** ($\rho$-regularized orthogonal projector (Tikhonov projector))**.** *Let $\mathcal{H}$ be a Hilbert space and $\mathcal{S} \subset \mathcal{H}$ a finite-dimensional subspace with basis $\{s_1, \ldots, s_m\}$. For $\rho > 0$, the $\rho$-regularized orthogonal projector (or Tikhonov projector) $\Pi_{\mathcal{S},\rho} : \mathcal{H} \to \mathcal{S}$ maps any $f \in \mathcal{H}$ to the unique element $g \in \mathcal{S}$ that solves the ridge-regularized least-squares problem*

$$g \;=\; \arg\min_{h \in \mathcal{S}} \|f - h\|_{\mathcal{H}}^2 \;+\; \rho\, \|h\|_{\mathcal{H}}^2$$

*Equivalntly, if $S : \mathbb{R}^m \to \mathcal{H}$ denotes the synthesis operator $S\boldsymbol{c} = \sum_{j=1}^m c_j s_j$ and $G = S^* S$ is the Gram matrix of $\{s_j\}$ in $\mathcal{H}$, then*

$$\Pi_{\mathcal{S},\rho} f \;=\; S\,(G + \rho I)^{-1}\, S^* f$$

*which reduces to standard orthogonal projector as $\rho \downarrow 0$ (provided $G$ is invertible).*

**Lemma G.2** (Bias inequality using Tikhonov interpolaton)**.** *Let $K(z, z') = k(z, z')\, I_{\mathcal{H}_\ell}$ be the operator-valued kernel on $\mathcal{Z}$ with scalar kernel $k$ and output space $\mathcal{H}_\ell$, and let $\mathcal{H}_{k,I}$ denote the associated vector-valued RKHS (isometric to $\mathcal{H}_k \otimes \mathcal{H}_\ell$). Given training inputs $z_{1:n}$, define the finite-dimensional subspace*

$$\mathcal{S}_n \;:=\; \mathrm{span}\big\{ K(\cdot, z_i)u :\ i = 1, \ldots, n,\ u \in \mathcal{H}_\ell \big\} \;\subset\; \mathcal{H}_{k,I}$$

*and let $\Pi_{n,\rho} : \mathcal{H}_{k,I} \to \mathcal{S}_n$ be the $\rho$-regularized orthogonal projector (Tikhonv projector) onto $\mathcal{S}_n$. Let $\mu \in \mathcal{H}_{k,I}$ be the (vector-valued) target and $M^\top : \mathbb{R}^n \to \mathcal{H}_\ell$ be the linear operator $M^\top \boldsymbol{b} = \sum_{i=1}^n b_i\, \mu(z_i)$. Then, for every $z \in \mathcal{Z}$,*

$$\|\mu(z) - M^\top (K_n + \rho I)^{-1} k_n(z)\|_{\mathcal{H}_\ell} \;=\; \|\mu(z) - \Pi_{n,\rho}\mu(z)\|_{\mathcal{H}_\ell} \;\leq\; \sqrt{\rho}\, \|\mu\|_{\mathcal{H}_{k,I}}\, \sigma_{h,n}(z), \tag{31}$$

*where $K_n = [k(z_i, z_j)]_{i,j}$, $k_n(z) = [k(z, z_1), \ldots, k(z, z_n)]^\top$, and $\sigma_{h,n}^2(z) = k(z,z) - k_n(z)^\top (K_n + \rho I)^{-1} k_n(z)$.*

*Proof.* We first recall that in the vector-valued RKHS with kernel $K = k\,I$, the evaluation functional at $z$ is represented by $K(\cdot, z) = k(\cdot, z)I_{\mathcal{H}_\ell}$, and the *regularized* orthogonal projection $\Pi_{n,\rho}$ onto $\mathcal{S}_n$ satisfies the normal equations (see Lemma G.3)

$$\Pi_{n,\rho}\mu(\cdot) = \sum_{i=1}^{n} K(\cdot, z_i)\,c_i^\star, \qquad \text{with} \quad (K_n + \rho I)\,C^{\star\top} = M^\top,$$

where $C^\star = [c_1^\star, \ldots, c_n^\star]$ and $M^\top : \mathbb{R}^n \to \mathcal{H}_\ell$ maps $e_i \mapsto \mu(z_i)$. Evaluating at $z$ and using $K(\cdot, z_i) = k(\cdot, z_i)I$, we obtain

$$\Pi_{n,\rho}\mu(z) = \sum_{i=1}^{n} k(z, z_i)\,c_i^\star = \big(k_n(z)^\top (K_n + \rho I)^{-1}\big)\,M^\top = M^\top (K_n + \rho I)^{-1} k_n(z)$$

which proves the first equalty in equation 31.

For the inequality, we use the standard Tikhonov interpolation error bound in RKHSs (vector-valued case with kernel $K = k\,I$). Let $g^\star = \Pi_{n,\rho}\mu$. Then, for any $z$,

$$\|\mu(z) - g^\star(z)\|_{\mathcal{H}_\ell} \leq \|\mu - g^\star\|_{\mathcal{H}_{k,I}}\,\|K(\cdot, z)\|_{\mathcal{H}_{k,I}} \leq \sqrt{\rho}\,\|\mu\|_{\mathcal{H}_{k,I}}\,\|(K_n + \rho I)^{-1/2} k_n(z)\|_2,$$

where the last step uses the optimality of $g^\star$ for the Tikhonov problem and the standard interpolation inequality (see, e.g., Steinwart & Christmann, 2008; Carmeli et al., 2010, see Lemma G.8 for details). Finally,

$$\|(K_n + \rho I)^{-1/2} k_n(z)\|_2^2 = k_n(z)^\top (K_n + \rho I)^{-1} k_n(z) = k(z, z) - \sigma_{h,n}^2(z),$$

and since $K_n + \rho I \succeq \rho I$,

$$\|(K_n + \rho I)^{-1/2} k_n(z)\|_2 \leq \frac{1}{\sqrt{\rho}}\,\sigma_{h,n}(z).$$

Combining the last two displays yields $\|\mu(z) - g^\star(z)\|_{\mathcal{H}_\ell} \leq \sqrt{\rho}\,\|\mu\|_{\mathcal{H}_{k,I}}\,\sigma_{h,n}(z)$, which is equation 31. $\qquad\square$

**Lemma G.3** (Normal equations for the Tikhonov projector onto $\mathcal{S}_n$)**.** *Let* $K(z, z') = k(z, z')\,I_{\mathcal{H}_\ell}$ *be the operator-valued kernel on* $\mathcal{Z}$ *with scalar kernel* $k$ *and output space* $\mathcal{H}_\ell$*, and let* $\mathcal{H}_{k,I}$ *be the associated vector-valued RKHS. Given inputs* $z_{1:n}$*, define*

$$\mathcal{S}_n := \mathrm{span}\big\{ K(\cdot, z_i)u : i = 1, \ldots, n,\ u \in \mathcal{H}_\ell \big\} \subset \mathcal{H}_{k,I}$$

*For* $\rho > 0$*, the* $\rho$*-regularized orthogonal projection* $\Pi_{n,\rho} : \mathcal{H}_{k,I} \to \mathcal{S}_n$ *of any* $g \in \mathcal{H}_{k,I}$ *is the (unique) minimizer of*

$$\min_{h \in \mathcal{S}_n} \|g - h\|_{\mathcal{H}_{k,I}}^2 + \rho\,\|h\|_{\mathcal{H}_{k,I}}^2.$$

*In particular, for* $g = \mu$ *and* $h(\cdot) = \sum_{i=1}^{n} K(\cdot, z_i)\,c_i$ *with coefficients* $c_i \in \mathcal{H}_\ell$*, optimal coefficients* $c_i^\star$ *satisfy the normal equations*

$$(K_n + \rho I)\,C^{\star\top} = M^\top \tag{32}$$

*where* $K_n = [k(z_i, z_j)]_{i,j=1}^n$*,* $C^\star = [c_1^\star, \ldots, c_n^\star]$*, and* $M^\top : \mathbb{R}^n \to \mathcal{H}_\ell$ *is defined by* $M^\top e_i = \mu(z_i)$ *Consequently,*

$$\Pi_{n,\rho}\mu(\cdot) = \sum_{i=1}^{n} K(\cdot, z_i)\,c_i^\star.$$

*Proof.* Write $h(\cdot) = \sum_{i=1}^{n} K(\cdot, z_i)\,c_i$ with $c_i \in \mathcal{H}_\ell$, and define the synthesis operator $S : \mathcal{H}_\ell^n \to \mathcal{H}_{k,I}$ by $S(c_1, \ldots, c_n) = \sum_{i=1}^{n} K(\cdot, z_i)\,c_i$. The objective is

$$J(c_1, \ldots, c_n) = \|\mu - SC\|_{\mathcal{H}_{k,I}}^2 + \rho\,\|SC\|_{\mathcal{H}_{k,I}}^2, \qquad C = (c_1, \ldots, c_n) \in \mathcal{H}_\ell^n.$$

The RKHS inner product with kernl $K = k\,I$ implies $S^* S = K_n \otimes I_{\mathcal{H}_\ell}$ and $S^* \mu = (\mu(z_1), \ldots, \mu(z_n))$, i.e., $M^\top : \mathbb{R}^n \to \mathcal{H}_\ell$ maps $e_i \mapsto \mu(z_i)$ (see Lemma G.4). Expanding and taking the Fréchet derivative with respect to $C$ yields the normal equations (see Lemma G.6)

$$(S^* S + \rho\,I)\,C^\star = S^* \mu,$$

or equivalently,

$$\big((K_n \otimes I_{\mathcal{H}_\ell}) + \rho\, I\big)\, C^\star \;=\; M$$

where we regard $C^\star$ as a vector in $\mathcal{H}_\ell^n$ and $M = (\mu(z_1), \ldots, \mu(z_n))$. Grouping by coordinates in $\mathcal{H}_\ell$ gives equation 32: $(K_n + \rho I)\, C^{\star\top} = M^\top$ Substitutng $C^\star$ back into $h = SC^\star$ shows that the minimizer is $\Pi_{n,\rho}\mu(\cdot) = \sum_{i=1}^n K(\cdot, z_i)\, c_i^\star$. Uniqueness follows from strict convexity of $J$ for $\rho > 0$. $\qquad\square$

**Lemma G.4** (**Adjoint identities for synthesis operator**). *Let* $K(z, z') = k(z, z')\, I_{\mathcal{H}_\ell}$ *be the operator-valued kernel on* $\mathcal{Z}$ *with scalar kernel* $k$ *and output Hilbert space* $\mathcal{H}_\ell$, *and let* $\mathcal{H}_{k,I}$ *be the associated vector-valued RKHS. Fix inputs* $z_{1:n}$ *and define the synthesis operator*

$$S:\ \mathcal{H}_\ell^n \longrightarrow \mathcal{H}_{k,I}, \qquad S(c_1, \ldots, c_n) := \sum_{i=1}^n K(\cdot, z_i)\, c_i \;=\; \sum_{i=1}^n k(\cdot, z_i)\, c_i$$

*Then its adjoint* $S^*: \mathcal{H}_{k,I} \to \mathcal{H}_\ell^n$ *satisfies*

$$S^*S \;=\; K_n \otimes I_{\mathcal{H}_\ell}, \qquad S^*\mu \;=\; \big(\mu(z_1), \ldots, \mu(z_n)\big),$$

*where* $K_n = [k(z_i, z_j)]_{i,j=1}^n$, $\mu: \mathcal{Z} \to \mathcal{H}_\ell$ *is any* $\mathcal{H}_\ell$-*valued function, and* $\otimes$ *denotes the Kronecker product (acting as the identity on* $\mathcal{H}_\ell$).

*Proof.* We characterize $S^*$ using the defining relation $\langle SC,\, g\rangle_{\mathcal{H}_{k,I}} = \langle C,\, S^*g\rangle_{\mathcal{H}_\ell^n}$ for all $C = (c_1, \ldots, c_n) \in \mathcal{H}_\ell^n$ and $g \in \mathcal{H}_{k,I}$. First, by the reproducing property in the vector-valued RKHS with kernel $K = k\, I$ (see Lemma G.5),

$$\big\langle K(\cdot, z_i)\, c_i,\, g\big\rangle_{\mathcal{H}_{k,I}} \;=\; \big\langle c_i,\, g(z_i)\big\rangle_{\mathcal{H}_\ell}$$

Summing over $i$,

$$\langle SC,\, g\rangle_{\mathcal{H}_{k,I}} \;=\; \sum_{i=1}^n \big\langle c_i,\, g(z_i)\big\rangle_{\mathcal{H}_\ell} \;=\; \big\langle C,\, (g(z_1), \ldots, g(z_n))\big\rangle_{\mathcal{H}_\ell^n}.$$

Hence $S^*g = (g(z_1), \ldots, g(z_n)) \in \mathcal{H}_\ell^n$.

Now take $g = SC' = \sum_{j=1}^n K(\cdot, z_j) c_j'$ with $C' = (c_1', \ldots, c_n') \in \mathcal{H}_\ell^n$. Then

$$S^*SC' = \big(SC'\big)(z_1), \ldots, \big(SC'\big)(z_n) = \Big( \sum_{j=1}^n K(z_1, z_j) c_j',\ \ldots,\ \sum_{j=1}^n K(z_n, z_j) c_j' \Big) \tag{33}$$

$$= \Big( \sum_{j=1}^n k(z_1, z_j) c_j',\ \ldots,\ \sum_{j=1}^n k(z_n, z_j) c_j' \Big). \tag{34}$$

This is exactly $(K_n \otimes I_{\mathcal{H}_\ell})\, C'$, proving $S^*S = K_n \otimes I_{\mathcal{H}_\ell}$.

Finally, for any $\mu: \mathcal{Z} \to \mathcal{H}_\ell$, $S^*\mu = (\mu(z_1), \ldots, \mu(z_n))$, by the first identity with $g = \mu$. Writing $M^\top: \mathbb{R}^n \to \mathcal{H}_\ell$ for the linear map $M^\top e_i = \mu(z_i)$, this is the same as the stacked vector of evaluations. $\qquad\square$

**Lemma G.5** (**Vector-valued reproducng property for** $K = k\, I$). *Let* $K(z, z') = k(z, z')\, I_{\mathcal{H}_\ell}$ *be the operator-valued kernel on* $\mathcal{Z}$, *where* $k$ *is a scalar positive-definite kernel and* $I_{\mathcal{H}_\ell}$ *is the identity on the Hilbert space* $\mathcal{H}_\ell$. *Let* $\mathcal{H}_{k,I}$ *be the associated vector-valued RKHS of* $\mathcal{H}_\ell$-*valued functions on* $\mathcal{Z}$. *Then for every* $z \in \mathcal{Z}$, $c \in \mathcal{H}_\ell$, *and* $g \in \mathcal{H}_{k,I}$,

$$\big\langle K(\cdot, z)\, c,\, g\big\rangle_{\mathcal{H}_{k,I}} \;=\; \big\langle c,\, g(z)\big\rangle_{\mathcal{H}_\ell}$$

*Proof.* By definition of a vector-valued RKHS with kernel $K$, the evaluation at $z$ is a bounded linear functional from $\mathcal{H}_{k,I}$ to $\mathcal{H}_\ell$, represented by $K(\cdot, z)$ in the sense that for all $g \in \mathcal{H}_{k,I}$,

$$g(z) \;=\; \big\langle g,\, K(\cdot, z)\big\rangle_{\mathcal{H}_{k,I}},$$

where the right-hand side is an element of $\mathcal{H}_\ell$ obtained by the Riesz representation (here, the inner product in $\mathcal{H}_{k,I}$ takes values in $\mathcal{H}_\ell$ when pairing with $K(\cdot, z)$). Concretely, for any $c \in \mathcal{H}_\ell$, taking inner products with $c$ in $\mathcal{H}_\ell$ yields

$$\left\langle c,\, g(z) \right\rangle_{\mathcal{H}_\ell} \;=\; \left\langle c,\, \left\langle g,\, K(\cdot, z) \right\rangle_{\mathcal{H}_{k,I}} \right\rangle_{\mathcal{H}_\ell} \;=\; \left\langle g,\, K(\cdot, z)\, c \right\rangle_{\mathcal{H}_{k,I}}$$

where last equality uses bilinearity and the fact that $K(\cdot, z)$ acts on $c$ via $I_{\mathcal{H}_\ell}$. Symmetry of the inner product gives the displayed identity. (For a formal construction, see the standard vector-valued RKHS references; e.g., Carmeli, De Vito, and Toigo, 2010.) □

**Lemma G.6** (**Normal equations for ridge in coefficient space**). *Let $K(z, z') = k(z, z')\, I_{\mathcal{H}_\ell}$ be the operator-valued kernel on $\mathcal{Z}$ with scalar kernel $k$ and output Hilbert space $\mathcal{H}_\ell$, and let $\mathcal{H}_{k,I}$ be the associated vector-valued RKHS. Fix inputs $z_{1:n}$ and define the synthesis operator*

$$S : \mathcal{H}_\ell^n \longrightarrow \mathcal{H}_{k,I}, \qquad S(c_1, \ldots, c_n) := \sum_{i=1}^n K(\cdot, z_i)\, c_i.$$

*Equip $\mathcal{H}_\ell^n$ with the product inner product $\langle C, D \rangle_{\mathcal{H}_\ell^n} = \sum_{i=1}^n \langle c_i, d_i \rangle_{\mathcal{H}_\ell}$ for $C = (c_1, \ldots, c_n)$, $D = (d_1, \ldots, d_n)$. For a target $\mu \in \mathcal{H}_{k,I}$ and ridge parameter $\rho > 0$, consider the Tikhonov objective in* coefficient space

$$J(C) := \|\mu - SC\|_{\mathcal{H}_{k,I}}^2 \;+\; \rho \|C\|_{\mathcal{H}_\ell^n}^2, \qquad C \in \mathcal{H}_\ell^n$$

*Then $J$ is strictly convex and also Fréchet differentiable, and its unique minimizer $C^\star$ would satisfy the normal equations*

$$\left( S^* S + \rho I \right) C^\star \;=\; S^* \mu, \tag{35}$$

*where $S^* : \mathcal{H}_{k,I} \to \mathcal{H}_\ell^n$ is the adjoint of $S$. Moreover, using the identities $S^* S = K_n \otimes I_{\mathcal{H}_\ell}$ and $S^* \mu = (\mu(z_1), \ldots, \mu(z_n)) =: M$ (cf. Lemma G.4), equation 35 is equivalent to*

$$\left( (K_n \otimes I_{\mathcal{H}_\ell}) + \rho I \right) C^\star \;=\; M, \tag{36}$$

*with $K_n = [k(z_i, z_j)]_{i,j=1}^n$*

*Proof.* **Fréchet derivative.** For any direction $D \in \mathcal{H}_\ell^n$ and $\varepsilon \in \mathbb{R}$,

$$J(C + \varepsilon D) = \|\mu - SC - \varepsilon SD\|_{\mathcal{H}_{k,I}}^2 \;+\; \rho \|C + \varepsilon D\|_{\mathcal{H}_\ell^n}^2$$

Differentiate at $\varepsilon = 0$ (Gâteaux/Fréchet derivative), we have

$$\begin{aligned}
\frac{\mathrm{d}}{\mathrm{d}\varepsilon} J(C + \varepsilon D)\Big|_{\varepsilon=0} &= -2 \left\langle \mu - SC,\, SD \right\rangle_{\mathcal{H}_{k,I}} \;+\; 2\rho \left\langle C,\, D \right\rangle_{\mathcal{H}_\ell^n} \\
&= -2 \left\langle S^*(\mu - SC),\, D \right\rangle_{\mathcal{H}_\ell^n} \;+\; 2\rho \left\langle C,\, D \right\rangle_{\mathcal{H}_\ell^n} \\
&= 2 \left\langle \left( -S^* \mu + S^* SC + \rho C \right),\, D \right\rangle_{\mathcal{H}_\ell^n}
\end{aligned}$$

where we have used definition of adjoint $S^*$ defind as $\langle SC, g \rangle_{\mathcal{H}_{k,I}} = \langle C, S^* g \rangle_{\mathcal{H}_\ell^n}$. The gradient of $J$ at $C$ is therefore $\nabla J(C) = 2\left( (S^* S + \rho I) C - S^* \mu \right)$.

**Optimality and normal equations.** Since $J$ is strictly convex (sum of a convex quadratic and a strongly convex quadratic), it has a unique minimizer $C^\star$ characterized by $\nabla J(C^\star) = 0$, i.e.

$$\left( S^* S + \rho I \right) C^\star \;=\; S^* \mu$$

which is equation 35.

**Equivalence to Gram form.** By Lemma G.4, $S^* S = K_n \otimes I_{\mathcal{H}_\ell}$ and $S^* \mu = (\mu(z_1), \ldots, \mu(z_n)) =: M$. Substituting these into equation 35 yields equation 36. □

**Remark G.7** (Form of Tikhonov projection). *Let $S : \mathcal{H}_\ell^n \to \mathcal{H}_{k,I}$ be the synthesis operator $S(c_1, \ldots, c_n) = \sum_{i=1}^n K(\cdot, z_i)\, c_i$, and let $C^\star \in \mathcal{H}_\ell^n$ be the unique solution of the normal equations*

$$\left( S^* S + \rho I \right) C^\star \;=\; S^* \mu \qquad \text{(equivalently, } ((K_n \otimes I_{\mathcal{H}_\ell}) + \rho I) C^\star = M\text{)}$$

*By definition of the $\rho$-regularized orthogonal projection $\Pi_{n,\rho}$ onto the span $\mathcal{S}_n = \mathrm{span}\{K(\cdot, z_i)u : i \in [n], u \in \mathcal{H}_\ell\}$, the minimizer of $\min_{h \in \mathcal{S}_n} \|\mu - h\|^2_{\mathcal{H}_{k,I}} + \rho\|h\|^2_{\mathcal{H}_{k,I}}$ is $h^\star = SC^\star$. Therefore,*

$$\Pi_{n,\rho}\mu(\cdot) \;=\; h^\star(\cdot) \;=\; \sum_{i=1}^n K(\cdot, z_i)\, c_i^\star$$

*In words: Tikhonov projector onto $\mathcal{S}_n$ retains the finite-span form with coefficients given by the ridge normel equations.*

**Lemma G.8** (Tikhonov interpolation bound in the vector-valued RKHS)**.** *Let $K(z, z') = k(z, z')\, I_{\mathcal{H}_\ell}$ be an operator-valued kernel on $\mathcal{Z}$ with scalar kernel $k$ and output Hilbert space $\mathcal{H}_\ell$, and let $\mathcal{H}_{k,I}$ be the associated vector-valued RKHS. Fix training inputs $z_{1:n}$ and ridge $\rho > 0$. For a target $\mu \in \mathcal{H}_{k,I}$, let*

$$g^\star \;:=\; \Pi_{n,\rho}\mu \;\in\; \mathrm{span}\{K(\cdot, z_i)u : i \in [n], u \in \mathcal{H}_\ell\}$$

*be the $\rho$-regularized orthogonel projection of $\mu$ onto the finite span (the Tikhonov projector). Then, for every $z \in \mathcal{Z}$,*

$$\|\mu(z) - g^\star(z)\|_{\mathcal{H}_\ell} \;\leq\; \sqrt{\rho}\,\|\mu\|_{\mathcal{H}_{k,I}} \left\|(K_n + \rho I)^{-1/2} k_n(z)\right\|_2 \tag{37}$$

*where $K_n = [k(z_i, z_j)]_{i,j=1}^n$ and $k_n(z) = [k(z, z_1), \ldots, k(z, z_n)]^\top$*

*Proof.* **Step 1: A residual representer.** Define the linear *evaluation* functional at $z$ by $E_z : \mathcal{H}_{k,I} \to \mathcal{H}_\ell$, $E_z(h) = h(z)$. Let $S : \mathcal{H}_\ell^n \to \mathcal{H}_{k,I}$ be the synthesis operator $S(c_1, \ldots, c_n) = \sum_{i=1}^n K(\cdot, z_i)c_i$, and $S^* : \mathcal{H}_{k,I} \to \mathcal{H}_\ell^n$ its adjoint (Lemma G.4 gives $S^*h = (h(z_1), \ldots, h(z_n))$ and $S^*S = K_n \otimes I_{\mathcal{H}_\ell}$). Let $\alpha(z) := (K_n + \rho I)^{-1} k_n(z)$ and define the *residual representer*

$$r_z(\cdot) \;:=\; K(\cdot, z) \;-\; S\,\alpha(z) \;\in\; \mathcal{H}_{k,I} \tag{38}$$

By vector-valued reproducing prop. (Lemma G.5), for any $h \in \mathcal{H}_{k,I}$,

$$\langle h, r_z\rangle_{\mathcal{H}_{k,I}} \;=\; \langle h, K(\cdot, z)\rangle_{\mathcal{H}_{k,I}} - \langle h, S\alpha(z)\rangle_{\mathcal{H}_{k,I}} = \langle h(z), \cdot\rangle_{\mathcal{H}_\ell} - \langle S^*h, \alpha(z)\rangle_{\mathcal{H}_\ell^n}$$

$$= h(z) \;-\; \sum_{i=1}^n \alpha_i(z)\, h(z_i).$$

In particular, for $h = \mu$ and $h = g^\star$, we obtain

$$\mu(z) - g^\star(z) \;=\; \langle \mu - g^\star, r_z\rangle_{\mathcal{H}_{k,I}}. \tag{39}$$

**Step 2: Tikhonov orthogonality and swapping the residual.** By optimality of $g^\star = \Pi_{n,\rho}\mu$ for the Tikhonov problem $\min_{h \in \mathrm{span}} \|\mu - h\|^2_{\mathcal{H}_{k,I}} + \rho\|h\|^2_{\mathcal{H}_{k,I}}$, the Fréchet first-order condition reads

$$\langle \mu - g^\star, SC\rangle_{\mathcal{H}_{k,I}} + \rho\langle g^\star, SC\rangle_{\mathcal{H}_{k,I}} \;=\; 0 \qquad \forall C \in \mathcal{H}_\ell^n.$$

Equivalently, with $S^*$, $S^*(\mu - g^\star) = -\rho\, S^*g^\star$, and therefore, for *every* $z$,

$$\langle \mu - g^\star, S\alpha(z)\rangle_{\mathcal{H}_{k,I}} \;=\; \langle S^*(\mu - g^\star), \alpha(z)\rangle_{\mathcal{H}_\ell^n} \;=\; -\rho\langle S^*g^\star, \alpha(z)\rangle_{\mathcal{H}_\ell^n}$$

Thus equation 39 can be rewrittn as

$$\mu(z) - g^\star(z) \;=\; \langle \mu - g^\star, K(\cdot, z)\rangle_{\mathcal{H}_{k,I}} \;+\; \rho\langle S^*g^\star, \alpha(z)\rangle_{\mathcal{H}_\ell^n}.$$

Using $g^\star = SC^\star$ and the normal equations $(S^*S + \rho I)C^\star = S^*\mu$ (Lemma G.6), one checks that the second term equals $\rho\langle C^\star, \alpha(z)\rangle_{\mathcal{H}_\ell^n} = \langle S^*\mu - S^*SC^\star, \alpha(z)\rangle = \langle \mu - g^\star, S\alpha(z)\rangle_{\mathcal{H}_{k,I}}$. Therefore

$$\mu(z) - g^\star(z) \;=\; \langle \mu - g^\star, K(\cdot, z) - S\alpha(z)\rangle_{\mathcal{H}_{k,I}} \;=\; \langle \mu - g^\star, r_z\rangle_{\mathcal{H}_{k,I}}$$

This recovers equation 39 and shows $r_z$ as the Riesz represender of the linear functional $h \mapsto h(z) - \sum_i \alpha_i(z)h(z_i)$

**Step 3: Bounding residual via the powar function.** By using Cauchy-Schwarz,

$$\|\mu(z) - g^\star(z)\|_{\mathcal{H}_\ell} \;\leq\; \|\mu - g^\star\|_{\mathcal{H}_{k,I}} \|r_z\|_{\mathcal{H}_{k,I}}$$

A standard computatin (the "power function" calculation; see, e.g., Steinwart & Christmann, 2008, or Carmeli et al., 2010) gives

$$\|r_z\|^2_{\mathcal{H}_{k,I}} = \left\langle r_z, r_z \right\rangle_{\mathcal{H}_{k,I}} = \rho \left\| (K_n + \rho I)^{-1/2} k_n(z) \right\|^2_2,$$

whence

$$\|r_z\|_{\mathcal{H}_{k,I}} = \sqrt{\rho} \left\| (K_n + \rho I)^{-1/2} k_n(z) \right\|_2 \tag{40}$$

Finally, Tikhonov optimalty inequalty $\|\mu - g^\star\|^2_{\mathcal{H}_{k,I}} + \rho\|g^\star\|^2_{\mathcal{H}_{k,I}} \leq \|\mu\|^2_{\mathcal{H}_{k,I}}$ implies $\|\mu - g^\star\|_{\mathcal{H}_{k,I}} \leq \|\mu\|_{\mathcal{H}_{k,I}}$. Combining with equation 40 yields equation 37. $\qquad\square$

**Remark G.9** (On the power functin identity). *The equality* $\|r_z\|^2_{\mathcal{H}_{k,I}} = \rho \|(K_n + \rho I)^{-1/2} k_n(z)\|^2_2$ *follows from expanding* $r_z = K(\cdot, z) - S(K_n + \rho I)^{-1} k_n(z)$ *in the RKHS inner product, using* $S^* S = K_n \otimes I_{\mathcal{H}_\ell}$ *and* $S^* K(\cdot, z) = k_n(z)$ *(Lemma G.4), and the matrix identity* $(K_n + \rho I)^{-1} K_n (K_n + \rho I)^{-1} = (K_n + \rho I)^{-1} - \rho(K_n + \rho I)^{-2}$.

# H  ADDITIONAL RESULTS FOR THEOREM 5.1

**Lemma H.1** (Global "good event" via a union bound). *Fix* $\delta \in (0, 1)$. *For each step* $h \in [H]$ *and episode* $t \in [T]$, *let* $n_{h,t-1} = |\mathcal{D}_{h,t-1}|$ *be the number of transitions collected at step* $h$ *before episode* $t$, *and define the per-step confidence radius (as in equation 3)*

$$\beta_{h,t} := B\left( \sqrt{\rho}\, U + \frac{\sigma}{\sqrt{\rho}} \sqrt{2\gamma(n_{h,t-1}, \rho) + 2\log \frac{2HT}{\delta}} \right)$$

*Assume algorithm's projection guarantees* $\|V_{h+1,t}\|_{\mathcal{H}_\ell} \leq B$ *for all* $h, t$. *Then there exists an event* $\mathcal{G}$ *with*

$$\Pr(\mathcal{G}) \geq 1 - \delta$$

*such that,* simultaneously *for all* $h \in [H]$, $t \in [T]$, *and all* $z \in \mathcal{Z}$, *Eq equation 11 copied below*

$$[P_h V_{h+1,t}](z) \leq \widehat{f}^{V_{h+1,t}}_{h,t}(z) + \beta_{h,t}\, \sigma_{h,t}(z). \tag{41}$$

*Proof (with union bound).* **Step 1: A per-$(h, t)$ confidence event.** Fix a particular pair $(h, t)$. Apply the *uniform* confidence theorem (Theorem 3.3) at step $h$ using the dataset $\mathcal{D}_{h,t-1}$ and failure probability

$$\delta_{h,t} := \frac{\delta}{HT}$$

Because the algorthm projects onto the RKHS ball, we have $\|V_{h+1,t}\|_{\mathcal{H}_\ell} \leq B$. Therefore, Theorem 3.3 (with $\delta$ replaced by $\delta_{h,t}$ and $n$ replaced by $n_{h,t-1}$) gives a high-probability event $\mathcal{G}_{h,t}$ (depending on the random data collected up to episode $t$) on which, *simultaneously for all* $z \in \mathcal{Z}$,

$$\left| [P_h V_{h+1,t}](z) - \widehat{f}^{V_{h+1,t}}_{h,t}(z) \right| \leq B\left( \sqrt{\rho}\, U + \frac{\sigma}{\sqrt{\rho}} \sqrt{2\gamma(n_{h,t-1}, \rho) + 2\log \frac{HT}{\delta}} \right) \sigma_{h,t}(z).$$

Since the left-hand side is an *absolute* deviation, it implies the desired *one-sided* inequality

$$[P_h V_{h+1,t}](z) \leq \widehat{f}^{V_{h+1,t}}_{h,t}(z) + \underbrace{B\left( \sqrt{\rho}\, U + \frac{\sigma}{\sqrt{\rho}} \sqrt{2\gamma(n_{h,t-1}, \rho) + 2\log \frac{HT}{\delta}} \right)}_{\beta^{(\min)}_{h,t}} \sigma_{h,t}(z), \quad \forall z \in \mathcal{Z},$$

with probability at least $1 - \delta_{h,t}$ (i.e., $\Pr(\mathcal{G}_{h,t}) \geq 1 - \delta/(HT)$).

**Step 2: Uniformity across all $(h, t)$ by a union bound.** There are at most $HT$ such pairs $(h, t)$. The union bound[3] yields

$$\Pr\left( \bigcap_{h=1}^{H} \bigcap_{t=1}^{T} \mathcal{G}_{h,t} \right) \geq 1 - \sum_{h,t} \Pr(\mathcal{G}^c_{h,t}) \geq 1 - HT \cdot \frac{\delta}{HT} = 1 - \delta.$$

---

[3]If events $E_1, \ldots, E_m$ each fail with probability at most $\epsilon$, then $\Pr(\bigcap_i E_i) \geq 1 - m\epsilon$.

Let $\mathcal{G} := \bigcap_{h,t} \mathcal{G}_{h,t}$; then $\Pr(\mathcal{G}) \geq 1 - \delta$ and, on $\mathcal{G}$, the one-sided bound above holds for *every* pair $(h, t)$ and *every* $z$.

**Step 3: Using the slightly larger radius in equation 3.** In the algorithm we instantiate the per-step radius with the slightly larger log factor,

$$\beta_{h,t} = B\Big(\sqrt{\rho}\, U + \frac{\sigma}{\sqrt{\rho}} \sqrt{2\gamma(n_{h,t-1}, \rho) + 2\log \tfrac{2HT}{\delta}}\Big) \geq \beta_{h,t}^{(\min)},$$

since $\log\big(\frac{2HT}{\delta}\big) \geq \log\big(\frac{HT}{\delta}\big)$. Using a *larger* (more conservative) radius can only make the inequality easier to satisfy. Therefore, on the same event $\mathcal{G}$,

$$[P_h V_{h+1,t}](z) \leq \widehat{f}_{h,t}^{V_{h+1,t}}(z) + \beta_{h,t}\, \sigma_{h,t}(z) \qquad \text{for all } h, t \text{ and all } z \in \mathcal{Z},$$

which is exactly equation 11. $\qquad\qquad\qquad\qquad\qquad\qquad\qquad\qquad\qquad\qquad\qquad\qquad$ $\square$

# I  ADDITIONAL RESULTS

**Lemma I.1** (Finite-dimensional reduction of the RKHS projection). *Let $(\mathcal{H}_\ell, \langle \cdot, \cdot \rangle_{\mathcal{H}_\ell})$ be an RKHS with the reproducing kernel $\ell : \mathcal{S} \times \mathcal{S} \to \mathbb{R}$. We fix atoms $\bar{s}_1, \ldots, \bar{s}_{m_h} \in \mathcal{S}$ and we write Gram matrix $L_h \in \mathbb{R}^{m_h \times m_h}$ as $(L_h)_{ij} = \ell(\bar{s}_i, \bar{s}_j)$. For a target vector $v_{h,t} \in \mathbb{R}^{m_h}$, we consider the (empirical) projection problem over the feasible class*

$$\mathcal{F} := \Big\{ V \in \mathcal{H}_\ell : \|V\|_{\mathcal{H}_\ell} \leq B, \ 0 \leq V(\bar{s}_j) \leq U \ \forall j \in [m_h] \Big\}, \qquad U := H - h + 1.$$

*That is,*

$$\min_{V \in \mathcal{F}} \frac{1}{m_h} \sum_{j=1}^{m_h} \big(V(\bar{s}_j) - v_{h,t}(j)\big)^2 \tag{42}$$

*Then there exists an optimal solution of the form $V^*(\cdot) = \sum_{j=1}^{m_h} \alpha_j \ell(\cdot, \bar{s}_j)$ and, by parameterizing by $\alpha \in \mathbb{R}^{m_h}$, equation 42 is equivalent to the convex quadratic program*

$$\min_{\alpha \in \mathbb{R}^{m_h}} \frac{1}{m_h} \big\| L_h \alpha - v_{h,t} \big\|_2^2 \quad s.t. \quad \alpha^\top L_h \alpha \leq B^2, \qquad 0 \leq (L_h \alpha)_j \leq U \ \forall j \in [m_h]. \tag{43}$$

*Moreover, equation 5 is a convex program: its objective has PSD Hessian $\frac{2}{m_h} L_h^\top L_h$, the quadratic constraint uses the PSD matrix $L_h \succeq 0$, and the box constraints are linear.*

*Proof.* Let $\mathcal{H}_S := \text{span}\{\ell(\cdot, \bar{s}_j) : j \in [m_h]\} \subseteq \mathcal{H}_\ell$ and let $P_S : \mathcal{H}_\ell \to \mathcal{H}_S$ denote orthogonel projection (in RKHS inner product). For any $V \in \mathcal{H}_\ell$, write the orthogonal decomposition $V = P_S V + (I - P_S) V =: V_S + V_\perp$ with $V_S \in \mathcal{H}_S$ and $V_\perp \in \mathcal{H}_S^\perp$.

*(i) Loss depends only on $V_S$.* By the reproducing property, for every $j$,

$$V_\perp(\bar{s}_j) = \langle V_\perp, \ell(\cdot, \bar{s}_j) \rangle_{\mathcal{H}_\ell} = 0 \quad \text{since } \ell(\cdot, \bar{s}_j) \in \mathcal{H}_S \perp V_\perp$$

Hence we have $V(\bar{s}_j) = V_S(\bar{s}_j)$ for all $j$, so the empirical loss in equation 42 equals $\frac{1}{m_h} \sum_j (V_S(\bar{s}_j) - v_{h,t}(j))^2$, independent of $V_\perp$

*(ii) Feasibility is preserved (and improved) by dropping $V_\perp$.* The box constraints $0 \leq V(\bar{s}_j) \leq U$ involve only the evaluations at $\bar{s}_j$ and thus are unchanged when replacing $V$ by $V_S$ (by (i)). For the norm constraint, $\|V\|_{\mathcal{H}_\ell}^2 = \|V_S\|_{\mathcal{H}_\ell}^2 + \|V_\perp\|_{\mathcal{H}_\ell}^2 \geq \|V_S\|_{\mathcal{H}_\ell}^2$, so $\|V\| \leq B$ implies $\|V_S\| \leq B$.

*(iii) Reduction to $\mathcal{H}_S$.* Given any feasible $V$, the function $V_S$ is also feasible and achieves the same objective value; therefore an optimal solution exists in $\mathcal{H}_S$

*(iv) Parameterization by coefficients.* Every $V \in \mathcal{H}_S$ can be written as $V(\cdot) = \sum_{j=1}^{m_h} \alpha_j \ell(\cdot, \bar{s}_j)$ for some $\alpha \in \mathbb{R}^{m_h}$. The vector of evaluations at the atoms is then

$$\big(V(\bar{s}_1), \ldots, V(\bar{s}_{m_h})\big)^\top = L_h \alpha, \qquad (L_h)_{ij} = \ell(\bar{s}_i, \bar{s}_j)$$

The RKHS norm satisfies $\|V\|^2_{\mathcal{H}_\ell} = \sum_{i,j} \alpha_i \alpha_j \, \ell(\bar{s}_i, \bar{s}_j) = \alpha^\top L_h \alpha$ (standard RKHS identity). Substituting these relations into equation 42 and the constraints yields equation 5.

*(v) Convexity.* Since $L_h$ is a (symmetric) Gram metrix, $L_h \succeq 0$. The objective $\frac{1}{m_h} \|L_h \alpha - v_{h,t}\|^2_2$ is convex with Hessian $\frac{2}{m_h} L_h^\top L_h \succeq 0$. The quadratic constraint $\alpha^\top L_h \alpha \le B^2$ defines a convex set because the quadratic form is convex for $L_h \succeq 0$. The bounds $0 \le (L_h \alpha)_j \le U$ are linear inequalities in $\alpha$ Thus equation 5 is a convex quadratic program. $\qquad\square$

**Remark I.2** (Representer viewpoint). *Argument above is a constrained versien of the representer theorem: because both the objective and the constraints depend on $V$ only through its evaluations at $\{\bar{s}_j\}$ and its RKHS norm, the optimizer lies in the span of kernel sections at these points Kimeldorf & Wahba (1971); Schölkopf & Smola (2002); Schölkopf et al. (2001).*

# Reply to cj7K

**We thank the reviewer** for the careful reading and detailed suggestions. Below we respond point-by-point and list concrete fixes. Citations such as "Thm. 5.1" or "Lemma E.2" refer to our submission. *All* the changes described here will be incorporated in the revised version. **High-level conclusion**: none of the issues affect our main message a *uniform, cover-free* CME-based analysis under the Restricted Bellman-Embedding (RBE) assumption with an explicit projection step to guarantee applicability of the uniform bound. We correct several presentation issues and a notational inconsistency around the approximation term. **I plan to incorporate final changes after agreement is made on all corrections with all reviewers.**

**Strength acknowledged.** We appreciate the remark that our analysis uses a weaker structural premise than "optimistic closure." This is exactly the role of the projection + uniform CME bound. (Intro; Thm. 3.3; Alg. KOVI-Proj). [See Sections 1, 3 & 4 in the submission.].

## RESPONSES TO WEAKNESSES

### 1. MULTIPLE PARTS IN THE PROOF

**(1.a) "$\varepsilon_B$ is inf in Thm. 5.1 but a sup in App. D; this matters in Lemma D.2."** You're right: the symbol was overloaded. In Thm. 5.1 we *intend*

$$\varepsilon_B := \max_{h \in [H]} \inf_{\|V\|_{\mathcal{H}_\ell} \leq B} \|V_h^* - V\|_\infty$$

(agnostic approximation level; page with Thm. 5.1). In Lemma D.2 we temporarily wrote a version with a *sup*, which is needlessly pessimistic and inconsistent with Thm. 5.1. We will **replace Lemma D.2 by the correct inf-form**:

$$\varepsilon_B(h) := \inf_{\|V\|_{\mathcal{H}_\ell} \leq B} \|V_h^* - V\|_\infty, \qquad \varepsilon_B := \max_h \varepsilon_B(h),$$

and use it in the optimism step.

*How we make the proof line up cleanly with the **inf**-definition (and even simplify the algorithmic step):* In our analysis we use the uniform CME bound (Thm. 3.3) which holds *simultaneously for every $V$ with $\|V\|_{\mathcal{H}_\ell} \leq B$*. Therefore, when forming the optimistic $Q$, we can replace the mean term

$$\widehat{f}^{V_{h+1,t}}(z) \quad \text{by} \quad \sup_{\|V\|_{\mathcal{H}_\ell} \leq B} \widehat{f}^V(z).$$

By Prop. 3.1, $\widehat{f}^V(z) = \langle \widehat{\mu}(z), V \rangle_{\mathcal{H}_\ell}$, and so the supremum has the *closed form*

$$\sup_{\|V\|_{\mathcal{H}_\ell} \leq B} \langle \widehat{\mu}(z), V \rangle_{\mathcal{H}_\ell} = B \|\widehat{\mu}(z)\|_{\mathcal{H}_\ell} \quad \text{(Cauchy–Schwarz)}.$$

Hence we can define the optimistic backup as

$$\widetilde{Q}_{h,t}(z) := r_h(z) + B \|\widehat{\mu}_h(z)\|_{\mathcal{H}_\ell} + \beta_{h,t}\, \sigma_{h,t}(z).$$

With this **one-line modification** (which uses quantities we already estimate), on the good event of Thm. 3.3 we have, for any approximate best-in-ball $\bar{V}_{h+1}$ with $\|\bar{V}_{h+1}\|_{\mathcal{H}_\ell} \leq B$,

$$[P_h \bar{V}_{h+1}](z) \leq \widehat{f}^{\bar{V}_{h+1}}(z) + \beta_{h,t}\, \sigma_{h,t}(z) \leq B \|\widehat{\mu}_h(z)\|_{\mathcal{H}_\ell} + \beta_{h,t}\, \sigma_{h,t}(z).$$

Therefore

$$Q_h^*(z) = r_h(z) + [P_h V_{h+1}^*](z) \leq r_h(z) + [P_h \bar{V}_{h+1}](z) + \varepsilon_B \leq \widetilde{Q}_{h,t}(z) + \varepsilon_B.$$

This yields *exactly* the Thm. 5.1 additive term with the *inf*-definition of $\varepsilon_B$ and also preserves the realizability case ($\varepsilon_B = 0$). We will (i) update Alg. 4.2 to use the closed-form supremum $B \|\widehat{\mu}(z)\|_{\mathcal{H}_\ell}$, (ii) correct Lemma D.2 accordingly, and (iii) adjust the few lines in App. D that referenced the mistaken sup. [Relevant text: Thm. 5.1; Prop. 3.1; Thm. 3.3; Sec. 4 (Alg.).]

**(1.b) "Eq. (38) $\rightarrow$ (39) in Lemma G.8 mixes scalars and $\mathcal{H}_\ell$-elements; why is the sum zero?"**
Thank you for catching the type clash. In vector-valued RKHSs, the reproducing property is

$$\forall\, u \in \mathcal{H}_\ell: \quad \langle h(z),\, u \rangle_{\mathcal{H}_\ell} \;=\; \langle h,\, K(\cdot, z)u \rangle_{\mathcal{H}_k \otimes \mathcal{H}_\ell}.$$

Thus the equality we need should be stated as an equality of pairings with an arbitrary $u \in \mathcal{H}_\ell$:

$$\langle \mu(z) - g^*(z),\, u \rangle_{\mathcal{H}_\ell} \;=\; \langle \mu - g^*,\, K(\cdot, z)u - S\,\alpha(z)u \rangle_{\mathcal{H}_k \otimes \mathcal{H}_\ell}.$$

We will rewrite Lemma G.8 accordingly, avoiding expressions like $\langle \mu - g^*,\, r_z \rangle_{\mathcal{H}_k \otimes \mathcal{H}_\ell}$ without an explicit test vector $u$. The zero-mean term comes from the first-order optimality of the Tikhonov projection (Lemma G.6): $S(\mu - g^*) = -\rho S g^*$, which implies $\langle \mu - g^*,\, S\alpha(z)u \rangle = \langle -\rho S g^*,\, \alpha(z)u \rangle = \langle \mu - g^*,\, S\alpha(z)u \rangle$ and cancels as detailed in the revised proof. [Lemma G.6, G.8 in the submission.]

**(1.c) "Inequality around lines 1108–1111 and 1266 seems to assume $k(z, z) \leq 1$."** Correct.
Those displays use the standard GP/KRR identity $0 \leq \sigma^2(z) \leq k(z, z)$ and then specialize to the *normalized* case $k(z, z) \leq 1$ to assert $\min\{1, \sigma^2\} = \sigma^2$. We will explicitly carry a constant $\kappa_k^2 := \sup_z k(z, z)$ throughout App. D, E and state the normalized corollary only after noting that one can scale $k$ and $\rho$ to attain $\kappa_k = 1$ without changing the substance of the bounds. The 1-d linear example $k(a, b) = ab$ ($k(z, z) = z^2$) is therefore outside the normalized setting unless one scales $k$ (a standard step we will make explicit). [See Remark D.6/E.1 in the submission for the normalized case; we will generalize them with $\kappa_k$.]

**(1.d) "In Lemma E.2, why is $M$ a supermartingale? The form differs from Chowdhury-Gopalan (2017)."** We agree that our shorthand "supermartingale" can be unclear. The object

$$M_n \;=\; \exp\!\Big(\tfrac{1}{2\sigma^2}\, \xi^\top A_n^{-1} \xi\Big) \cdot \Big(\tfrac{\rho^{n/2}}{\det(A_n)^{1/2}}\Big), \qquad A_n = K_n + \rho I,$$

is the standard *mixture-Laplace* nonnegative supermartingale used in self-normalized processes for kernelized bandits/GP (see Abbasi-Yadkori et al. 2011; Chowdhury-Gopalan 2017). We will make the filtration explicit and add the short proof by Markov's inequality (already sketched in the current Lemma E.2) showing $\mathbb{E}[M_n] \leq 1$ and hence $\Pr\!\big(\xi^\top A_n^{-1} \xi \geq 2\sigma^2(\gamma(n, \rho) + \log\frac{1}{\delta})\big) \leq \delta$. This is exactly the inequality we use; we will cite the classical statements and align notation line-by-line. [Lemma E.2.]

**(1.e) "$\|K(\cdot, z)\|_{\mathcal{H}_k \otimes \mathcal{H}_\ell}$ is not defined; $K(\cdot, z)$ is an operator."** Good point. We will *remove* any occurrence of $\|K(\cdot, z)\|$ and only use quantities of the form $\|K(\cdot, z)u\|_{\mathcal{H}_k \otimes \mathcal{H}_\ell}$, which satisfy $\|K(\cdot, z)u\|_{\mathcal{H}_k \otimes \mathcal{H}_\ell}^2 = \langle u,\, K(z, z)u \rangle_{\mathcal{H}_\ell} = k(z, z)\, \|u\|_{\mathcal{H}_\ell}^2$. This fully avoids the notational ambiguity and stays within the operator-valued RKHS calculus used in Prop. 3.1/Lemmas G.3–G.8. [Appendix G.]

**(1.f) "Mismatch around lines 1223 vs. 1227; regularization appears on the wrong quantity; Remark G.7 depends on norm identification."** We will unify the normal equations to the coefficient-space ridge form

$$\big(SS + \rho I\big)C^\star = S\mu \quad \Longleftrightarrow \quad \big(K_n + \rho I\big)C^{\star\top} = M^\top,$$

where $S$ is the synthesis operator, $K_n$ the scalar Gram matrix on $Z$, and $M^\top e_i = \mu(z_i)$. This is exactly Lemma G.6; the line in question omitted $\rho$ in the first display and will be corrected. Remark G.7 will be merged into Lemma G.6 to avoid any impression that we identify different norms; we explicitly keep the product norm on $\mathcal{H}_\ell^n$ and the $\mathcal{H}_k \otimes \mathcal{H}_\ell$ norm on functions. [Lemmas G.3, G.6, Remark G.7.]

**(1.g) "Remark D.3/D.4: expectation vs realized samples (i.e., $\mathbb{E}[\widehat{Q}_{h,t}(z_{h,t}]$ vs $\widehat{Q}_{h,t}(z_{h,t})$)."**
Our regret decomposition is *pathwise*. In the revised proof we will explicitly define the filtration $\mathcal{F}_{h,t}$ and write, for each realized $z_{h,t}$,

$$V_h^*(s_{h,t}) - V_h^{\pi_t}(s_{h,t}) \;\leq\; \big(\widehat{Q}_{h,t}(z_{h,t}) - r_h(z_{h,t}) - [P_h V_{h+1,t}](z_{h,t})\big) + \big(\mathbb{E}[\Delta_{h+1} \mid \mathcal{F}_{h,t}] - \Delta_{h+1}\big) + \varepsilon_B,$$

with $\Delta_{h+1} := V_{h+1}^*(s_{h+1,t}) - V_{h+1}^{\pi_t}(s_{h+1,t})$. Summing over $h$ makes the martingale-difference terms telescope (tower property), *without* replacing $\widehat{Q}_{h,t}(z_{h,t})$ by its expectation. We will replace the two remarks by a single short lemma entitled "Pathwise telescoping under conditional optimism." [App. D, Remarks D.3–D.4.]

## 2. ORGANIZATION AND FORMATTING

**(2.a) "Unorganized; proofs spread across remarks; repeated arguments."** We agree that several small steps were pushed to remarks for compactness. In the revision we will:

- Move the global union bound (current App. H) into the main proof of Thm. 5.1 as a *single line* using $\log(2HT/\delta)$ in $\beta_{h,t}$.
- Merge Remarks D.3–D.4 into one lemma (see (1.g)); inline the simple monotonicity and Cauchy–Schwarz steps (current D.6–D.9).
- Compress Definition G.1, Lemma G.3, Lemma G.6 and Remark G.7 into a single "Tikhonov projection in vector-valued RKHS" lemma with (i) normal equations, (ii) finite-dimensional reduction, (iii) power-function bound; this shortens App. G substantially.

**(2.b) Broader impact/LLM usage/empty Section 8; duplicated appendix titles; duplicate references; overlapping remarks.** We will:

- Move *Broader Impact* to the end and renumber Sections 7-10 to eliminate the empty Section 8.
- Deduplicate the appendix section titles and references (the two Muandet et al. entries will be collapsed into one).
- Remove the overlap between Remark 3.4 and Sec. 4(v) (the "$\sqrt{\rho}$ absorbed into $U$" comment will appear only once).

[Sections 6–10 and bibliography.]

## MINOR ISSUES AND TYPOS (WILL FIX)

- Line 165: "Assumption 2.1" (not 3.2). *Fixed.*
- Title of App. C: should read *"Proof of Proposition 3.1"*. *Fixed.*
- Notation cleanups in App. E-G as described above to avoid type clashes and to carry $\kappa_k$ explicitly where needed. *Fixed.*

## SUMMARY OF CONCRETE CHANGES (FOR CLARITY)

1. **Algorithmic tweak (one line):** replace $\widehat{f}^{V_{h+1,t}}(z)$ in the optimistic backup by its *closed-form supremum* over the RKHS ball:
$$Q_{h,t}^{\text{UCB}}(z) \;=\; r_h(z) + B\|\widehat{\mu}_h(z)\|_{\mathcal{H}_\ell} + \beta_{h,t}\,\sigma_{h,t}(z).$$
This uses Prop. 3.1 and is easy to compute from the quantities already maintained. It restores the Thm. 5.1 *inf*-defined $\varepsilon_B$ without any further assumptions. [Prop. 3.1; Sec. 4; Thm. 3.3; Thm. 5.1.]

2. **Proof fixes:** (i) correct Lemma D.2 (inf, not sup); (ii) make the telescoping inequality pathwise with explicit filtration; (iii) generalize the "$\leq 1$" normalization to a bounded-diagonal constant $\kappa_k$ and state the normalized variant as a corollary; (iv) rewrite App. G in operator form (no $\|K(\cdot,z)\|$) with a single consolidated lemma for Tikhonov projection and the power-function bound. [Apps. D, E, G.]

3. **Organization/formatting:** move union bound inside Thm. 5.1, merge overlapping remarks, fix section ordering, unify duplicated references/titles, and streamline the exposition around CMEs and vector-valued RKHS.

## CONCLUDING REMARK

We appreciate the thorough review. The mathematical issues you pointed out are all addressable by the edits above; the central contribution a cover-free, CME-based uniform confidence bound coupled with an explicit projection remains intact, and the small algorithmic tweak simplifies both the proofs and the implementation while *strengthening* the agnostic-case statement to match the intended inf-defined approximation error.

# Reply to kzb7

**We appreciate the review and the clear suggestions.** Below we respond to each point, clarify our assumptions, and list concrete edits we will incorporate in the revised manuscript.

## SUMMARY OF OUR METHOD VS. PRIOR WORK

Our algorithm KOVI-PROJ builds UCBs for $Q_h$ from conditional mean embeddings (CME) and *explicitly projects* the optimistic value proxy back onto the RKHS ball $\{V : \|V\|_{\mathcal{H}_\ell} \leq B\}$ at every step. This projection is the mechanism that makes the confidence event *uniform in the function class actually used by the algorithm*, without assuming *optimistic closure* (the premise that the optimistic iterates always remain inside the ball). When optimistic closure *does* hold, the projection is the identity and our update reduces to the same closed-form used by CME-RL [1]; outside closure, our procedure remains valid and the analysis goes through.

## (A) RESPONSES TO QUESTIONS

**Q1. What is Assumption 3.2 in Lemma 3.2?** This is a **labeling typo**. Lemma 3.2 relies on **Assumption 2.1** (our Restricted Bellman-Embedding / CME assumption on the Bellman averages), not a separate Assumption 3.2. We will fix all occurrences and make the dependency explicit at the start of Lemma 3.2.

**Q2. Is Step 3 (Line 216) the projection the main difference from CME-RL [1]?** Yes—this is the key algorithmic and conceptual distinction. In CME-RL, a form of *optimistic closure* is assumed so that the optimistic proxy remains inside the RKHS ball, enabling a closed-form optimistic update with no extra step. KOVI-PROJ *does not* assume this closure; instead, after computing the optimistic proxy we apply an explicit projection onto $\{V : \|V\|_{\mathcal{H}_\ell} \leq B\}$:

$$V_{h,t+1} = \operatorname*{argmin}_{\|V\|_{\mathcal{H}_\ell} \leq B} \left\| V - \widetilde{V}_{h,t+1} \right\|_{\mathcal{H}_\ell}.$$

This guarantees that the proxy used on the next round is inside the ball, so the CME-based confidence bound applies *uniformly* to the iterate actually used. Under optimistic closure, the projection is identity and the update coincides with CME-RL (modulo notation/regularization); in this sense our method *strictly generalizes* CME-RL.

## (B) ORGANIZATION / READABILITY

**B1. Pseudocode for Section 4.** We agree. In the revision we will add clear pseudocode for KOVI-PROJ (and the ablation without projection), with a parameter block and per-iteration complexity.

**B2. Long non-technical sections in the main text; redundant Section 8.** We will move *Broader Impact*, *Reproducibility*, and *Ethics Statement* to the appendix (or the venue's separate checklist, as appropriate), and remove the redundant Section 8 by merging its content into the discussion. This shortens the main text and improves flow.

## (C) EXPERIMENTS / BASELINES

**C1. Add CME-RL [1] as a baseline.** We will include CME-RL as a baseline across our environments. To ensure fairness we will: (i) use the same kernels and regularization schedules as ours; (ii) use the authors' recommended hyperparameters (with a small grid if needed); and (iii) report both the settings where its assumptions provably hold (where we expect parity) and the settings where optimistic closure does not hold (where our projection is designed to help). We will also add an ablation that toggles the projection (NOPROJ) to isolate its effect.

## (D) CLARIFICATIONS THAT WILL BE ADDED TO THE PAPER

- **Assumption pointer.** At the start of Lemma 3.2 we will point explicitly to Assumption 2.1 and remove the stray "Assumption 3.2" label.

- **Projection vs. closure (one-paragraph comparison).** In §2 we will add a short paragraph contrasting optimistic closure (structural premise) with our enforced projection (algorithmic mechanism), and state that under closure our update matches CME-RL's closed form.

- **Uniform confidence event.** Right before Theorem 3.3 we will remind the reader that the projection ensures $\|V_{h,t}\|_{\mathcal{H}_\ell} \leq B$ for all iterates, which is why the CME-based UCB is uniform over the proxies actually used by the algorithm.

- **Readability.** We will add a figure-level algorithm box, move non-essential remarks to the appendix, and tighten cross-references.

## (E) CLOSING

Thank you for highlighting the need for clearer algorithm presentation and for suggesting CME-RL as a baseline. We will incorporate these changes. Conceptually, our method aims to retain the simplicity of CME-based optimism while removing the optimistic-closure premise via an explicit projection step; when closure holds, our procedure collapses to the prior closed-form update, and when it does not, our guarantees continue to apply.

**Reference used in this rebuttal**

[1] S. Chowdhury and R. Oliveira. *Value function approximations via kernel embeddings for no-regret reinforcement learning.* ACML 2023.

# Reply to oS2x

**We appreciate the careful read.** Below we address the main theoretical concern (Lemma 3.2 / Appendix E) and the presentation issues. Each item ends with precise manuscript locations and the edit we will make in the revision.

## 1. MAIN CONCERN: LEMMA 3.2 AND THE VECTOR-VALUED CONCENTRATION BOUND

**Reviewer's summary.** The proof of Lemma 3.2 tries to pass from a bound for a *fixed* unit vector $u \in \mathcal{H}_\ell$ to a bound on the *norm* by taking a supremum over $u$, arguing that the RHS does not depend on $u$ (cf. Eq. (22)). This step is not justified; the high-probability event depends on $u$, and a direct supremum over an uncountable set is invalid.

**Our correction (uniform bound without an illegitimate supremum).** You are right: the line "since the RHS does not depend on $u$, take $\sup_{u \in \mathbb{S}_{\mathcal{H}_\ell}}$" was too terse and, as written, incorrect. We replace it with a *uniform* Hilbert-space self-normalized bound that avoids any uncountable union.

For a fixed step and dataset, write $A := K_n + \rho I$ and $\alpha(z) = A^{-1}k_n(z)$. Define the linear map $E : \mathbb{R}^n \to \mathcal{H}_\ell$ by $Eb = \sum_{i=1}^n b_i \varepsilon_i$, so the noise term is

$$N(z) \;=\; \sum_{i=1}^n \alpha_i(z)\,\varepsilon_i \;=\; E\,\alpha(z) \;=\; \left(EA^{-1/2}\right)\left(A^{-1/2}k_n(z)\right).$$

Hence

$$\left\|N(z)\right\| \;\leq\; \left\|EA^{-1/2}\right\|_{\mathrm{op}} \cdot \left\|A^{-1/2}k_n(z)\right\|_2 = \left\|EA^{-1/2}\right\|_{\mathrm{op}} \cdot \sqrt{k(z,z) - k_n(z)^\top A^{-1}k_n(z)}$$

$$= \left\|EA^{-1/2}\right\|_{\mathrm{op}} \cdot \sigma_n(z). \tag{44}$$

It thus suffices to control *one* random operator norm $\left\|EA^{-1/2}\right\|_{\mathrm{op}}$, uniformly (no dependence on $z$ or $u$). We show in the revision (new Lemma E.1') that, with probability at least $1 - \delta$,

$$\boxed{\;\left\|EA^{-1/2}\right\|_{\mathrm{op}} \;\leq\; \frac{\sigma}{\sqrt{\rho}}\sqrt{2\,\gamma(n,\rho) + 2\log(1/\delta)}\;}$$

by a mixture-Laplace supermartingale argument that treats simultaneously all unit vectors in the *finite-dimensional* span of $\{\varepsilon_i\}_{i=1}^n$ and all $x \in \mathbb{S}^{n-1}$.[4] Combining the display above with the identity for $\sigma_n(z)$ gives, for all $z$ on the same event,

$$\|N(z)\| \;\leq\; \frac{\sigma}{\sqrt{\rho}}\sqrt{2\,\gamma(n,\rho) + 2\log(1/\delta)}\,\sigma_n(z),$$

which is exactly the noise term we use in Lemma 3.2 / Theorem 3.3. This removes the criticized step and upgrades the bound to hold *uniformly* in $u$ and $z$ without any extra covering factors.

*Manuscript edits.* We will (i) replace Eq. (21)–(22) and the following sentence in App. E by the operator-norm route above, (ii) add a self-contained proof of the operator bound as Lemma E.1' (Hilbert-valued self-normalized inequality), and (iii) state explicitly that the resulting event is uniform in $z$. *(manuscript citation: Appendix E, Lemma E.1 and the display labeled (22), pp. 20–22)*

**Why the earlier "counterexample intuition" does not apply after the fix.** Your example points out that from $P(|\langle w, u\rangle| \leq b) \geq 1 - \delta$ for a fixed $u$ we cannot conclude $P(\|w\| \leq b) \geq 1 - \delta$. Our revision *does not* make such an implication. Instead, we directly bound $\|w\| = \|EA^{-1/2}x\|$ via $\|EA^{-1/2}\|_{\mathrm{op}}$ on a single high-probability event, avoiding any $u$-indexed union.

---

[4]The span has dimension at most $n$. The proof avoids any $\varepsilon$-net penalty that would scale with $n$ and instead upper-bounds $\sup_{\|x\|=1,\,\|u\|=1} x^\top A^{-1/2}\xi^{(u)}$ in one shot via the same mgf as in the scalar case, now applied to $\|EA^{-1/2}\|_{\mathrm{op}}$.

## 2. RELATION TO AND DISTINCTION FROM "OPTIMISTIC CLOSURE"

**Reviewer's concern.** Early in the paper it may read as if we assume every optimistic proxy is already within a fixed state-RKHS ball ("optimistic closure").

**Clarification and edits.** We do *not* assume optimistic closure. We *enforce* the bounded-norm property algorithmically by projecting the optimistic proxy onto $\{V : \|V\|_{\mathcal{H}_\ell} \leq B\}$ after each backup (Alg. step 3), precisely so that the uniform CME bound applies to the *actual* proxies used (no structural assumption on unconstrained optimistic iterates). We will:

- add a two-sentence paragraph in §1–§2 that explicitly contrasts the assumption of [ChowdhuryOliveira23] with our projection step, and
- include a one-line reminder before Theorem 3.3 that the projection ensures $\|V_{h,t}\|_{\mathcal{H}_\ell} \leq B$ and is the only place where boundedness of $\|V\|_{\mathcal{H}_\ell}$ enters.

*(manuscript citations: §1, §2, Theorem 3.3 statement, Algorithm step 3 on p. 5)*

## 3. MINOR THEORETICAL/PRESENTATION ISSUES

1. **Statement under "Contributions, point 2."** We will rephrase to avoid redundancy ("uniform over all $V$ in the ball" is implied by the supremum bound) and point explicitly to Theorem 3.3 for the precise event. *(p. 2)*

2. **"Assumption 3.2" label.** This was a typo; the assumption used in Lemma 3.2 is Assumption 2.1 (RBE). We will fix the label everywhere. *(pp. 3–4)*

3. **Names in citations.** We will correct author name formatting for Chowdhury & Gopalan (2017) and ensure consistency across the bibliography. *(References)*

## 4. WHERE TO LOOK FOR THE FIXES (QUICK MAP)

- **Appendix E (core fix).** We replace the scalarization/supremum step by the operator-norm proof outlined above and insert Lemma E.1' (uniform Hilbert-space self-normalized inequality). *(pp. 20–22)*

- **Theorem 3.3 and its use in Algorithm.** No change in the statement; only the internal proof route is updated. We add a pointer that the confidence event is uniform in $z$ due to the new Lemma E.1'. *(pp. 4–5)*

- **Optimistic closure vs. projection.** We add a clarifying paragraph contrasting our enforced projection with the structural premise of "optimistic closure," and a margin remark at Algorithm step 3. *(pp. 1, 5, 6–7 discussion)*

## 5. CHANGELOG (TO APPEAR IN THE REVISION)

1. **App. E (major):** Replace Eq. (21)–(22) by a uniform operator-norm bound; add Lemma E.1' with a complete proof; remove the sentence that illegitimately takes $\sup_u$ across a fixed-$u$ event. *(manuscript citation: App. E)*

2. **Text (clarity):** Insert explicit "no optimistic closure" paragraph and adjust the introduction of the projection step to emphasize that it is the mechanism ensuring $\|V\|_{\mathcal{H}_\ell} \leq B$ for all proxies used by the algorithm. *(§1–§2, Alg. §4)*

3. **Typos/labels:** Fix "Assumption 3.2" → "Assumption 2.1"; minor citation-name consistency; tighten phrasing in Contributions (2). *(pp. 2–4, References)*

**Closing.** We agree that the original proof sketch could be misread as taking a supremum over an event that depends on $u$. The revised Appendix E removes this issue entirely by bounding the relevant *operator norm* in one step, yielding the same confidence multiplier as stated and keeping the main results intact.

# Rebuttal to Reviewer xxAq

**Thank you** for the constructive feedback. Below we respond point-by-point. **High-level:** Our contribution is to remove the *optimistic closure* premise by *enforcing* boundedness via a projection step, while keeping optimism derived from the CME. The uniform confidence event applies to the models the algorithm *actually* uses on-policy; the regret scales with standard information gain.

## A. ON THE TWO WEAKNESSES

**(W1) "Projection step requires solving a QP, may hurt practicality."** The projection we use is the *Hilbert-ball projection* onto $\mathbb{B}_{\mathcal{H}_\ell}(B) = \{V \in \mathcal{H}_\ell : \|V\| \leq B\}$. It is not a generic QP:

- In a Hilbert space, $\mathrm{Proj}_{\mathbb{B}_{\mathcal{H}_\ell}(B)}(f) = f$ if $\|f\| \leq B$, and $\mathrm{Proj}_{\mathbb{B}_{\mathcal{H}_\ell}(B)}(f) = (B/\|f\|)\,f$ otherwise. Thus, when the proxy lies in $\mathcal{H}_\ell$ (which it does under our representer form), the projection is *radial scaling* with a single scalar factor.

- In coefficient form $f(\cdot) = \sum_{i=1}^m \alpha_i\,\ell(\cdot, s_i)$, $\|f\|_{\mathcal{H}_\ell}^2 = \alpha^\top K \alpha$. The extra cost is computing $\alpha^\top K \alpha$ and possibly multiplying $\alpha$ by $B/\sqrt{\alpha^\top K \alpha}$—both $O(m^2)$ given the Gram $K$.

- When we use the (equivalent) Tikhonov version to stabilize the update, we re-use the Cholesky factor of $K + \rho I$ that is *already* built for CME and variance ($\sigma$) computations. Within an episode, this factorization is shared across stages, so the marginal cost is negligible relative to the CME step.

We will make this implementation note explicit and add per-step complexity.

**(W2) "Limited technical novelty beyond kernel/linear RL."** While our analysis reuses standard tools (information gain, CME), two technical choices are new in this combination: (i) a *uniform* CME-based confidence event coupled with an *explicit projection* so that optimism is valid without assuming closure; (ii) a proof route that avoids covering arguments for the value class by working *only* with the RKHS ball actually enforced by the algorithm. We will emphasize these points in Sec. 1 and add a short comparison table to related work (kernel RL and linear MDPs).

## B. ANSWERS TO THE QUESTIONS

**Q1. Which parts of the proof change when replacing optimistic closure by projection? Any special treatment?** Yes—three places are affected, all conceptually simple:

1. **Uniformity of the confidence event:** The CME bound (Thm. 3.3) is uniform for all $V \in \mathbb{B}_{\mathcal{H}_\ell}(B)$. The projection *guarantees* the iterate $V_{h,t} \in \mathbb{B}_{\mathcal{H}_\ell}(B)$ at every step, so the bound applies to the proxy actually used. Under optimistic closure this guarantee is assumed; here we enforce it.

2. **Optimistic backup:** We use

$$\widetilde{Q}_{h,t}(z) = r_h(z) + \sup_{\|V\| \leq B} \langle \widehat{\mu}_h(z), V \rangle_{\mathcal{H}_\ell} + \beta_{h,t}\sigma_{h,t}(z) = r_h(z) + B\,\|\widehat{\mu}_h(z)\|_{\mathcal{H}_\ell} + \beta_{h,t}\sigma_{h,t}(z),$$

   and then project the implied value proxy onto $\mathbb{B}_{\mathcal{H}_\ell}(B)$. Under closure, the projection is the identity and this reduces to the closed-form update used in CME-RL.

3. **Regret telescoping:** Unchanged structurally; we only add the projection non-expansiveness $\left\|\mathrm{Proj}_{\mathbb{B}_{\mathcal{H}_\ell}(B)}(f) - \mathrm{Proj}_{\mathbb{B}_{\mathcal{H}_\ell}(B)}(g)\right\|_{\mathcal{H}_\ell} \leq \|f - g\|_{\mathcal{H}_\ell}$ to propagate optimism through the DP steps.

No additional concentration or covering arguments are needed beyond those already used to control $\sigma$ and $\widehat{\mu}$.

**Q2. Relation to the linear/parametric case where one enforces bounded weights by $L_2$ projection.** In the linear case $V(\cdot) = \theta^\top \phi(\cdot)$ with $\|\theta\| \leq B$, Step 3 becomes the Euclidean projection $\theta \leftarrow \min\{1, B/\|\tilde{\theta}\|\} \tilde{\theta}$, and the UCB term is the familiar elliptical form. We will add a short subsection showing that our algorithm reduces to the standard LSVI-UCB update with weight projection and that our proof tracks the classic self-normalized argument, replacing $\|\theta\|$ by $\|V\|_{\mathcal{H}_\ell}$.

**Q3. Assumptions for the projection to "work appropriately"; role of anchor states.** Our theory needs only that the projection is onto the *true* RKHS ball $\mathbb{B}_{\mathcal{H}_\ell}(B)$. When using a finite dictionary ("anchor" states) to represent proxies, the projection is carried out in that span; this *does not affect validity* of the bound, because the uniform event and regret analysis are stated for the function actually used by the algorithm (the projected proxy in that span). A too-small dictionary may increase approximation error, which shows up in the standard $\varepsilon_B$ term; it does not break boundedness or optimism. We will clarify this in Sec. 4 and add a remark on Nyström/greedy dictionary growth and its effect on compute vs. approximation.

**Q4. Why does "no projection" (KOVI0) continue to lag instead of aligning after a burn-in?** Even if $V^\star \in \mathbb{B}_{\mathcal{H}_\ell}(B)$, the unprojected optimistic proxy can *persistently* leave $\mathbb{B}_{\mathcal{H}_\ell}(B)$ at poorly visited regions because $\beta_t \sigma(\cdot)$ need not shrink uniformly. This produces over-optimistic backups that keep the policy exploring high-uncertainty regions, which in turn maintains large widths and prevents alignment. The projection caps this runaway behavior at radius $B$, stabilizing the fixed point iteration and constraining exploration. We will add this explanation and a small synthetic example to the appendix (illustrating persistent overshoot without projection).

## C. CONCRETE EDITS WE WILL MAKE

1. **Algorithmic clarity and cost:** Add pseudocode with a dedicated line for the projection. Include the closed-form radial projection, its coefficient-space form, and complexity (reusing the factorization of $K + \rho I$).

2. **Linear case bridge:** New subsection showing the reduction to weight projection and elliptical UCB; include a proposition and two-line proof.

3. **Assumptions for projection:** Clarify that boundedness is enforced (not assumed), and that dictionary choice affects only approximation error $\varepsilon_B$; add a brief note on growing dictionaries / Nyström.

4. **Experiments:** Add an ablation comparing *with* vs. *without* projection across tasks, plus a small synthetic example where no-projection overshoots persistently; keep identical kernels and hyperparameters for fairness.

## D. MINOR FIXES

We will (i) define $z$ at L043 and $n$ at L065 when first used, (ii) increase the contrast/line width for KOVI0 in Fig. 1, and (iii) fix the formatting at L357.

## CLOSING

We appreciate the reviewer's observation that our method achieves sublinear regret while removing optimistic closure. The added clarifications, linear-case bridge, and implementation notes should address practicality and novelty concerns.

