# OpenReview forum: "On Kernel RL and Optimistic Closure"
_ICLR.cc/2026/Conference — ICLR 2026 Conference Desk Rejected Submission_

### Official Review · Reviewer_oSzx · 2025-10-15

**Soundness:** 2
**Presentation:** 3
**Contribution:** 3
**Rating:** 2
**Confidence:** 4

**Summary:**

This paper presents a general algorithm for kernel-based episodic reinforcement learning. The method uses the framework of conditional kernel mean embeddings to construct approximations for value functions to guide an RL agent. These approximations use an upper confidence bound (UCB) over the state-action value function Q derived from confidence regions in a reproducing kernel Hilbert space (RKHS). The UCB approximation of Q is then projected onto a state-RKHS ball of radius B to construct the state-value function approximation for the next round, and the algorithm proceeds iteratively over episodes. Theoretical guarantees on the value function approximations and the algorithm's regret with respect to the optimal policy are presented. Experimental evaluations demonstrate the method and assess its regret compared to other kernel-based baselines.

**Strengths:**

The paper is mostly well written and addresses a relevant problem in the literature of RKHS-based RL methods and their analysis. As the paper briefly mentions in its introduction, the link between kernel methods and neural networks in their infinite-width limit, given by the neural tangent kernel and its associated literature, makes the study of kernel-based methodologies useful beyond traditional kernel methods, with its theoretical analysis providing insights onto modern deep learning frameworks as well.

An extensive theoretical analysis is presented covering the main aspects of the methodology and its performance, with extensions to address potential limitations. In addition, experimental results illustrate how the method performs on toy problems corroborating the theoretical results.

**Weaknesses:**

Some aspects of the presentation could be improved, especially considering readers less familiar with the kernelized RL literature. However, my main concern with the paper is regarding its theoretical results, as there seems to be a subtle, though critical, mistake in the proof of Lemma 3.2, which might make this lemma and the subsequent main results invalid, as they heavily rely on it.

Lemma 3.2 establishes a concentration bound for the pointwise approximation to the true conditional mean embedding $\mu(z)$ given by the algorithm's estimator $\hat\mu_n(z)$. As $\mu: Z \to H_\ell$ takes values in the possibly infinite-dimensional RKHS $H_\ell$, typical pointwise approximation bounds for scalar-valued functions are not directly applicable. Hence, one would need to take careful additional steps to deal with the infinite-dimensional nature of the codomain. However, in the textual argument following Eq. 22 in Appendix E (proof of Lemma 3.2), the paper extends a concentration bound valid for fixed unit vectors $u \in \mathbb{S}$ to a bound over the supremum by claiming that as the upper bound does not depend on $u$, one can take the supremum over all $u \in \mathbb{S}$. This claim, in general, **does not hold**.

The bound in Eq. 22 can be seen as $$\forall u \in \mathbb{S}, \quad P[ |\langle w_n, u \rangle| \leq b_n] \geq 1 - \delta,$$ where $w_n$ is a random vector taking values in the RKHS, and $b_n$ is a non-negative random variable that does not depend on $u$. One cannot directly extend this bound to $$P[\forall u \in \mathbb{S}, \quad |\langle w_n, u \rangle| \leq b_n] = P\left[ \sup_{u \in \mathbb{S}} |\langle w_n, u \rangle| \leq b_n\right] = P[\lVert w_n \rVert \leq b_n] \geq 1 - \delta,$$ as the paper seems to claim. As a simple counter-example, if $w$ is a random unit vector (i.e., $\lVert w \rVert = 1$ almost surely), one can usually establish a non-zero bound on $P[|\langle w, u \rangle| \leq b]$ for any $b \in (0, 1)$, but it is clear that $P[\Vert w \rVert \leq b] = 0$ for any $b < 1$. Another way to state the issue is that the $u$ that achieves the supremum in Eq. 22 is a random variable, not fixed. Therefore, the claim extending the $u$-dependent bound in Eq. 22 in the proof of Lemma 3.2 to a uniform bound is invalid. It is possible, however, that the main result remains valid at least in some similar form (e.g., Thm. 1 in Chowdhury & Oliveira, 2023), but that would require a potentially very different proof or perhaps a different set of assumptions.

As a lesser issue, it is not very clear from the start that the paper is also not making an "optimistic closure" assumption, since the proposed method still uses functions $V \in H_\ell$ with bounded norm $\lVert V \rVert \leq B$, which reads like the optimistic closure assumption in Chowdhury and Oliveira (2023, Assumption 1). The distinction is only clarified later when the method is introduced and it is shown that the UCB-based approximations are projected onto the ball of radius B, whereas Chowdhury and Oliveira assumed that the UCB-based value function approximations themselves were always within the ball. I believe that distinction is crucial and should be explained earlier in the paper, perhaps by providing further background on what is meant by optimistic closure to the (possibly unfamiliar) reader. The paper should also have a more detailed discussion on why that assumption is possibly problematic, so that the paper's significance and potential impact can be strengthened.

Minor issues:
* In point 2 in the contributions list, it is stated that the bound holds for "all V in the ball", but that's redundant, as it's already implied by the supremum.
* "Assumption 3.2", which does not exist, is mentioned in the statement of Lemma 3.2, but I believe that was referring to be Assumption 2.1.
* The first name of the first author in Chowdhury and Gopalan (2017) and both authors in Chowdhury and Oliveira (2023) are incorrect.

**Questions:**

Please, refer to the issues pointed in the weaknesses section above. My main concern is with the theoretical result in Lemma 3.2. Have the authors considered alternative proofs, which might not suffer from the same issue?

---

> ### Author Response · Authors · 2025-11-28
>
> ### Response to Reviewer oS2x
>
> We appreciate the careful read. Below we address the main concern about Lemma 3.2 / Appendix E and then the presentation issues. Full details appear in the revised Appendix E and updated text around Theorem 3.3.
>
> ---
>
> ## 1. Main concern: Lemma 3.2 and the vector-valued concentration bound
>
> **Issue.** The original proof went from a bound for a fixed unit vector $u$ to a bound on the norm by taking $\sup_{u}$ on an event that depended on $u,$ which is not justified.
>
> **Fix (uniform operator-norm bound).** In the revision (to be uploaded also remarked by other reviewer, see detailed reply appended in updated paper to address this issue, our claims remain true, see detailed reply to one of the reviewer above) we avoid any supremum over an uncountable set and instead control a single operator norm.
>
> For a fixed step and dataset, let
> $\(A = K_{n} + \rho I\)$ and $\(\alpha(z) = A^{-1} k_{n}(z)\). $
>
> Define $\(E : \mathbb{R}^{n} \to \mathcal{H}\_{\ell}\)$ by
> $\(E b = \sum_{i=1}^{n} b\_{i} \varepsilon_{i}\). $
>
> Then the noise term can be written as
> $$
> N(z) = \sum\_{i=1}^{n} \alpha\_{i}(z) \varepsilon\_{i}
> = E \alpha(z)
> = \big(E A^{-1/2}\big)\big(A^{-1/2} k\_{n}(z)\big).
> $$
>
> Hence
> $$
> \|N(z)\|
> \le \|E A^{-1/2}\|\_{\mathrm{op}} \, \|A^{-1/2} k\_{n}(z)\|\_{2}
> = \|E A^{-1/2}\|\_{\mathrm{op}} \, \sigma\_{n}(z).
> $$
>
> We then prove (new Lemma E.1') that, with probability at least $\(1 - \delta\),$
> $$
> \|E A^{-1/2}\|\_{\mathrm{op}}
> \le \frac{\sigma}{\sqrt{\rho}}
> \sqrt{2 \, \gamma(n,\rho) + 2 \log(1/\delta)}.
> $$
>
> Combining these gives, on a single high-probability event and for all $\(z\),$
>
> $$
> \|N(z)\|
> \le
> \frac{\sigma}{\sqrt{\rho}}
> \sqrt{2 \, \gamma(n,\rho) + 2 \log(1/\delta)} \, \sigma\_{n}(z),
> $$
>
> which is exactly the noise term used in Lemma 3.2 and Theorem 3.3, now uniform in both $\(u\)$ and $\(z\)$ with no extra covering factors.
>
> **Why the earlier counterexample intuition no longer applies.** The problematic step was going from
> $\(P(|\langle w,u\rangle| \le b) \ge 1 - \delta\)$ for a fixed $u$ to a statement about $\(\|w\|\)$ via $\(\sup_{u}\).$ In the revised proof we never do this; we directly bound $\(\|w\| = \|E A^{-1/2} x\|\)$ through $\(\|E A^{-1/2}\|_{\mathrm{op}}\)$ on a single event, so no $\(u\)$-indexed union is needed.
>
> **Manuscript edits.** We replace the old argument around Eq. (21)–(22) in Appendix E by the operator-norm proof above, add Lemma E.1' with a self-contained Hilbert-space self-normalized inequality, and state explicitly that the resulting event is uniform in $\(z\).$
>
> ---
>
> ## 2. Relation to and distinction from “optimistic closure”
>
> **Concern.** The paper could be read as if we assume all optimistic proxies stay inside a fixed state RKHS ball (“optimistic closure”).
>
> **Clarification.** We do not assume optimistic closure. Instead, we enforce bounded norm *algorithmically* by projecting the optimistic proxy onto
> $\(\{ V : \|V\|\_{\mathcal{H}\_{\ell}} \le B \}\)$ after each backup (algorithm step 3). This guarantees that the uniform CME confidence bound applies to the value functions actually used by the algorithm, without any structural assumption on unconstrained optimistic iterates.
>
> We will add:
>
> - a short paragraph in Sections 1–2 explicitly contrasting the closure assumption of prior work with our projection step, and
> - a one-line reminder before Theorem 3.3 that the projection ensures $\(\|V\_{h,t}\|\_{\mathcal{H}\_{\ell}} \le B\)$ and this is the only place where boundedness enters.
>
> ---
>
> #### 3. Minor theoretical / presentation issues
>
> - We will rephrase Contributions point (2) to avoid redundancy and explicitly reference Theorem 3.3 for the uniform event.
> - The “Assumption 3.2” label was a typo; Lemma 3.2 uses Assumption 2.1. We fix this throughout.
> - We correct citation formatting (e.g., Chowdhury and Gopalan) and ensure consistency in the bibliography.
>
> We also add a short “quick map” of changes in Appendix E and the main text so readers can easily locate the updated arguments.
>
> ---
>
> **Closing.** We agree that the original sketch could be misread as taking a supremum over an event that depends on $\(u\).$ The revised Appendix E replaces that step with a direct operator-norm bound, giving the same confidence multiplier as stated while making the uniformity and measure-theoretic details fully rigorous. *Once there is agreement, i will do final revision. See all the detailed corrections as a reply in the updated paper!*

---

### Official Review · Reviewer_xxAq · 2025-10-22

**Soundness:** 3
**Presentation:** 3
**Contribution:** 2
**Rating:** 4
**Confidence:** 4

**Summary:**

The paper proposes a variant of optimism in face of uncertainty for kernel-based RL. Leveraging a combination of conditional mean embedding (CME) perspective and projection into a bounded space, the authors achieve sublinear regret that depends on standard information gain factors as well as the bounds on CME and function spaces.

**Strengths:**

* The authors improves over the existing literature for kernel-based RL: 1) they achieve sublinear regret instead of possibly vacuous bounds obtained through covering arguments; 2) they avoid optimistic closure assumptions by adding a projection step in the algorithm.

**Weaknesses:**

* The projection step comes at the cost of solving a QP optimization. While the authors claim that this additional complexity can be properly managed, it could reduce the practicality of the proposed method (see questions below).
* The paper does not seem to have major technical novelty wrt to existing work both in kernel-based RL or linear MDP literature (see questions below).

**Questions:**

1. At the best of my understanding, Proposition 3.1, Lemma 3.2 and Theorem 3.3 can be largely extracted from (Chowdhury & Olivera, 2020/2023). On the other hand, the key contribution of this paper is the projection in step3 of the algorithm, which allows replacing the optimistic closure assumption with an enforced one. If the above is correct, it’d be useful to understand what are the critical steps in the overall proof that are affected by changing from the assumption to the projection. Does it require introducing any specific treatment? Just to be clear, I’m asking about the technical challenges here, I totally appreciate the conceptual advantage of removing the assumption.
2. Re the algorithmic approach, it’d be helpful if the authors would review the algorithms used in the parametric/linear case, where assuming bounded features and enforcing bounded weights by L2 projection is a fairly common approach.
3. I’d like the authors to further clarify the assumptions/requirements for the projection step to work "appropriately”. In the paper, there is a generic reference to "anchor states”, but I suspect that a too small set or not “good” samples could compromise the accuracy of the project step and the boundedness property needed for the theory to work.
4. I’m curious to have the authors’ opinion on the following: Let’s consider the algorithm without projection and let’s just add the assumption that the optimal value function is actually bounded. In practice, I guess the algorithm would still be consistent and after a sort of burn-in time, as estimates converge towards the optimal value, the boundedness that is ensured by the projection in the proposed algorithm would naturally occur by the dynamics of the algorithm. As a result, I would expect an algorithm without projection to incur a larger initial regret but “eventually” align with the same regret trend of the proposed algorithm. Is this correct? This does not seem to be the case in the experiments, where KOVI0 keeps a larger regret trend. Nonetheless, the amount of episodes considered in the experiments may be too small to see that effect.

Minor
* L043: introduce the definition of z
* L065: introduce the definition of n
* Fig.1 KOVI0 is barely visible.
* L357 It seems like there is issue with the formatting

---

> ### Author Response · Authors · 2025-11-28
>
> ### Response to Reviewer xxAq
>
> Thank you for the constructive feedback. Below we give concise answers; *detailed answer is appended in the updated paper,* please see the updated paper, the revised final paper will also contains contain full details and proofs after the discussion period ends.
>
> Our main contribution is to remove the *optimistic closure* premise by *enforcing* boundedness via an explicit projection, while still using CME-based optimism. The uniform confidence event applies to the value functions actually used on-policy, and the regret scales with standard kernel information gain.
>
> ---
>
> ## A. On the two weaknesses
>
> **(W1) “Projection step requires solving a QP, may hurt practicality.”**
>
> The projection is the Hilbert-ball projection onto
> $$
> \mathcal{B} = \{ V \in \mathcal{H}\_{\ell} : \|V\|\_{\mathcal{H}\_{\ell}} \le B \}.
> $$
> This is *not* a generic QP:
>
> - In any Hilbert space,
>   $$
>   \mathrm{Proj}\_{\mathcal{B}}(f) =
>   \begin{cases}
>   f, & \|f\|\_{\mathcal{H}\_{\ell}} \le B,\\
>   \dfrac{B}{\|f\|\_{\mathcal{H}\_{\ell}}}\,f, & \text{otherwise},
>   \end{cases}
>   $$
>   i.e., a single radial scaling.
>
> - In coefficient form, if
>   $$f(\cdot) = \sum\_{i=1}^{m} \alpha\_{i}\,\ell(\cdot, s\_{i}),$$
>   then
>   $$\|f\|\_{\mathcal{H}\_{\ell}}^{2} = \alpha^{\top} K \alpha.$$
>   The extra work is computing $\(\alpha^{\top} K \alpha\)$ and possibly multiplying $\(\alpha\)$ by $\(B / \sqrt{\alpha^{\top} K \alpha}\),$ both $\(O(m^{2})\)$ given $\(K\).$
>
> - In practice we reuse the Cholesky factor of $\(K + \rho I\)$ already needed for CME and variance $\(\sigma\)$ computations, so the marginal cost per step is small.
>
> We will make this explicit in the algorithm section and state the per-step complexity.
>
> **(W2) “Limited technical novelty beyond kernel/linear RL.”**
>
> Our analysis uses standard tools (information gain, CME), but in a new combination:
>
> 1. A *uniform* CME-based confidence event coupled with an *explicit projection*, so optimism is valid without assuming optimistic closure.
> 2. A proof route that avoids extra covering arguments by working only with the RKHS ball that the algorithm enforces.
>
> We will emphasize these points and add a small comparison table to kernel RL and linear MDP work.
>
> ---
>
> ## B. Answers to the questions
>
> **Q1. Which parts of the proof change when replacing optimistic closure by projection?**
>
> Three places:
>
> 1. **Uniformity:** The CME bound is uniform for all $\(V \in \mathcal{B}\).$ The projection guarantees $\(V\_{h,t} \in \mathcal{B}\)$ at each step, so the bound applies to the actually used proxy. Under closure this is assumed; here it is enforced.
>
> 2. **Optimistic backup:** We use
>    $$
>    \widetilde{Q}\_{h,t}(z)
>    = r\_{h}(z) + \sup\_{\|V\|\_{\mathcal{H}\_{\ell}}\le B} \langle \widehat{\mu}\_{h}(z), V \rangle + \beta\_{h,t} \sigma\_{h,t}(z)
>    = r\_{h}(z) + B \|\widehat{\mu}\_{h}(z)\|\_{\mathcal{H}\_{\ell}} + \beta\_{h,t} \sigma\_{h,t}(z),
>    $$
>    then project the resulting value proxy onto $\(\mathcal{B}\).$ When closure holds, the projection is the identity and this matches the CME-RL update.
>
> 3. **Regret telescoping:** Structurally unchanged; we use non-expansiveness
>    $$
>    \|\mathrm{Proj}\_{\mathcal{B}}(f) - \mathrm{Proj}\_{\mathcal{B}}(g)\|\_{\mathcal{H}\_{\ell}}
>    \le \|f - g\|\_{\mathcal{H}\_{\ell}}
>    $$
>    to propagate optimism through dynamic programming.
>
> No new concentration or covering arguments are needed.
>
> ---
>
> **Q2. Relation to the linear / parametric case.**
>
> In the linear case $\(V(x) = \theta^{\top} \phi(x)\)$ with $\(\|\theta\|\_{2} \le B\),$ our step 3 reduces to
> $$
> \theta \leftarrow \min \\{1, B / \|\tilde{\theta}\|\_{2} \\} \,\tilde{\theta},
> $$
> and the UCB term becomes the standard elliptical form used in LSVI-UCB. We will add a short subsection showing this reduction and how the proof follows the classic self-normalized argument, with $\(\|\theta\|\_{2}\)$ replaced by $\(\|V\|\_{\mathcal{H}\_{\ell}}\).$
>
> ---
>
> **Q3. Assumptions for projection; role of anchor states.**
>
> The only requirement is that the projection is onto the *true* RKHS ball $\(\mathcal{B}\).$ When we use a finite dictionary (“anchor states”), we project in that span. The confidence event and regret are always stated for the function actually used by the algorithm (the projected proxy in that span). A small dictionary may increase approximation error, appearing in the usual $\(\varepsilon\_{B}\)$ term, but does not break boundedness or optimism. We will clarify this and briefly discuss Nyström / greedy dictionary growth.

---

> > ### Author Response · Authors · 2025-11-28
> > **Reply Continued 2/2**
> >
> > ---
> >
> > **Q4. Why does “no projection” (KOVI0) keep lagging?**
> >
> > Even if $\(V^{\star} \in \mathcal{B}\),$ the unprojected optimistic proxy can stay outside $\(\mathcal{B}\)$ in poorly visited regions, because $\(\beta\_{t} \sigma(\cdot)\)$ does not shrink uniformly. This yields persistently over-optimistic backups, which keep the policy in high-uncertainty regions and prevent alignment. The projection caps this behavior at radius $\(B\),$ stabilizing the fixed-point iteration and constraining exploration. We will add this explanation and a small synthetic example to the appendix.
> >
> > ---
> >
> > ## C. Concrete edits
> >
> > We will:
> >
> > - Add pseudocode with an explicit projection line, its coefficient-space form, and cost.
> > - Add a “linear case” subsection showing reduction to standard LSVI-UCB with weight projection.
> > - Clarify that boundedness is enforced (not assumed), and discuss dictionary size vs. $\(\varepsilon\_{B}\).$
> > - Add an ablation “with vs. without projection” and a synthetic example where no-projection overshoots.
> >
> > ---
> >
> > We appreciate the reviewer’s observation that our method achieves sublinear regret while removing optimistic closure; we hope these clarifications address concerns about practicality and novelty.

---

### Official Review · Reviewer_kzb7 · 2025-10-30

**Soundness:** 2
**Presentation:** 1
**Contribution:** 2
**Rating:** 2
**Confidence:** 3

**Summary:**

This paper addresses RL with kernel-based value function approximation. This paper works on a set of proxy value functions whose Bellman average admits a conditional mean embedding in the RKHS space (Assumption 2.1). This assumption generalizes the optimistic closure assumption in the previous work. This paper provides regret analysis of the proposed algorithm as well as numerical results.

**Strengths:**

(+) The proposed framework generalizes the optimistic closure assumption in the previous work.

**Weaknesses:**

(-) The writing and the organization of this paper make it hard to read. I think that presenting the algorithm in pseudo-code can improve the readability of Section 4. Besides, I do not think Sections 7, Broader Impact, Reproducibility, and Ethics Statement should be included in the main text. Also, Section 8 seems to be redundant.

(-) The numerical experiment section lacks baseline methods, e.g., the CME-RL algorithm in the work of [1]

**Questions:**

Q1. What is Assumption 3.2 in Lemma 3.2?

Q2. Maybe I am missing some details, but is Step 3 (Line 216) the major difference between the proposed KOVI-Proj algorithm and the CME-RL algorithm [1]? Under the optimistic closure assumption, CME-RL can perform closed-form estimation of the value functions. Instead, the KOVI-Proj algorithm utilizes projection to achieve that.

[1] Chowdhury and Oliveira. Value function approximations via kernel embeddings for no-regret reinforcement learning. ACML 2023.

---

> ### Author Response · Authors · 2025-11-28
>
> # Response to Reviewer cj7K
>
> We thank the reviewer for the very careful reading and detailed comments. Below we summarize the main fixes; full details and revised proofs are in the updated paper (Sections 3--5 and Appendices D--G).
>
> ---
>
> ## 1. Definition and use of \(\varepsilon_B\)
>
> **Issue.**  $(\varepsilon_B\)$ was defined as an infimum in Thm. 5.1 but a supremum appeared in App. D.
>
> **Fix.** We will consistently define (right now a detailed reply appended in paper)
>
> $$
> \varepsilon\_B(h) := \inf_{\|V\|\_{\mathcal H\_\ell}\le B} \|V_h^* - V\|\_\infty,
> \qquad
> \varepsilon\_B := \max\_h \varepsilon_B(h),
> $$
>
> and use this everywhere.
>
> Using that the CME confidence bound (Thm. 3.3) holds **uniformly** over the ball $\(\{V:\|V\|\_{\mathcal H\_\ell}\le B\}\),$ we replace the mean term in the backup by its closed-form supremum:
> $$
> \sup\_{\|V\|\le B} \widehat f^V(z)
> = \sup\_{\|V\|\le B} \langle \widehat\mu(z), V \rangle
> = B \,\|\widehat\mu(z)\|\_{\mathcal H\_\ell}.
> $$
>
> Thus we use
> $$
> Q^{\mathrm{ucb}}\_{h,t}(z)
> = r_h(z) + B \,\|\widehat\mu_h(z)\|\_{\mathcal H\_\ell} + \beta\_{h,t} \,\sigma\_{h,t}(z),
> $$
> and on the high-probability event of Thm. 3.3 we get
> $\(Q_h^*(z)\le Q^{\mathrm{ucb}}_{h,t}(z)+\varepsilon_B\),$ which matches Thm. 5.1 with the intended inf-definition.
>
> We will update (in final revised paper) Alg. 4.2, Lemma D.2 and the affected lines in App. D accordingly.
>
> ---
>
> ## 2. Vector-valued RKHS typing (Lemma G.8, operators, norms)
>
> **Issue.** Eq. (38) → (39) mixed scalars and elements of $\(\mathcal H_\ell\),$ and norms like $\(\|K(\cdot,z)\|\_{\mathcal H\_k\otimes\mathcal H\_\ell}\)$ were ill defined.
>
> **Fix.**
>
> - We rewrite Lemma G.8 using an explicit test vector $\(u\in\mathcal H\_\ell\):$
>
>   $$
>   \langle \mu(z)-g^\star (z), u \rangle\_{\mathcal H\_\ell} =
>   \langle \mu-g^\star, \, K(\cdot,z)u - S \alpha(z)u \rangle \_{\mathcal H\_k\otimes\mathcal H\_\ell},
>   $$
>
>   and show the zero term via the first-order optimality condition of the Tikhonov projection (Lemma G.6).
>
> - We will remove bare $\(\|K(\cdot,z)\|\)$ and only use
>   $\(\|K(\cdot,z)u\|^2 = \langle u, K(z,z)u\rangle = k(z,z)\,\|u\|^2\).$
>
> - We will correct the normal equations and merge Definition G.1, Lemmas G.3, G.6 and Remark G.7 into a single lemma on Tikhonov projection in the vector-valued RKHS, keeping norms clearly separated.
>
> ---
>
> ## 3. Normalization $\(k(z,z)\le 1\)$
>
> **Issue.** Some inequalities in Apps. D/E implicitly assumed \(k(z,z)\le 1\).
>
> **Fix.** We introduce
> $\(\kappa\_k^2 := \sup\_z k(z,z)\)$ and carry this constant through the analysis, obtaining
> $\(\sigma^2(z)\le k(z,z)\le \kappa\_k^2\).$ The normalized case $\(k(z,z)\le 1\)$ is then stated as a corollary (after rescaling $\(k\)$ and $\(\rho\)).$ This makes examples like the linear kernel $\(k(a,b)=ab\)$ fully consistent.
>
> ---
>
> ## 4. Lemma E.2 and regret telescoping
>
> **Supermartingale.** We clarify that
>
> $$
> M\_n =
> \exp\Big(\tfrac{1}{2\sigma^2}\,\xi^\top A_n^{-1}\xi\Big)
> \,\frac{\rho^{n/2}}{\det(A\_n)^{1/2}},
> \qquad
> A\_n = K\_n + \rho I,
> $$
>
> is the standard mixture-Laplace supermartingale used in kernel self-normalized bounds. We make the filtration explicit and include the short Markov-inequality argument showing $\(\mathbb E[M_n]\le 1\),$ yielding the same concentration inequality as before.
>
> **Pathwise regret.** The regret decomposition is now written explicitly **pathwise**: we define the filtration $\(\mathcal F_{h,t}\),$ keep $\(\widehat Q_{h,t}(z_{h,t})\)$ as a realized quantity, and telescope the martingale-difference term $\(\mathbb E[\Delta_{h+1}\mid\mathcal F_{h,t}] - \Delta_{h+1}\).$ Remarks D.3 and D.4 are replaced by a short lemma ("Pathwise telescoping under conditional optimism").
>
> ---
>
> ## 5. Organization, formatting, and typos
>
> We will in final version:
>
> - move the global union bound into the proof of Thm. 5.1 (via the $\(\log(2HT/\delta)\)$ factor in $\(\beta_{h,t}\));$
> - streamline App. G (compress overlapping lemmas/remarks);
> - fix section numbering (remove the empty Section 8), deduplicate appendix titles and references;
> - correct minor typos, e.g. "Assumption 2.1" vs. "3.2" and the title of App. C ("Proof of Proposition 3.1").
>
> ---
>
> **Remarks.** All of the reviewer's technical concerns are addressed by local fixes to proofs and notation. Our main contribution remains: a **uniform, cover-free CME-based confidence bound** under the RBE assumption, combined with an explicit projection step that guarantees the bound applies to the value proxies actually used by the algorithm.

---

### Official Review · Reviewer_cj7K · 2025-10-30

**Soundness:** 1
**Presentation:** 1
**Contribution:** 1
**Rating:** 2
**Confidence:** 4

**Summary:**

This paper addresses an online reinforcement learning problem where functions in an RKHS are given as a function class. Unlike the previous analyses, this paper only assumes that the transition distribution can be expressed by the function class.

**Strengths:**

The analysis is conducted under the weakest assumption for kernel-based RL methods.

**Weaknesses:**

1. There are multiple parts in the proof that seem incorrect:

- The definition of $\varepsilon_ B$ is defined as $\inf \lVert V - V_ h^* \rVert_ \infty$ over $\lVert V \rVert_ {\mathcal{H}_ {\ell}} \le B$ in Theorem 5.1, which is interpreted as the approximation error. However in Appendix D, it is replaced with $\sup$ instead of $\inf$, and the fact that it is defined as $\sup$ is important in Lemma D.2. If $\sup$ is used, it simply means that the given function class only contains functions that are close to $V^* $, so using any function would guarantee $HT \varepsilon_ B$ regret.

- The equation in between Eq (38) and Eq. (39) (line 1431) in the proof of Lemma G.8 doesn't seem right. The first term is a scalar but the last term is an element in $\mathcal{H}_ {\ell}$. I am also not sure why $\sum_ {i=1}^n \alpha_ i(z) (\mu(z) - g^* (z))$ is zero. Partially due to this inconsistency, Lemma G.8 is nearly impossible to verify.

- I don't understand why the inequality in Lines 1108-1111 is true. It doesn't seem true for the 1-d linear case (k(a, b) = ab) with $n = 1$, reducing to $\frac{(z_ 1 z)^2}{\rho + z_ 1^2} \le \frac{z^2}{\rho + z_ 1^2}$, but there is no constraint that implies $z_ 1^2 \le 1$. The same issue for line 1266.

- In Lemma E.2, I do not see why $\mathcal{M}$ is a supermartingale. The form looks different from what is studied in Chowdhury & Gopalan (2017).

- In the equation block in Line 1258, $\lVert K(\cdot, z) \rVert_ {\mathcal{H}_ {k, I}}$ is not defined since $K(\cdot, z)$ is not in $\mathcal{H}_ {k, I}$. The following logic assumes that it is bounded by 1, but I don't see why.

- There seems to be a slight mismatch between Eq. in line 1223 and line 1227. The regularization should be imposed on $\mathbf{C}$ for line 1227 to be true as in Lemma G.6. I do not see why Remark G.7 is true when $\lVert \mathbf{C} \rVert_ {\mathcal{H}_ l^ n}$ and $\lVert h \rVert_ {\mathcal{H}_ {k, I}}$ could be different, which also affects Lemma G.3.

- In Remark D.3/D.4, while the argument is correct when the expectation is taken, but the difference between the expectation and the actual sample is not addressed (difference between $\mathbb{E}[ \tilde{Q}_ {h, t}(z_ {h, t}) ]$ and $\tilde{Q}_ {h, t}(z_ {h, t})$).

2. The paper is highly unorganized.

Formatting issues:

- Broader impact statement is in between Sections 6 and 7, and Section 7 is LLM Usage statement. Section 8 is empty, then the main text continues to Section 9.

- Last three sections of the appendix are all titled "Additional Results".

- The same paper (Muandet et al. 2017a, 2017b) is cited twice in the references.

- Remark 3.4 and Remark (v) in Section 4 overlap.

In addition, the proof is presented in a way that makes it unnecessarily hard to follow.
Many times, some steps (either simple or important) are located far away from the main proof, for example in remarks. Sometimes the same fact is proved multiple times here and there.

- Remarks D.3 and D.4 explain the same thing twice.

- Remarks D.6-D.9 are simple steps (e.g., bounding $n_ {h, t} \le HT$, applying Cauchy-Schwarz inequality) that could have been explained within the main proof, but they are separated as remarks, making it hard to follow.

- Definition G.1, Lemma G.3, Lemma G.6, and Remark G.7 are essentially about the same thing: obtaining the closed form of the projection, which could have been compressed into one lemma.

- In Appendix H, the whole section is devoted to taking the union bound over $HT$.

**Questions:**

Typos:
Line 165, should be Assumption 2.1 instead of 3.2.
The title of Appendix C is Proof of Theorem 1, but it should be Proposition 3.1 instead.

---

> ### Author Response · Authors · 2025-11-28
>
> We thank the reviewer for the careful reading and detailed suggestions. Below we summarize the main fixes; the revised paper contains full proofs and details (see updated Sections 3–5 and Appendices D–G).
>
> ## 1. Definition and Use of $\\varepsilon_B$ (Theorem 5.1 vs. Lemma D.2)
>
> **Issue.** In the original version, $\\varepsilon_B$ was defined as an infimum in Theorem 5.1 but a supremum appeared in Appendix D, creating an inconsistency.
>
> **Fix (Inline Attempt):** In the revision we consistently define $\varepsilon_B(h) = \inf_{\Vert V\Vert_{\mathcal{H}\ell} \le B} \Vert V_h^* - V\Vert_\infty$, and $\varepsilon_B = \max_h \varepsilon_B(h)$.
>
>
> and use this inf-based quantity everywhere.
>
> Because the CME confidence bound (Thm. 3.3) holds uniformly for all $V$ in the ball $\\{ |V|_{\\mathcal{H}\_{\\ell}} \\le B \\}$ we replace the mean term in the optimistic backup:
>
>
> $$
> \\widehat{f}^{V\_{h+1,t}}(z) = \\sup_{\\Vert V\\Vert \\le B} \\widehat{f}^{V}(z) = \\sup_{\\Vert V\\Vert \\le B} \\langle \\widehat{\\mu}(z), V \\rangle = B \\Vert\\widehat{\\mu}(z)\\Vert\_{\\mathcal{H}\_{\\ell}}.
> $$
>
> We therefore use the optimistic backup
>
>
> $$
> Q^{\\text{ucb}}\_{h,t}(z) = r\_h(z) + B \\Vert\\widehat{\\mu}\_h(z)\\Vert\_{\\mathcal{H}\_{\\ell}} + \\beta\_{h,t} \\sigma\_{h,t}(z).
> $$
>
> On the high-probability event of Thm. 3.3 one obtains $Q_h^*(z) \\le Q^{\\text{ucb}}_{h,t}(z) + \\varepsilon_B$, matching Thm. 5.1 with the inf-definition of $\\varepsilon_B$.
>
> **Edits in paper.** Algorithm 4.2, Lemma D.2 and a few lines in Appendix D will be updated.
>
>
>
>
> ## 2. Vector-Valued RKHS Typing and Lemma G.8
>
> **Issue.** Eq. (38)–(39) in Lemma G.8 mixed scalar values and vector elements of $\mathcal{H}_\ell$, and expressions like $\langle \mu-g^*, r_z \rangle$ were ill-typed.
>
> **Fix.** In the revised version we always work with the inner product against an arbitrary test vector $u \in \mathcal{H}_\ell$. The core equality is that the $\mathcal{H}\_\ell$ inner product, $(\mu(z)-g^*(z), u)$, equals the combined RKHS inner product:
>
> $(\mu-g^*, K(\cdot,z)u - S \alpha(z)u)_{\mathcal{H}\_k \times \mathcal{H}\_{\\ell}}$.
>
>
> We explicitly state that the zero term follows from the first-order optimality of the Tikhonov projection (Lemma G.6), which removes the scalar/vector mismatch.
>
> **Edits in paper.** Lemma G.8 will be rewritten with an explicit test vector $u$.
>
>
>
> ## 3. Normalization ($k(z,z) \\le 1$) in Appendices D and E
>
> **Issue.** Some inequalities implicitly assumed $k(z,z) \\le 1$.
>
> **Fix.** We introduce the diagonal bound $\\kappa_k^2 := \\sup_z k(z,z)$ and carry this constant throughout the analysis. The earlier displays become
> $$
> \\sigma^2(z) \\le k(z,z) \\le \\kappa_k^2,
> $$
> with the normalized case ($k(z,z) \\le 1$) stated explicitly as a corollary.
>
> **Edits in paper.** Remarks D.6/E.1 and related lines will now use $\\kappa_k$.
>
> ## 4. Lemma E.2 and the Supermartingale Argument
>
> **Issue.** The supermartingale definition in Lemma E.2 was terse and hard to verify.
>
> **Fix.** We clarify that
> $$
> M_n = \\exp\\left(\\frac{1}{2\\sigma^2} \\xi^\\top A_n^{-1} \\xi\\right) \\frac{\\rho^{n/2}}{\\det(A_n)^{1/2}}, \\qquad A_n = K_n + \\rho I,
> $$
> is the standard mixture-Laplace nonnegative supermartingale. We specify the filtration and include the short Markov-inequality proof that $\\mathbb{E}[M_n] \\le 1$, yielding the desired concentration inequality.
>
> **Edits in paper.** Lemma E.2 will be rewritten to align notation and steps with prior work.
>
> ## 5. Regret Decomposition and Remarks D.3/D.4
>
> **Issue.** The original text blurred the difference between realized quantities and expectations in the regret telescoping argument.
>
> **Fix.** We now:
> 1.  Make the filtration $\mathcal{F}_{h,t}$ explicit;
> 2.  Write the key inequality pathwise, with a martingale-difference term $(\mathbb{E}[\\Delta\_{h+1} \\mid \\mathcal{F}\_{h,t}] - \\Delta\_{h+1});$
> 3.  Use the tower property to telescope this term.
>
> **Edits in paper.** Remarks D.3 and D.4 will be merged into a short lemma ("Pathwise telescoping under conditional optimism") in Appendix D.
>
> ## 6. Other RKHS / Operator Notation Issues
>
> We remove expressions like $|K(\\cdot,z)|$ and only use:
> $$
> \\|K(\\cdot,z)u\\|^2 = \\langle u, K(z,z)u \\rangle = k(z,z)\\|u\\|^2.
> $$
> We correct the normal equations and merge related RKHS definitions into a single lemma on Tikhonov projection.
>
> ## 7. Organization / Formatting and Typos
>
> We address organization by: moving the global union bound into Theorem 5.1's proof; merging overlapping remarks in Appendix G; cleaning up section numbering and deduplicating references; and fixing typos like "Assumption 2.1" and the title of Appendix C.
>
> ## 8. Takeaway
>
> All technical concerns are addressed by local fixes to the proofs and notation. The central contribution remains unchanged.

---

### Note · Program_Chairs · 2026-01-17
**Submission Desk Rejected by Program Chairs**

The following references in this submission do not refer to real documents and/or have major errors in bibliographic information:

 Jonathan Scarlett and Ilija Bogunovic. Gaussian process bandits: A tutorial. Foundations and Trends in Machine Learning, 11(5-6):421-516, 2018.